# RECODE-H: A Benchmark for Research Code Development with Interactive Human Feedback

**Chunyu Miao[1], Henry Peng Zou[1,*,†], Yangning Li[2,*,†], Yankai Chen[1,3,4,†], Yibo Wang[1,†], Fangxin Wang[1], Yifan Li[5], Wooseong Yang[1], Bowei He[4,6], Xinni Zhang[5], Dianzhi Yu[5], Hanchen Yang[1], Hoang H Nguyen[1], Yue Zhou[1], Jie Yang[1], Jizhou Guo[7], Wenzhe Fan[1], Chin-Yuan Yeh[8], Panpan Meng[9], Liancheng Fang[1], Jinhu Qi[1], Wei-Chieh Huang[1], Zhengyao Gu[1], Yuwei Han[1], Langzhou He[1], Yuyao Yang[1], Yinghui Li[2], Hai-Tao Zheng[2], Xue Liu[3,4], Irwin King[5], Philip S. Yu[1]**

[1]University of Illinois Chicago [2]Tsinghua University [3]McGill University [4]MBZUAI
[5]The Chinese University of Hong Kong [6]City University of Hong Kong
[7]Shanghai Jiao Tong University [8]National Taiwan University [9]Xi'an Jiaotong University

`{cmiao8, pzou3, ychen588, ywang633}@uic.edu`

## Abstract

Large language models (LLMs) show promise for supporting scientific research implementation, yet their ability to generate correct, executable code remains limited. Existing works largely adopt one-shot settings, ignoring the iterative and feedback-driven nature of realistic scientific research workflows. To address this gap, we present RECODE-H, a benchmark of 102 tasks from research papers and repositories that evaluates LLMs through multi-turn interactions with human feedback. It includes structured instructions, unit tests, and a five-level feedback hierarchy to reflect realistic researcher–agent collaboration. We further present ReCodeAgent, a framework that integrates feedback into an iterative code-generation process. ExperimentsExperiments[1]. with leading LLMs, including GPT-5, Claude-Sonnet-4, DeepSeek-V3.1, and Gemini 2.5, show substantial performance gains with richer feedback, while also highlighting ongoing challenges in generating complex research code. RECODE-H establishes a foundation for developing adaptive, feedback-driven LLM agents in scientific research implementation.

## 1 Introduction

Large language models (LLMs) have been increasingly adopted across the scientific research pipeline, assisting tasks from ideation to writing (Zhang et al., 2025b; Si et al., 2024; Chen et al., 2025b; Li et al., 2025b; Wu et al., 2025b; Zou et al., 2025b; Zhang et al., 2025a). However, generating correct and executable research code remains a difficult problem, not only because it requires long-range reasoning and robust verification (Padigela et al., 2025; Starace et al., 2025; Zhu et al., 2025c), but also because the input contexts in research settings are often complex, indirect, and noisy. Research papers describe methods through high-level narratives, mathematical formulas, and domain-specific conventions, while leaving many implementation details implicit. As a result, translating these fragmented and underspecified descriptions into functional code remains a fundamental challenge for current LLMs (Li et al., 2025c;a).

Existing benchmarks for research code generation (Zheng et al., 2023; Sun et al., 2023; Toledo et al., 2025; Hua et al., 2025) primarily evaluate models in a non-interactive setting, where they are expected to produce correct code in a single response. This design neglects the crucial role of human feedback in realistic workflows: on the one hand, users often cannot fully specify their requirements

---

[1]We will open-source our benchmark, and code at ⭕ GitHub

| Domain | Benchmark | Repo | Unittest | Feedback | Task |
|---|---|---|---|---|---|
| **General Code Generation** | BigCodeBench | ✗ | ✓ | ✗ | Function Code Generation |
| | ConvCodeWorld | ✗ | ✓ | Verbal Feedback | Function Code Generation |
| | MINT | ✗ | ✓ | Lazy User Feedback | Function Code Generation |
| | InterCode | ✗ | ✓ | Execution Feedback | Function Code Generation |
| **Research Code Generation** | MLE-bench | ✗ | ✗ | ✗ | Machine Learning Engineering |
| | MLAgentBench | ✓ | ✓ | ✗ | Machine Learning Engineering |
| | PaperBench | ✓ | ✗ | ✗ | Reproduce ICML Papers |
| | SciReplicate-Bench | ✓ | ✓ | ✗ | Code generation for research |
| | RECODE-H | ✓ | ✓ | Hierarchical Researcher Feedback | Code generation for research |

Table 1: Overview of benchmarks and their characteristics. Categories are abbreviated for compactness: Our benchmark introduces a structured feedback hierarchy (Section 3.2) to support systematic evaluation of interactive research code generation under progressively richer forms of guidance.

in one shot. On the other hand, LLMs rarely generate perfectly aligned code on the first attempt (Zou et al., 2025b). In practice, effective implementation relies on multi-turn interactions, where models must accurately interpret and leverage iterative user feedback (Li et al., 2024d). However, current benchmarks focus solely on end-to-end correctness, without evaluating models' capabilities in interactive refinement.

To fill this gap, we introduce **RECODE-H** (**Research CO**de **DE**velopment), a benchmark designed to evaluate how LLMs generate and refine research code through interactive feedback with human researchers. The benchmark consists of 102 tasks drawn from real research papers across machine learning, natural language processing, computer vision, and computational science, each paired with its original codebase. Unlike simple function-completion tasks, these tasks focus on repository-level code generation, requiring models to implement classes, functions, or modules that correspond to methodological descriptions in real research codebases. Building RECODE-H is challenging (Cao et al., 2025), as it requires aligning paper descriptions with code implementations, selecting representative tasks, and maintaining expert-level quality at scale. We address these challenges through a hybrid LLM-assisted and human-curated pipeline. RECODE-H has three key features:

- **PhD-level difficulty and expert annotation.** All tasks are drawn from real research projects and manually curated to ensure clarity, realism, and high annotation quality.

- **Feedback-level controlled difficulty.** Task difficulty is systematically controlled through the structured feedback hierarchy as introduced in Section 3.2, enabling fine-grained evaluation of models' ability to leverage multi-turn feedback guidance.

- **Research method focus.** Tasks center on the faithful implementation of research methods, typically requiring the development of several functions up to entire classes, rather than isolated function completion.

In our experiments, we evaluate seven leading LLMs, including GPT-5, DeepSeek-V3.1, Claude-Sonnet-4, and the Gemini family, on RECODE-H. We also introduce ReCode Agent, a stronger baseline that leverages human feedback through structured multi-turn interactions. ReCode Agent progressively integrates diagnostic signals, refinement instructions, and correction feedback to guide model revisions, providing a more faithful simulation of real research workflows. Experimental results show that all models substantially benefit from interactive feedback, with even minimal diagnostic signals nearly doubling pass rates and recall compared to the no-feedback setting. For example, GPT-5's recall improves from 29.4% without feedback to 71.6% with the most detailed feedback, while DeepSeek-V3.1 shows a similar jump from 10.8% to 70.6%. Larger models, such as GPT-5 and DeepSeek-V3.1, demonstrate stronger adaptation to progressively richer feedback, while Claude-Sonnet-4 and Gemini lag behind. Increasing feedback granularity not only improves success rates but also accelerates convergence, enabling stronger models to solve tasks in fewer turns. Error analysis further reveals that failures are dominated by misinterpretation of paper instructions and gaps in domain knowledge, rather than syntax or integration issues. These findings highlight the effectiveness of ReCode Agent and structured feedback, establishing a solid baseline for future work on research code generation.

Overall, our work makes the following contributions:

- We introduce **RECODE-H**, the first benchmark for *multi-turn interactive research code generation*, providing a high-quality dataset that evaluates how LLMs generate and refine code under iterative human feedback.

- We propose **ReCode Agent**, a strong baseline that effectively leverages structured human feedback through iterative interaction to implement and refine research code in realistic repository settings.

- Through extensive experiments on seven leading LLMs, we reveal key limitations of current models and demonstrate the effectiveness of our method, offering insights for building future feedback-driven research agents.

## 2 RELATED WORK

**LLMs for Research Development.** LLM-based agents have been applied across the scientific workflow, from survey writing and idea generation to paper review (Wang et al., 2024b; Si et al., 2025; Weng et al., 2024; He et al., 2026; Zhang et al., 2025a). Early benchmarks focused on end-to-end workflows (Toledo et al., 2025; Kon et al., 2025; Chan et al., 2024; Chen et al., 2024c; Starace et al., 2025; Edwards et al., 2025), but implementation remains a major bottleneck: current models struggle to translate textual descriptions into executable code. Recent datasets such as SciReplicate (Xiang et al., 2025), ResearchCodeBench (Hua et al., 2025), and LMR-Bench (Yan et al., 2025) shift toward code-level evaluation. Our work builds on this direction by introducing a qualitatively new evaluation paradigm by incorporating **multi-turn, feedback-driven** interactions. RECODE-H is the first benchmark to systematically assess an agent's ability to iteratively revise and refine research code through a structured hierarchy of feedback, reflecting the iterative cycles of diagnosis, correction, and alignment that characterize real scientific practice.

**Interactive Code Generation.** In software engineering (SWE), benchmarks span a wide spectrum, from function-level code generation tasks (Chen et al., 2021; Austin et al., 2021; Chen et al., 2023a; Zhuo et al., 2024; Hendrycks et al., 2021; Athiwaratkun et al., 2022) to repository-level generation (Jimenez et al., 2023; Li et al., 2024a;b; Ding et al., 2023). Most evaluations adopt a one-shot setting, where models produce solutions without further interaction with humans. More recently, as summarized in Table 1, Han et al. (2025); Wang et al. (2023b); Yang et al. (2023) have explored multi-turn interactive code generation by introducing various forms of feedback. However, these tasks remain limited to relatively simple function-level generation within the SWE domain. In contrast, our benchmark targets research code generation at the repository level, where the nature and complexity of feedback differ substantially from those in existing SWE-oriented benchmarks.

## 3 BENCHMARK

RECODE-H is designed to evaluate the ability of large language models to generate and refine research code in realistic, feedback-driven workflows. It consists of 102 tasks drawn from research papers and their corresponding repositories, each paired with structured instructions, explanatory comments, and unit tests to ensure reproducibility and reliability. The detailed benchmark statistics are provided in Appendix B.

### 3.1 BENCHMARK CONSTRUCTION

We developed a collaborative framework for constructing RECODE-H. To support this process, we assembled an annotation team of 26 annotators. All annotators are Ph.D.-level researchers with at least one publication at top-tier computer science conferences and are familiar with the target methods or algorithms, as well as their implementation code. The annotation process follows a multi-step pipeline: (1) paper and code selection, (2) annotation of explanatory comments for code, (3) construction of code generation instructions, and (4) development of unit tests. This design ensures both the reliability and reproducibility of the benchmark. An annotated example could be found in Appendix N.

**Paper and Code Selection.** All papers and repositories in RECODE-H were manually curated by expert annotators across diverse AI research domains, including machine learning algorithms, NLP,

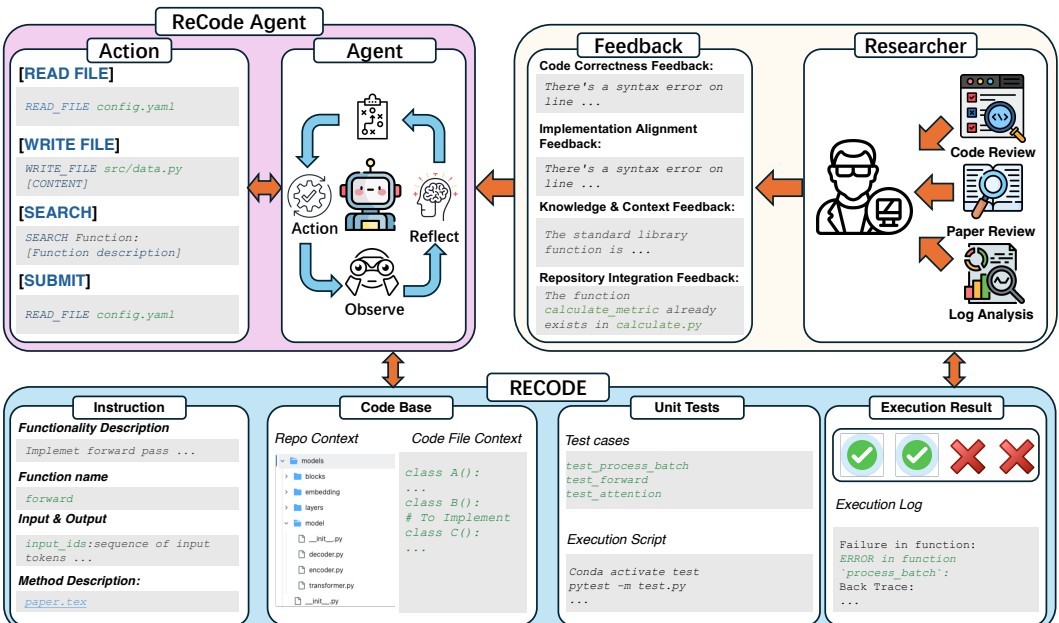

Figure 1: Illustration of the RECODE-H workflow, where LLM agents iteratively generate, test, and refine research code through structured researcher feedback.

computer vision, graph learning, time-series modeling, diffusion models, and recommendation systems (detailed distribution in Appendix B). The paper selection process follows a strict curation protocol to ensure realism, reproducibility, and alignment between research descriptions and implementation details.

Specifically, we include only papers that satisfy the following criteria: (1) the paper was published in 2023 or later; (2) the venue is a top-tier computer science conference (CVPR, ICML, NeurIPS, or ICLR)(3) the authors provide an official open-source repository that is correct and executable; and (4) the paper and repository exhibit a clear correspondence between methodological descriptions and code structure. Annotators verify each repository by running the official scripts to ensure the targeted components execute successfully within standard hardware constraints (less than 24 GB GPU memory). Only repositories meeting all criteria are included in RECODE-H, ensuring that the benchmark reflects realistic, well-specified, and executable research code tasks. The full paper list is revealed in Appendix K.

**Annotation of Explanatory Comments for Code.** The correspondence between code and paper descriptions is often distributed across multiple functions or classes, and the mapping is not always explicit. To address this, we add explanatory comments that clarify the relationship between the code and its associated paper. These comments will help produce consistent feedback, which is crucial for the reproducible evaluation process of RECODE-H.

To reduce annotation workload and ensure consistent comment content, we adopt a human-LLM collaborative approach. We employ `Gemini-2.5-Pro` to generate explanatory comments, taking as input both the paper text and the identified code segments. The comments primarily focus on three aspects: (1) the correspondence between the code and the paper description, (2) any discrepancies between the paper and the actual implementation, and (3) implementation details present in the code but absent from the paper. To ensure reliability, all generated comments are subsequently reviewed and validated for correctness.

**Construction of Code Generation Instructions.** Once the correspondence between the code and the paper description is established, we construct detailed code generation instructions. Each instruction specifies the target function name, its intended functionality, and the input and output parameters, including their names, data types, and semantic roles. For functionality descriptions, we mainly focus on illustrating the relationship between the function and the paper description and add

necessary explanations to the functionality, ensuring the instructions remain faithful to the original research context.

To promote consistency, we employ `Gemini-2.5-Pro` to generate an initial draft of the instruction in a standardized format. These drafts are then manually reviewed and refined by annotators to guarantee accuracy, clarity, and alignment with the source paper and code.

**Development of Unit Tests.** Unit tests are the core mechanism for evaluating functional correctness in RECODE-H. While LLMs (`Gemini-2.5-Pro`) were used only to draft initial candidate test cases; all tests were subsequently reviewed, corrected, and finalized by expert annotators. Annotators verified every test for correctness and completeness and enforced a strict 80% code coverage requirement for the canonical implementation. This ensures that the tests exercise the majority of logic branches rather than relying on superficial or syntactic checks. To accommodate complex research code functions, including those that internally perform complex computations, such as training updates, optimization routines, or data preprocessing workflows, we adopt a general differential testing strategy. Rather than evaluating full end-to-end system behavior, each unit test executes both the canonical implementation and the model-generated implementation under identical, fixed-seed conditions and compares their intermediate or final outputs. This comparison ensures functional equivalence within appropriate numerical tolerances, allowing us to verify the correctness of individual computational steps even when the overall task involves inherently dynamic or stateful processes. This approach provides a reliable and scalable mechanism for validating algorithmic correctness across a broad range of complex research tasks. These differential tests provide a reliable and reproducible evaluation framework across all task types in RECODE-H.

## 3.2 Feedback Hierarchy

In real-world scenarios, the feedback provided to an LLM agent after code execution can vary substantially. Factors such as the execution environment, the expertise of the feedback provider, and the effort invested in analyzing and writing feedback to the code all influence the form and quality of the feedback. Among these, the provider's expertise and the depth of analysis play the most significant roles in determining how informative the feedback is. To systematically evaluate LLM agents under varying feedback conditions, we design a five-level feedback hierarchy, where each level provides progressively more guidance:

- **Level 0:** Minimal feedback. The agent is only informed that code execution failed and that the execution result log is available.
- **Level 1:** Execution result plus a high-level error description that briefly characterizes the failure.
- **Level 2:** In addition to Level 1, an explanation of why the error occurred, offering diagnostic insight.
- **Level 3:** Extends Level 2 by providing natural language guidance that specifies how the implementation should be corrected. The feedback outlines the appropriate algorithmic modifications or design choices needed to align the code with the canonical implementation, without supplying any executable code.
- **Level 4:** The most detailed setting. Beyond Level 3, the exact ground-truth code snippet is provided. This level isolates the model's ability to accurately integrate a correct solution into the existing repository, serving as an upper-bound condition for evaluating instruction adherence and repository-level integration.

**Feedback Generation.** We employ `GPT-o4-mini` to simulate an expert and generate feedback to ensure the benchmark's reproducibility and scalability. After each code-generation round, the feedback model provides concise, actionable feedback. The feedback is conditioned on code execution results, including test outcomes and error logs. It also leverages the canonical code and annotated comments to clarify functionality and verify alignment with the paper's description. Using an LLM to produce feedback in this loop preserves the iterative nature of real research workflows while ensuring that feedback is standardized and reproducible across runs. We chose `GPT-o4-mini` because of its cost efficiency and high-quality feedback, as discussed in Appendix F.

**Evaluation Metrics.** We assess LLM agents on RECODE-H by functional correctness and code similarity. Functional correctness is measured with test cases using Mean Reciprocal Rank (MRR), which assigns a score of $\frac{1}{k}$ based on the first turn $k$ where correct code appears, and Recall@$n$, which

is the proportion of tasks solved correctly within $n$ turns. We also report the average proportion of test cases that pass to capture partial correctness. Code similarity is evaluated against canonical implementations using CodeBLEU (Ren et al., 2020), which combines lexical and structural signals, and CodeBERTScore (Zhou et al., 2023), which measures semantic similarity via embeddings.

# 4 RECODEAGENT

To evaluate our benchmark, we introduce a feedback-driven LLM code agent, ReCode Agent. The agent iteratively incorporates researcher feedback to generate or modify code, aiming to produce fully correct and executable implementations. ReCode Agent engages in multi-turn interactions with a researcher, refining its output until all tests pass or a predefined interaction limit is reached.

**Agent Strategy.** Our agent strategy follows the ReAct framework (Yao et al., 2023) and is organized into four stages. (1) Observation. It gathers the current repository state, execution logs from previously submitted code, and researcher feedback. (2) Reflection. It analyzes failures and gaps against the task specification, integrates feedback into actionable insights, and ensures consistency with repository constraints. (3) Planning. It formulates a concise, structured plan for the next step, specifying the goal, target files or spans, and the intended effect. (4) Action. It executes one operation from the predefined action space according to the plan. It should be noticed that, in our agent design, the use of "one-shot" refers to the human–agent interaction pattern rather than the agent's internal ReAct steps. Existing benchmarks support only a single human prompt, after which the agent operates autonomously. In RECODE-H, the agent engages in multi-turn human–agent collaboration, even though its internal reasoning–acting loop remains the same as standard ReAct.

**Memory Management.** To keep context length bounded under multi-round interaction, the agent maintains a memory similar to Reflexion (Shinn et al., 2023). We enforce a threshold on the number of recent memories to keep. When the memory exceeds the threshold, the agent compacts prior observations and actions into a concise summary that preserves the unresolved failures, design decisions, and generation context information. This memory compression promotes consistency across rounds while avoiding context bloat.

# 5 EXPERIMENTS

**Experimental Setup.** We evaluate ReCode Agent on RECODE-H using seven mainstream LLMs, spanning both reasoning and not-reasoning models. Specifically, we evaluated the GPT(OpenAI, 2025) family (`GPT-5`, `GPT-5-mini`, `GPT-5-nano`), the Gemini(Team et al., 2023) family (`Gemini-2.5-pro` and `Gemini-2.5-flash`), `Claude-Sonnet-4`(Anthropic, 2025) from Anthropic, and `DeepSeek-V3.1`(DeepSeek, 2025). Each model is assessed under multiple feedback conditions in a multi-turn setting to examine performance across varying levels of human interaction. For code generation, we fix the decoding temperature to 0 and top-$p$ to 1, ensuring deterministic outputs. The evaluation for each task proceeds for up to 10 rounds of feedback generation interaction. Within a single interaction turn between the LLM agent and the simulated human, we limit the agent to have at most 3 actions before automatic submission. And the threshold number of memory the LLM agent keeps is set to 5.

## 5.1 OVERALL RESULTS AFTER 10 INTERACTION TURNS

**Overview.** As shown in the Table 2, the richer feedback consistently improves LLM performance. With minimal feedback, models achieve low pass rates and limited correctness. As feedback becomes more detailed, from Levels 1 to 3, both success rates and efficiency improve, reflected in higher MRR, Recall, and test case pass rates. At the most detailed level, all models show substantial gains, with `GPT-5` and `Deepseek-V3.1` benefiting the most.

**Model size and capability play a clear role in performance.** Within the GPT family, larger models consistently achieve higher scores across all feedback levels, showing that scaling up enhances the ability to utilize feedback effectively. However, this trend is less evident in Gemini and Claude than in small-sized GPT models. Despite their relatively large sizes, their improvements from richer feedback are modest compared to `GPT-5` or `Deepseek-V3.1`. We attribute this to a lower feedback adoption rate as these models appear less efficient at incorporating feedback signals into subsequent

| Model | Feedback Level | Pass Rate | | | Similarity | |
| --- | --- | --- | --- | --- | --- | --- |
| | | MRR(%) | Recall(%) | Test Case(%) | CodeBLEU | CodeBERT |
| 🌀 OpenAI GPT | | | | | | |
| GPT-5-nano | Level 0 | 1.4 | 5.9 | 23.8 | 0.252 | 0.854 |
| | Level 1 | 3.0 | 10.8(+4.9) | 31.9(+8.1) | 0.258 | 0.821 |
| | Level 2 | 4.1 | 16.7(+5.9) | 39.4(+7.5) | 0.262 | 0.843 |
| | Level 3 | 4.6 | 19.6(+2.9) | 42.2(+2.8) | 0.266 | 0.817 |
| | Level 4 | 9.1 | 35.3(+15.7) | 54.7(+12.5) | 0.290 | 0.857 |
| GPT-5-mini | Level 0 | 1.3 | 19.6 | 42.3 | 0.262 | 0.870 |
| | Level 1 | 6.2 | 38.2(+18.6) | 62.8(+20.5) | 0.299 | 0.884 |
| | Level 2 | 7.8 | 47.1(+8.9) | 70.2(+7.4) | 0.303 | 0.887 |
| | Level 3 | 8.8 | 52.9(+5.8) | 73.2(+3.0) | 0.303 | 0.888 |
| | Level 4 | 11.1 | 66.7(+13.8) | 81.0(+7.8) | 0.309 | 0.889 |
| GPT-5 | Level 0 | 6.0 | 29.4 | 53.7 | 0.287 | 0.879 |
| | Level 1 | 7.5 | 45.1(+15.7) | 68.8(+15.1) | 0.294 | 0.884 |
| | Level 2 | 9.3 | 55.9(+10.8) | 77.4(+8.6) | 0.302 | 0.885 |
| | Level 3 | 10.6 | 63.7(+7.8) | 83.7(+6.3) | 0.304 | 0.886 |
| | Level 4 | 11.9 | 71.6(+7.9) | 84.3(+0.6) | 0.317 | 0.888 |
| Ⓐ Anthropic Claude | | | | | | |
| Claude-sonnet-4 | Level 0 | 5.2 | 14.7 | 35.1 | 0.287 | 0.885 |
| | Level 1 | 6.0 | 24.5(+9.8) | 48.5(+13.4) | 0.303 | 0.881 |
| | Level 2 | 7.5 | 33.3(+8.8) | 51.8(+3.3) | 0.300 | 0.888 |
| | Level 3 | 7.7 | 33.3(+0.0) | 53.5(+1.7) | 0.312 | 0.890 |
| | Level 4 | 11.4 | 48.0(+14.7) | 64.4(+1.7) | 0.335 | 0.892 |
| 🐋 Deepseek | | | | | | |
| DeepSeek-V3.1 | Level 0 | 5.1 | 10.8 | 30.7 | 0.292 | 0.886 |
| | Level 1 | 8.9 | 26.5(+15.7) | 47.2(+16.5) | 0.292 | 0.890 |
| | Level 2 | 12.1 | 43.1(+16.6) | 60.6(+13.4) | 0.301 | 0.891 |
| | Level 3 | 14.1 | 49.0(+5.9) | 57.6(-3.0) | 0.308 | 0.892 |
| | Level 4 | 21.0 | 70.6(+21.6) | 77.3(+19.7) | 0.341 | 0.897 |
| ✦ Google Gemini | | | | | | |
| Gemini-2.5-flash | Level 0 | 3.9 | 8.8 | 32.2 | 0.294 | 0.889 |
| | Level 1 | 7.4 | 27.5(+18.7) | 52.6(+20.4) | 0.300 | 0.891 |
| | Level 2 | 9.6 | 34.3(+6.8) | 58.7(+6.1) | 0.309 | 0.893 |
| | Level 3 | 12.0 | 47.1(+12.8) | 70.7(+12.0) | 0.311 | 0.893 |
| | Level 4 | 14.2 | 58.8(+11.7) | 77.6(+6.9) | 0.348 | 0.899 |
| Gemini-2.5-pro | Level 0 | 4.3 | 12.7 | 35.5 | 0.265 | 0.876 |
| | Level 1 | 6.1 | 16.7(+4.0) | 40.1(+4.6) | 0.278 | 0.880 |
| | Level 2 | 9.6 | 37.3(+20.6) | 57.6(+17.5) | 0.285 | 0.883 |
| | Level 3 | 10.4 | 37.3(+0.0) | 58.0(+0.4) | 0.291 | 0.874 |
| | Level 4 | 17.6 | 58.8(+21.5) | 70.6(+12.6) | 0.331 | 0.888 |

Table 2: This table reports performance of diverse LLMs on the RECODE-H benchmark over 10 rounds of interaction. Metrics include Mean Reciprocal Rank (MRR), Recall, average test case pass rate, and code similarity scores (CodeBLEU and CodeBERTScore). Stronger feedback consistently improves functional correctness and alignment with ground-truth implementations, with higher feedback levels leading to more effective error correction and refinement

generations, as discussed in Section 5.4. As a result, their final performance lags behind other large models that adapt more quickly and fully to iterative guidance.

**Non-linear gains across feedback levels.** We observe the improvements across feedback levels in Table 2 are non-linear, with the largest performance boost often observed from Level 0 to Level 1. For example, most models nearly double their Recall and test case pass rates once minimal diagnostic information is provided, indicating that even shallow guidance strongly accelerates error correction. In contrast, the gains from Level 2 to Level 3 and from Level 3 to Level 4 are more moderate, suggesting diminishing returns as feedback becomes increasingly detailed. Nevertheless, the highest level of feedback, which provides explicit code snippets, still delivers substantial improvements in functional correctness, especially for stronger models like `GPT-5` and `Deepseek-V3.1`. This pattern demonstrates that while structured feedback is highly valuable, the marginal benefits taper off as it approaches full supervision.

**Differences across model families.** The results also reveal notable differences across model families. The GPT family shows consistent and significant improvements as feedback becomes richer, with `GPT-5` and `GPT-5-mini` maintaining high scores across all metrics. In contrast,

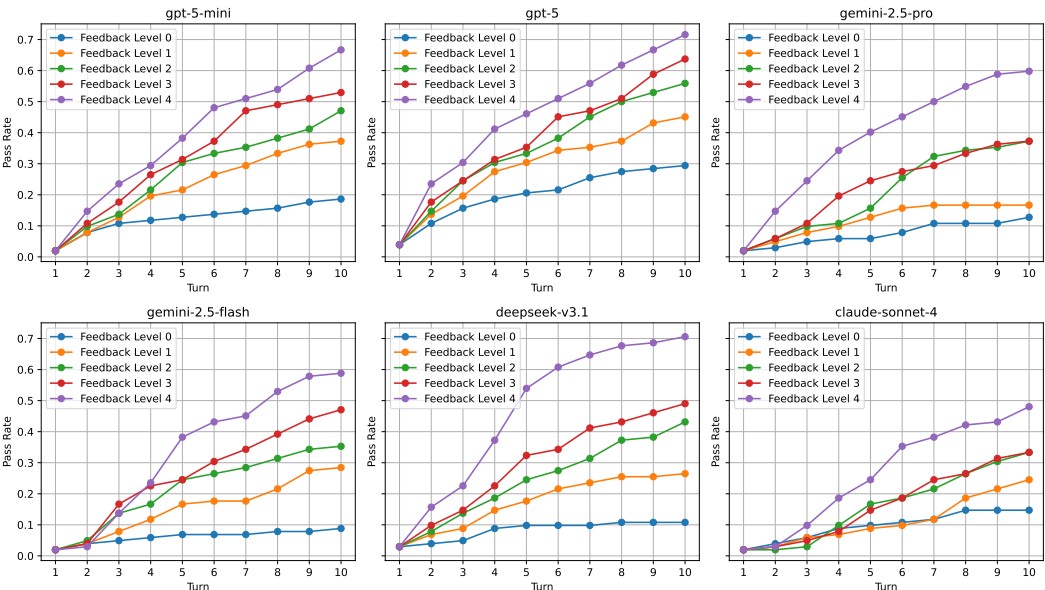

Figure 2: Pass rate trajectories across interaction turns under varying feedback levels. Richer feedback consistently boosts model performance, with the largest gains appearing in early turns. Stronger models like `GPT-5` and `Deepseek-V3.1` adapt more effectively, while `Gemini-2.5-flash` and `Claude-Sonnet-4` plateau earlier.

`Claude-Sonnet-4` shows only moderate gains, plateauing earlier and failing to fully utilize the detailed feedback. The Gemini family presents a split. `Gemini-2.5-flash` perform better than `Gemini-2.5-pro` across all levels. while `Gemini-2.5-pro` shows competitive results at higher feedback levels but still lags behind `GPT-5` and `Deepseek-V3.1`. Notably, `Deepseek-V3.1` demonstrates the largest relative improvement from Level 0 to Level 4, indicating a high sensitivity to feedback and strong adaptability in multi-turn interactions. These differences suggest that, beyond model size, the architecture and training approach of each family play a crucial role in determining how effectively models can incorporate iterative feedback into code generation.

**Recall and MRR improvements align.** A consistent trend across Table 2 is that Recall and MRR improve as feedback levels increase. `GPT-5`'s Recall rises steadily from 0.294 at Level 0 to 0.716 at Level 4, and its MRR improves from 0.060 to 0.119 over the same range.

## 5.2 PERFORMANCE DYNAMICS ACROSS INTERACTION TURNS

Figure 2 illustrates the test case pass rate with the increase of interactive turns. The trajectories clearly show that richer feedback not only improves the final success rate but also accelerates convergence in early turns. Both `GPT-5` and `DeepSeek-V3.1` rapidly increase their pass rates within the first 3–4 turns when provided with Level 3 or Level 4 feedback, whereas with Level 0 feedback their improvement remains gradual and plateaus at much lower levels. In addition, the first-turn results shown in Figure 2 also serve as our single-turn baseline. In this initial turn, the model receives only the task prompt without any interactive feedback, corresponding to the highlighted single-turn setting. As the figure indicates, subsequent turns that incorporate richer feedback yield substantial improvements over this baseline. After 10 turns, all models demonstrate markedly higher pass rates, confirming that multi-turn interaction provides a significant uplift over a single-turn scenario.

In contrast, smaller or weaker models exhibit slower gains and limited sensitivity to feedback richness, resulting in lower overall performance. `Gemini-2.5-pro` and `Claude-Sonnet-4` exhibit unstable performance gaps between feedback levels as interaction turns increase, which aligns with the inconsistent feedback adoption rates discussed in Section 5.4 and Appendix J. Among them, `Claude-Sonnet-4` shows the weakest performance trajectory across turns: the differences be-

tween feedback Levels 1–3 are far less pronounced than in other models, indicating difficulty in effectively leveraging moderate feedback.

## 5.3 ERROR ANALYSIS

To better understand the role of feedback in guiding multi-turn code generation, we conducted a fine-grained analysis of the errors encountered during the benchmark and the corrective signals associated with them. Our analysis revealed that errors can be systematically categorized into four types based on their root causes, each reflecting a different source of failure in the generation process.

**Type 1: Code Syntax and Runtime Errors.** This category encompasses fundamental programming issues that are independent of high-level algorithm design. Typical cases include syntax violations, runtime failures such as type mismatches or out-of-bounds indices, and basic logic flaws such as incorrect loop termination or variable misuse.

**Type 2:Paper and Instruction Misunderstanding Errors.** These errors arise when the generated code diverges from the intended method or fails to meet the benchmark requirements. Common manifestations include incorrect implementations of mathematical formulas, omission or misapplication of algorithmic steps, and mismatches in input/output formats or tensor shapes. They reflect a misalignment between the model's output and the research description it is expected to reproduce.

**Type 3: Missing Knowledge and Context Errors.** Some failures result from incomplete domain knowledge or implicit assumptions that are not explicitly stated in the task description. Examples include neglecting to use standard library functions, misunderstanding technical terminology, or overlooking repository-specific conventions. These errors highlight the model's difficulty in bridging gaps between textual instructions and domain practices.

**Type 4: Repository Integration Errors.** Finally, certain errors are caused not by the algorithm itself, but by poor integration with the surrounding codebase. These include reimplementing existing helper functions, misusing predefined modules or classes, or disregarding established coding conventions. Such errors limit the usability of generated code within the broader repository and emphasize the importance of context-aware generation.

| LLM | Type1(%) | Type2(%) | Type3(%) | Type4(%) |
|---|---|---|---|---|
| Gpt-5 | 11.35 | 34.04 | 50.25 | 4.36 |
| Gpt-5-mini | 11.53 | 27.06 | 55.22 | 6.19 |
| Gpt-5-nano | 20.82 | 37.32 | 34.95 | 6.90 |
| Gemini-2.5-pro | 14.94 | 39.91 | 37.34 | 7.82 |
| Gemini-2.5-flash | 16.16 | 26.15 | 49.41 | 8.28 |
| Deepseek-chat | 20.60 | 31.64 | 40.30 | 7.46 |
| Claude-sonnet-4 | 26.45 | 32.26 | 33.75 | 7.54 |

Table 3: Error type distribution across models on the RECODE-H benchmark.

In this experiment, we employ `GPT-5` to classify the reasons for errors. To ensure the reliability of the classification, we randomly sampled 100 cases and verified them with human annotators, achieving 98% agreement with `GPT-5`'s predictions, providing strong evidence of classification accuracy. As shown in Table 3, the majority of failures are attributed to higher-level semantic issues rather than low-level coding mistakes. Specifically, paper and instruction misunderstanding errors(Type 2), missing knowledge and context errors(Type 3) dominate across all models, whereas syntax and runtime errors(Type 1) occur less frequently, and repository integration errors(Type 4) are the least common. This distribution shows that modern LLMs have largely overcome basic coding challenges. However, they still struggle to align implementations faithfully with research descriptions and to bridge implicit domain knowledge. In addition, occasional but impactful gaps in repository awareness remain. A more detailed discussion of error type patterns and their implications is provided in Appendix I.

## 5.4 Feedback Adoption

We analyze how often models adopt the provided feedback and whether adoption results in a correct fix. `GPT-5` is employed as a classifier to determine whether the feedback is incorporated into the revised code and whether the targeted error is resolved. Our evaluation shows that both adoption and fix rates vary substantially across models, guidance levels, and feedback types. Detailed Table 10 and Table 11 supporting these findings are provided in Appendix J.

**Adoption as a necessary pathway.** Across all models and settings, nearly every successfully corrected error is one where the model explicitly adopted the provided feedback. Cases where errors were fixed without adoption are exceedingly rare, underscoring that feedback-driven improvement is the main driver of repair.

**Effect of guidance level on adoption.** Stronger feedback guidance increases the likelihood of adoption overall, but the magnitude of this increase differs substantially across models. Models that exhibit significant improvement in pass rates as guidance level rises, such as `GPT-5`, `GPT-5-mini`, and `DeepSeek-V3.1`, also show clear gains in feedback adoption. For instance, `GPT-5` adoption rises from 80.2% at Level-1 to 90.1% at Level-4, with `DeepSeek-V3.1` and `GPT-5-mini` following similar upward trajectories. This pattern suggests that the models most capable of leveraging feedback effectively are increasingly receptive to guidance as it becomes more explicit.

**Declining or fluctuating adoption among weaker improvers.** By contrast, models that show low pass rate under stronger guidance, such as `Gemini-2.5-pro` and `Claude-Sonnet-4`, exhibit a different pattern of adoption. For instance, the adoption rate of `Claude-Sonnet-4` decreases as the feedback guidance level increases, while the adoption rate of `Gemini-2.5-pro` fluctuates around 70%. These feedback adoption patterns align with their weaker performance in the benchmark.

**Simple and highly-specific feedback is adopted most.** Models exhibit the highest adoption rates for feedback targeting Code Correctness and Repository Integration errors. For example, `DeepSeek-chat` adopts nearly 80% of syntax-related feedback. In contrast, adoption rates of feedback for implementation alignment error (Type 2) are consistently lower across models, with `GPT-5-nano` showing the lowest rate at 56.1%. This pattern suggests that models struggle to address subtle logical errors that require a deeper understanding of the research method's intent.

## 5.5 Discussion and Insights

Our results highlight that the main barrier in research code generation is semantic fidelity. Models most often fail by misinterpreting methodological intent or missing domain knowledge, rather than through syntactic errors. Feedback effects are also highly non-linear. The minimal diagnostic signals yield the largest improvements, while richer natural-language guidance yields diminishing returns until explicit code snippets are supplied. Furthermore, feedback utilization varies sharply across models, indicating that effective incorporation of iterative guidance is a model-dependent capability. These observations suggest that future research should prioritize stronger semantic grounding and more stable multi-turn refinement. In practice, concise execution-level feedback is most effective, while human effort is best spent addressing conceptual misalignments.

## 6 CONCLUSION

We introduce RECODE-H, a benchmark for evaluating LLM-based research agents in realistic scientific workflows with feedback-driven code implementation. Unlike prior work focused on end-to-end pipelines or one-shot generation, RECODE-H emphasizes interactive evaluation, reflecting core aspects of real research practice. Our experiments reveal three findings: (1) scientific grounding, not coding, is the primary bottleneck, as models struggle with methodological interpretation and implicit domain knowledge; (2) feedback yields strongly non-linear gains, with minimal signals unlocking substantial improvements; and (3) feedback receptivity is a distinct capability, with stronger models benefiting from richer guidance while weaker ones plateau. Future work includes extending the benchmark to broader research stages, multi-agent settings, and human-in-the-loop feedback. Overall, effective research agents will require not only coding ability but also adaptive reasoning and sustained interaction with complex research environments.

ACKNOWLEDGMENTS

This work is supported in part by the National Science Foundation (NSF) under Grants III-2106758 and POSE-2346158.

Yangning is supported by the Basic Research Fund of Shenzhen City (Grant No. JCYJ20240813112009013).

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

## A  LLM USAGE STATEMENT

In this work, LLMs were used only to aid or polish writing. Specifically, initial drafts of the manuscript were written by the authors, after which LLMs were employed to enhance clarity and richness of expression. All content was subsequently reviewed and revised by the authors, and the final version reflects the authors' own corrections and approval. No parts of the research design, experiments, analysis, or results relied on LLMs.

## B  BENCHMARK STATISTICS

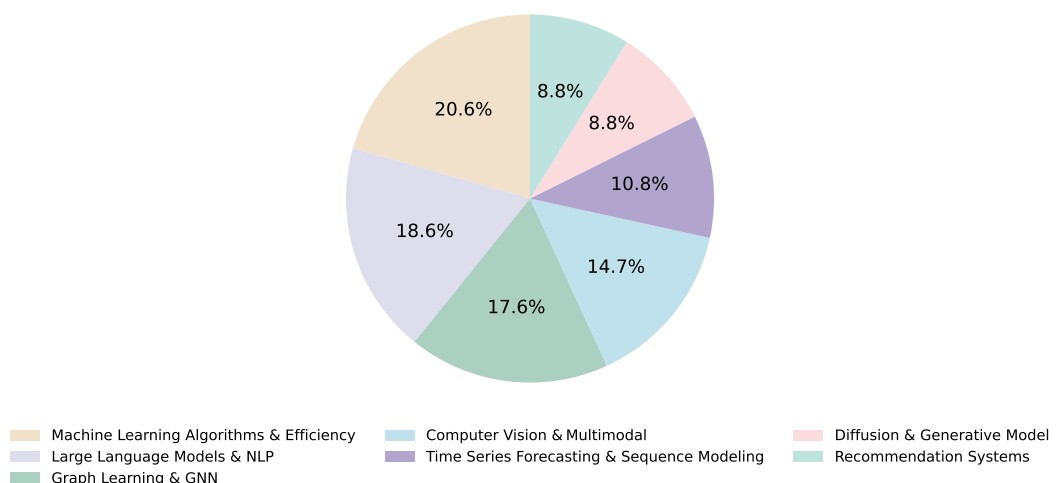

Figure 3: The domain of the tasks within RECODE-H.

Figure 3 illustrates the distribution of task domains in RECODE-H. The benchmark covers a diverse range of research areas. This balanced coverage highlights RECODE-H emphasis on capturing the breadth of modern AI research.

## C  DATA CONTAMINATION ANALYSIS

A common concern in benchmarks is that the evaluated LLMs may have been exposed to the source papers or repositories during pre-training, particularly for works published before late 2023. To examine whether such contamination materially affects model performance, we analyze results stratified by the publication year of the underlying research papers. Our dataset includes tasks from 2023, 2024, and 2025, enabling a coarse but meaningful assessment of year dependent performance trends.

Table 4 reports evaluation metrics across years for seven LLMs. If pre-training contamination were a dominant factor, models would be expected to show substantially higher performance on 2023 tasks, which are the most likely to appear in pre-training corpora. Instead, we observe the opposite pattern across nearly all models: performance on 2023 tasks is consistently the lowest, 2024 tasks improve markedly, and 2025 tasks often match or exceed 2024 levels—despite being too recent to be included in any model's training data.

These findings suggest that contamination is unlikely to be a primary driver of model performance. Instead, differences across years more plausibly reflect (1) evolving research trends, (2) variation in methodological complexity, and (3) stronger alignment between newer tasks and modern model capabilities. While eliminating contamination entirely is infeasible for frontier models trained on broad web corpora, this analysis indicates that leakage does not explain the performance patterns observed in RECODE-H.

| Model | Year | MRR | Recall | Test Case | CodeBLEU | CodeBERT |
|---|---|---|---|---|---|---|
| GPT-5 | 2023 | 0.116 | 0.588 | 0.806 | 0.297 | 0.884 |
| | 2024 | 0.230 | 0.685 | 0.847 | 0.322 | 0.890 |
| | 2025 | 0.277 | 0.833 | 0.853 | 0.318 | 0.888 |
| GPT-5-mini | 2023 | 0.065 | 0.294 | 0.659 | 0.288 | 0.890 |
| | 2024 | 0.187 | 0.741 | 0.856 | 0.306 | 0.890 |
| | 2025 | 0.216 | 0.733 | 0.806 | 0.329 | 0.889 |
| GPT-5-nano | 2023 | 0.051 | 0.176 | 0.490 | 0.275 | 0.885 |
| | 2024 | 0.097 | 0.352 | 0.562 | 0.301 | 0.853 |
| | 2025 | 0.106 | 0.433 | 0.536 | 0.283 | 0.849 |
| DeepSeek | 2023 | 0.111 | 0.588 | 0.802 | 0.342 | 0.901 |
| | 2024 | 0.212 | 0.704 | 0.817 | 0.342 | 0.897 |
| | 2025 | 0.234 | 0.767 | 0.830 | 0.343 | 0.896 |
| Gemini-2.5 Pro | 2023 | 0.127 | 0.588 | 0.723 | 0.332 | 0.897 |
| | 2024 | 0.202 | 0.574 | 0.676 | 0.336 | 0.891 |
| | 2025 | 0.154 | 0.600 | 0.740 | 0.330 | 0.879 |
| Gemini-2.5 Flash | 2023 | 0.090 | 0.471 | 0.727 | 0.331 | 0.903 |
| | 2024 | 0.161 | 0.630 | 0.782 | 0.355 | 0.900 |
| | 2025 | 0.128 | 0.567 | 0.784 | 0.354 | 0.896 |
| Claude | 2023 | 0.087 | 0.412 | 0.561 | 0.336 | 0.891 |
| | 2024 | 0.109 | 0.500 | 0.675 | 0.334 | 0.894 |
| | 2025 | 0.130 | 0.467 | 0.622 | 0.337 | 0.892 |

Table 4: Performance of LLMs across paper publication years.

## D ReCodeAgent Tool Design

Recent LLM agent research has rapidly expanded to include advanced agent strategies (Yao et al., 2023; Huang & Caragea, 2025; Zou et al., 2025a;c), memory mechanisms(Huang et al., 2026), enhanced reasoning paradigms(Zhang et al., 2025a; Yang et al., 2025), and safety controls(Huang et al., 2025). While these directions improve capability, they also introduce architectural complexity and evaluation confounds. In contrast, ReCodeAgent follows a minimal design principle, restricting the agent to a small, well-defined set of repository interaction tools. This lightweight design reduces hidden state and emergent behaviors, improves reproducibility, and isolates the core problem of iterative code refinement under structured tool constraints. ReCodeAgent interacts with the repository through a fixed set of tools that enable file inspection, code retrieval, rewriting, and execution based validation. These tools define the agent's action space and support reproducible multi-turn code refinement.

**READ.** The `READ` tool returns the full contents of any repository file, allowing the agent to recover context and understand existing implementations before making changes.

**RETRIEVE.** The `RETRIEVE` tool retrieves the most relevant functions or code regions given a natural-language query, enabling the agent to locate where specific functionality is implemented across the repository.

**REPLACE.** The `REPLACE` tool rewrites the entire contents of the target file with newly generated code. Although earlier versions of the system also included a span-based `EDIT` tool for detailed modifications, empirical evaluation showed that models frequently misidentified edit boundaries or produced inconsistent patches, resulting in structural corruption. Whole-file rewriting via `REPLACE` consistently yielded more reliable results. Therefore, `REPLACE` is retained as the sole editing mechanism.

**SUBMIT.** The `SUBMIT` tool executes the current implementation and returns unit test outcomes and execution logs, providing the diagnostic signals that guide iterative refinement.

**BROWSE.** The `BROWSE` tool lists all files and directories in the repository, allowing the agent to inspect project structure and locate relevant modules.
These tools support ReCodeAgent's iterative workflow: inspecting repository context, locating rel-

evant code, applying revisions, and validating correctness. This structured tool interface ensures transparency and reproducibility in evaluating research code development.

# E   EVALUATION OF GENERATED FEEDBACK

We additionally conduct a small-scale study using real human feedback. We collect 50 intermediate human and LLM interaction samples in which annotators provide natural language feedback on code that fails unit tests, following the same iterative protocol used for LLM augmented feedback. Figure 4 shows that human feedback yields measurable but consistently smaller performance gains compared with LLM-augmented expert feedback. Human annotations are typically partial, noisy and identifying only surface level issues and lacking the deeper diagnostic reasoning or prescriptive correction that LLM augmented feedback provides. This qualitative gap is consistent with prior observations from MINT(Wang et al., 2023b) and CONVCODEWORLD(Han et al., 2025).

Although our benchmark uses simulated expert feedback for reproducibility, we acknowledge that it does not capture the full variability of real human feedback, including imprecision, underspecification, and heterogeneity in expertise or communication style. Our framework can in principle incorporate these phenomena by introducing controlled perturbations—such as probabilistic error injection, incomplete messages, stylistic variation, or expertise-conditioned feedback distributions—within the existing feedback-sampling pipeline.

We restrict human annotation to this analysis due to the high cost and variability of real feedback, which limits scalable and reproducible evaluation. LLM-augmented feedback therefore serves as a standardized and repeatable reference for the main benchmark, while extending the framework with more realistic noise remains an important direction for future work.

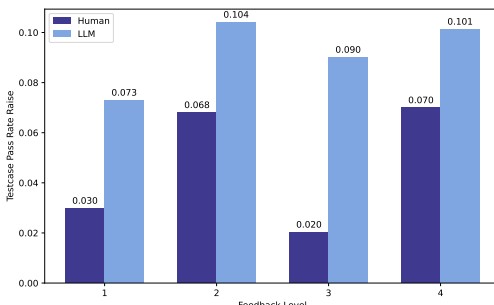

Figure 4: The rise of average pass rate of GPT-5 when guided by GPT-o4-mini and real human feedback.

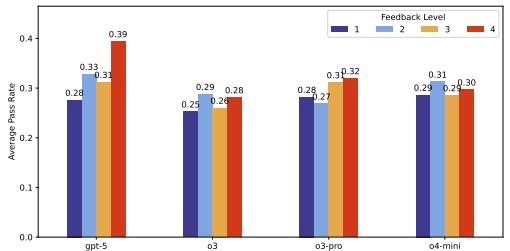

Figure 5: Average pass rate of GPT-5-mini when guided by different feedback models across feedback levels. GPT-5 feedback yields the strongest improvements, particularly at Level 4.

# F   ABLATION STUDY ON FEEDBACK MODEL

## F.1   FEEDBACK QUALITY

We conduct an ablation study to evaluate how different reasoning models used for feedback generation influence the overall interactive code generation process. Specifically, we test GPT-5-mini as the code generator while varying the feedback model among GPT-5, GPT-o3, GPT-o3-pro, and GPT-o4-mini.

Table 5 reports the average test case pass rates across these settings. Overall, GPT-5-mini achieves pass rates around 30% when guided by external feedback. Among the feedback models, GPT-5 provides the strongest improvements, yielding the highest pass rate 32%. GPT-o3-pro and GPT-o4-mini produce comparable results around 30%, while GPT-o3 feedback leads to the lowest performance of 27%.

Figure 5 further examines how pass rates evolve across different feedback guidance levels. The trends for GPT-5, GPT-o3, and GPT-o4-mini align with those observed in Table 2. Notably, GPT-5 demonstrates a clear advantage at Level 4 feedback, achieving a significant improvement over

| Feedback Model | Pass Rate |
|---|---|
| GPT-5 | 0.32 |
| GPT-o3 | 0.27 |
| GPT-o3-pro | 0.30 |
| GPT-o4-mini | 0.30 |

Table 5: Ablation study of average pass rate using different feedback models.

other models. This highlights `GPT-5`'s stronger ability to provide precise and corrective code-level feedback that accelerates convergence to correct implementations.

## F.2 FEEDBACK COST

| Model | Avg. Input Tokens | Avg. Output Tokens | Avg. API Cost($) |
|---|---|---|---|
| GPT-5 | 20 098.62 | 7781.91 | 0.102 |
| GPT-o3 | 20 099.38 | 2997.10 | 0.064 |
| GPT-o3-pro | 20 089.31 | 3267.83 | 0.663 |
| GPT-o4-mini | 20 103.95 | 5060.34 | 0.044 |

Table 6: Comparison of average token usage and API cost across different feedback models.

In this section we compare different feedback model in the aspect of cost. Table 6 shows that when evaluated on the same tasks, all feedback models process a similar number of input tokens, but differ in output size and API cost. GPT-5 provides the most effective feedback but at a moderate cost (0.102 \$ per feedback), while o3-pro is disproportionately expensive (0.663 \$). In contrast, o3 and especially o4-mini deliver much lower costs, with o4-mini offering the best cost-efficiency.

## G DATA LEAKAGE ANALYSIS

In generating feedback, we provided the LLMs with canonical code annotated with explicit comments to ensure correctness. While necessary for accurate feedback, this setup raises concerns about potential code leakage, particularly at feedback Levels 1-3. At these levels, the LLMs are expected to offer natural language guidance for correcting errors rather than reproducing ground truth implementations. To assess the risk, we sampled 1,000 feedback instances and evaluated leakage by detecting occurrences of ground truth code snippets within the feedback.

Our analysis shows that code leakage at Levels 1-3 is negligible, remaining below 2%. Results across models are summarized in Table 7.

| Model | Level 1 | Level 2 | Level 3 | Level 4 |
|---|---|---|---|---|
| GPT-5 | 0 | 0 | 0.00 | 0.020 |
| GPT-o3 | 0 | 0 | 0.01 | 0.305 |
| GPT-o4-mini | 0 | 0 | 0.00 | 0.195 |
| GPT-o3-pro | 0 | 0 | 0.02 | 0.330 |

Table 7: Code leakage rates across models. Leakage remains negligible ($< 1\%$) for Levels 1-3, but is more pronounced at Level 4 where ground-truth snippets are explicitly provided.

We further analyzed the extent of coverage—that is, the proportion of canonical code revealed in leaked snippets. These results are reported in Table 8. Again, leakage is effectively absent at Levels 1-2, minimal at Level 3, and substantially higher at Level 4.

## H EXTENDED ANALYSIS UNDER INCREASED INTERACTION BUDGETS

To further characterize model performance under varying feedback conditions, we extend the maximum number of researcher LLM interaction rounds from 10 to 15. This extended setting allows us to examine whether remaining errors stem from insufficient feedback or from iterative integration dynamics.

| Model | Level 1 | Level 2 | Level 3 | Level 4 |
|---|---|---|---|---|
| GPT-5 | 0 | 0 | 0.00 | 0.245 |
| GPT-o3 | 0 | 0 | 0.02 | 0.837 |
| GPT-o4-mini | 0 | 0 | 0.00 | 0.478 |
| GPT-o3-pro | 0 | 0 | 0.01 | 0.676 |

Table 8: Proportion of canonical code covered by leaked snippets. Leakage coverage is near-zero for Levels 1-3, but increases significantly at Level 4.

Table 9 reports Recall@15 for GPT-5 across all feedback levels. Performance consistently improves with additional turns, indicating that multi-turn refinement continues to contribute meaningfully beyond the default horizon.

| Feedback Level | Recall@15 (GPT-5) |
|---|---|
| Level 0 | 0.549 |
| Level 1 | 0.779 |
| Level 2 | 0.880 |
| Level 3 | 0.963 |
| **Level 4** | **0.972** |

Table 9: Recall@15 for GPT-5 under different feedback levels.

Lower feedback levels exhibit the largest marginal gains. *Level 0*, which provides only execution logs, improves substantially but remains constrained by weak error localization. *Level 1* and *Level 2* benefit from the extended horizon as well, reflecting that coarse and mid-level diagnostic information often requires multiple refinement cycles before errors are resolved.

Higher feedback levels demonstrate accelerated convergence. *Level 3* approaches saturation by 15 turns, driven by the effectiveness of targeted natural-language correction. *Level 4* reaches near-complete correctness, with remaining discrepancies attributable to multi-step repository integration, such as interface alignment and dependency resolution, rather than missing supervisory information.

The extended turn evaluation indicates that richer feedback enables faster and more reliable convergence, while lower levels remain limited by error localization and conceptual understanding challenges.

## I    ERROR TYPE ANALYSIS

The Table 3 reveals that while all models share common categories of mistakes, the distribution of these errors varies significantly across LLM families, producing distinct patterns of weakness. Broadly, low-level syntax and runtime errors are relatively uncommon, showing that most modern models have surpassed the stage of struggling with surface level coding rules. Instead, the majority of failures stem from higher level challenges, such as faithfully interpreting research descriptions or bridging implicit domain knowledge.

Within the GPT family, model size plays a decisive role. The largest variants, `GPT-5` and `GPT-5-mini`, produce relatively few syntax mistakes, with their errors dominated by missing knowledge and instruction misalignment. This shows that scaling tends to reduce shallow failures while pushing the challenge into semantic fidelity. By contrast, `GPT-5-nano` displays a very different profile: it is far more prone to low-level bugs and misunderstandings, indicating that smaller models lack the stability and consistency to translate complex descriptions into working code.

The Gemini models demonstrate another interesting split. `Gemini-2.5-flash` resembles larger GPT models, with its main failures concentrated in missing knowledge and context, reflecting a tendency to overlook implicit assumptions or repository-specific conventions. In contrast, `Gemini-2.5-pro` is far more vulnerable to instruction misunderstandings, producing errors that stem from misreading or misapplying the core methodological steps. This divergence between two variants of the same family highlights that architectural or training differences can lead to fundamentally distinct error patterns.

DeepSeek and Claude each illustrate contrasting limitations. `DeepSeek-V3.1` shows relatively balanced distributions of misunderstandings and knowledge gaps. `Claude-Sonnet-4`, on the other hand, stands out for having the highest rate of syntax and runtime errors among all models. While it shares semantic weaknesses with other LLMs, its disproportionate low-level fragility undermines reliability and signals weaker baseline robustness in code execution.

Taken together, these patterns emphasize that while the frontier of error types has shifted away from syntax toward semantic fidelity, different model families exhibit characteristic signatures of failure. Some models, like `GPT-5`, excel at suppressing shallow mistakes yet still falter on implicit domain knowledge, while others, like `Gemini-2.5-pro` or `Claude-Sonnet-4`, expose deeper struggles with faithfully grounding research instructions or ensuring execution stability.

## J  FEEDBACK ADOPTION ANALYSIS TABLE

Table10 provides statistics on feedback adoption and error resolve reate across different feedback levels. Table11 provides statistics on feedback adoption and error resolve reate across different error types.

| Model | Feedback Level | A (%) | NA (%) | AS (%) | AP (%) | ANS (%) | NAS (%) | NASP (%) | NANS (%) |
|---|---|---|---|---|---|---|---|---|---|
| GPT-5 | 1 | 80.2 | 19.8 | 50.9 | 5 | 24.3 | 0.5 | 0.5 | 18.9 |
| GPT-5 | 2 | 93 | 7 | 65.4 | 3.7 | 23.8 | 0 | 0 | 7 |
| GPT-5 | 3 | 91 | 9 | 66.3 | 6 | 18.6 | 0.5 | 0 | 8.5 |
| GPT-5 | 4 | 91.8 | 8.2 | 65 | 5.5 | 21.4 | 0 | 0 | 8.2 |
| GPT-5-mini | 1 | 68.8 | 31.2 | 35.9 | 3.5 | 29.4 | 0.9 | 0 | 30.3 |
| GPT-5-mini | 2 | 87.1 | 12.9 | 55.4 | 5 | 26.7 | 0 | 0 | 12.9 |
| GPT-5-mini | 3 | 96.3 | 3.7 | 61.9 | 10.2 | 24.2 | 0 | 0 | 3.7 |
| GPT-5-mini | 4 | 97.6 | 2.4 | 73.8 | 4.8 | 19 | 0 | 0 | 2.4 |
| GPT-5-nano | 1 | 45.2 | 54.8 | 30 | 4.6 | 10.6 | 0 | 0 | 54.8 |
| GPT-5-nano | 2 | 68.1 | 31.9 | 38.9 | 7.4 | 21.8 | 0 | 0 | 31.9 |
| GPT-5-nano | 3 | 68.7 | 31.3 | 45.7 | 5.3 | 17.7 | 1.2 | 0 | 30 |
| GPT-5-nano | 4 | 75.6 | 24.4 | 45.9 | 7 | 22.7 | 0 | 0 | 24.4 |
| Gemini-2.5-pro | 1 | 51.5 | 48.5 | 36.1 | 3 | 12.4 | 0 | 0 | 48.5 |
| Gemini-2.5-pro | 2 | 65.8 | 34.2 | 46.5 | 3.9 | 15.4 | 0 | 0 | 34.2 |
| Gemini-2.5-pro | 3 | 58.5 | 41.5 | 35.1 | 8.3 | 15.1 | 0 | 0 | 41.5 |
| Gemini-2.5-pro | 4 | 58.9 | 41.1 | 51.3 | 1.5 | 6.1 | 0.5 | 0 | 40.6 |
| Gemini-2.5-flash | 1 | 80.7 | 19.3 | 40.6 | 7.5 | 32.5 | 0 | 0 | 19.3 |
| Gemini-2.5-flash | 2 | 88.4 | 11.6 | 46.9 | 6.3 | 35.3 | 0 | 0 | 11.6 |
| Gemini-2.5-flash | 3 | 94.1 | 5.9 | 65.1 | 4.3 | 24.7 | 0.5 | 0 | 5.4 |
| Gemini-2.5-flash | 4 | 95.6 | 4.4 | 62.1 | 5.5 | 28 | 0 | 0 | 4.4 |
| DeepSeek-V3.1 | 1 | 75.3 | 24.7 | 47.1 | 7.3 | 20.8 | 0 | 0 | 24.7 |
| DeepSeek-V3.1 | 2 | 87.9 | 12.1 | 51.8 | 7.3 | 28.7 | 0 | 0 | 12.1 |
| DeepSeek-V3.1 | 3 | 88.1 | 11.9 | 65.2 | 6.3 | 16.7 | 0 | 0 | 11.9 |
| DeepSeek-V3.1 | 4 | 86.8 | 13.2 | 68.8 | 5.1 | 12.9 | 0 | 0 | 13.2 |
| Claude-sonnet-4 | 1 | 82 | 18 | 64.6 | 7.8 | 9.7 | 0.5 | 0.5 | 17 |
| Claude-sonnet-4 | 2 | 71.4 | 28.6 | 57.6 | 3.3 | 10.5 | 0.5 | 0 | 28.1 |
| Claude-sonnet-4 | 3 | 67.8 | 32.2 | 55.1 | 3.4 | 9.3 | 0.5 | 0.5 | 31.2 |
| Claude-sonnet-4 | 4 | 77.6 | 22.4 | 68.3 | 2 | 7.3 | 0.5 | 0 | 22 |

Table 10: Adoption and error-resolution outcomes across models and guidance levels. Columns denote: A = adopted (%), NA = non-adopted (%), AS = adopted and solved (%), AP = adopted and partially solved (%), ANS = adopted and not solved (%), NAS = non-adopted and solved (%), NASP = non-adopted and partially solved (%), and NANS = non-adopted and not solved (%).

| Model | Error Type | A (%) | NA (%) | AS (%) | AP (%) | ANS (%) | NAS (%) | NASP (%) | NANS (%) |
|---|---|---|---|---|---|---|---|---|---|
| GPT-5 | T1 | 96.7 | 3.3 | 72.5 | 2.2 | 22 | 0 | 0 | 3.3 |
| GPT-5 | T2 | 87.9 | 12.1 | 52.4 | 6.6 | 28.9 | 0.4 | 0.4 | 11.4 |
| GPT-5 | T3 | 89.1 | 10.9 | 66.3 | 4.5 | 18.4 | 0.2 | 0 | 10.7 |
| GPT-5 | T4 | 85.7 | 14.3 | 68.6 | 5.7 | 11.4 | 0 | 0 | 14.3 |
| GPT-5-mini | T1 | 94.7 | 5.3 | 66.3 | 3.2 | 25.3 | 0 | 0 | 5.3 |
| GPT-5-mini | T2 | 78.5 | 21.5 | 50.2 | 5.8 | 22.4 | 0.9 | 0 | 20.6 |
| GPT-5-mini | T3 | 89 | 11 | 56.7 | 6.2 | 26.2 | 0 | 0 | 11 |
| GPT-5-mini | T4 | 96.1 | 3.9 | 51 | 9.8 | 35.3 | 0 | 0 | 3.9 |
| GPT-5-nano | T1 | 69.9 | 30.1 | 53.9 | 6.2 | 9.8 | 0.5 | 0 | 29.5 |
| GPT-5-nano | T2 | 56.1 | 43.9 | 30.3 | 6.4 | 19.4 | 0.3 | 0 | 43.6 |
| GPT-5-nano | T3 | 66.7 | 33.3 | 40.1 | 6.2 | 20.4 | 0 | 0 | 33.3 |
| GPT-5-nano | T4 | 65.6 | 34.4 | 45.3 | 1.6 | 18.8 | 0 | 0 | 34.4 |
| Gemini-2.5-pro | T1 | 72.7 | 27.3 | 54.7 | 3.9 | 14.1 | 0 | 0 | 27.3 |
| Gemini-2.5-pro | T2 | 44.7 | 55.3 | 32.7 | 3.5 | 8.5 | 0 | 0 | 55.3 |
| Gemini-2.5-pro | T3 | 63.7 | 36.2 | 44.7 | 4.4 | 14.7 | 0.3 | 0 | 35.9 |
| Gemini-2.5-pro | T4 | 77.6 | 22.4 | 52.2 | 7.5 | 17.9 | 0 | 0 | 22.4 |
| Gemini-2.5-flash | T1 | 93.5 | 6.5 | 69.1 | 2.4 | 22 | 0.8 | 0 | 5.7 |
| Gemini-2.5-flash | T2 | 90.5 | 9.5 | 51.8 | 10.6 | 28.1 | 0 | 0 | 9.5 |
| Gemini-2.5-flash | T3 | 88.3 | 11.7 | 48.4 | 4.3 | 35.6 | 0 | 0 | 11.7 |
| Gemini-2.5-flash | T4 | 87.3 | 12.7 | 54 | 6.3 | 27 | 0 | 0 | 12.7 |
| DeepSeek-V3.1 | T1 | 87.4 | 12.6 | 70.5 | 2.4 | 14.5 | 0 | 0 | 12.6 |
| DeepSeek-V3.1 | T2 | 83.3 | 16.7 | 54.7 | 5 | 23.6 | 0 | 0 | 16.7 |
| DeepSeek-V3.1 | T3 | 83.5 | 16.5 | 56.3 | 7.9 | 19.3 | 0 | 0 | 16.5 |
| DeepSeek-V3.1 | T4 | 85.3 | 14.7 | 48 | 14.7 | 22.7 | 0 | 0 | 14.7 |
| Claude-sonnet-4 | T1 | 79.4 | 20.6 | 62.6 | 3.3 | 13.6 | 0.5 | 0 | 20.1 |
| Claude-sonnet-4 | T2 | 68.2 | 31.8 | 55.9 | 3.1 | 9.2 | 0 | 0 | 31.8 |
| Claude-sonnet-4 | T3 | 78.8 | 21.2 | 67.8 | 6.2 | 4.8 | 1.1 | 0.4 | 19.8 |
| Claude-sonnet-4 | T4 | 67.2 | 32.8 | 52.5 | 3.3 | 11.5 | 0 | 1.6 | 31.1 |

Table 11: Adoption and error-resolution outcomes across models and error type/ Columns denote: A = adopted (%), NA = non-adopted (%), AS = adopted and solved (%), AP = adopted and partially solved (%), ANS = adopted and not solved (%), NAS = non-adopted and solved (%), NASP = non-adopted and partially solved (%), and NANS = non-adopted and not solved (%).

## K  PAPER LIST

The paper list are in Tables 12, 13, 14 and 15.

| Paper Title | Venue |
| --- | --- |
| 1. PEDANTS: Cheap but Effective and Interpretable Answer Equivalence (Li et al., 2024f) | EMNLP 2024 |
| 2. Differential Transformer (Ye et al., 2024) | ICLR 2025 |
| 3. Mapping the Multiverse of Latent Representations (Wayland et al., 2024) | ICML 2024 |
| 4. TABDIFF: A Mixed-Type Diffusion Model for Tabular Data Generation (Shi et al., 2024) | ICLR 2025 |
| 5. Mapping the Multiverse of Latent Representations (Wayland et al., 2024) | ICML 2024 |
| 6. Task-Agnostic Machine-Learning-Assisted Inference (Miao & Lu, 2024) | NIPS 2024 |
| 7. Universal Neural Functionals (Zhou et al., 2024) | NeurIPS 2024 |
| 8. Dendritic Integration Inspired Artificial Neural Networks Capture Data Correlation (Liu et al., 2023a) | NeurIPS 2024 |
| 9. Transformers without Normalization (Zhu et al., 2025b) | CVPR 2025 |
| 10. SimPO: Simple Preference Optimization with a Reference-Free Reward (Meng et al., 2024) | NeurIPS 2024 |
| 11. Knowledge Distillation Based on Transformed Teacher Matching (Zheng & Yang, 2024) | ICLR 2024 |
| 12. Data Unlearning in Diffusion Models (Alberti et al., 2025) | ICLR 2025 |
| 13. It's Never Too Late: Fusing Acoustic Information into Large Language Models for Automatic Speech Recognition (Chen et al., 2024a) | ICLR 2024 |
| 14. GraphMETRO: Mitigating Complex Graph Distribution Shifts via Mixture of Aligned Experts (Wu et al., 2023) | NeurIPS 2024 |
| 15. Balanced Data Sampling for Language Model Training with Clustering (Shao et al., 2024) | ACL 2024 |
| 16. LESS: Selecting Influential Data for Targeted Instruction Tuning (Xia et al., 2024) | ICML 2024 |
| 17. Copula Conformal Prediction for Multi-step Time Series Forecasting (Sun & Yu, 2022) | ICLR 2024 |
| 18. Conformal Language Model Reasoning with Coherent Factuality (Rubin-Toles et al., 2025) | ICLR 2025 |
| 19. Data-efficient Fine-tuning for LLM-based Recommendation (Lin et al., 2024a) | SIGIR 2024 |
| 20. Language Models with Conformal Factuality Guarantees (Mohri & Hashimoto, 2024) | ICML2024 |
| 21. Universal Neural Functionals (Zhou et al., 2024) | NIPS 2024 |
| 22. Frequency-domain MLPs are More Effective Learners in Time Series Forecasting (Yi et al., 2023) | NIPS 2023 |
| 23. A Time Series is Worth 64 Words: Long-term Forecasting with Transformers (Nie, 2022) | ICLR 2023 |
| 24. Are Transformers Effective for Time Series Forecasting? (Zeng et al., 2023) | AAAI 2023 |
| 25. iTransformer: Inverted Transformers Are Effective for Time Series Forecasting (Liu et al., 2023c) | ICLR 2024 |
| 26. TimeMixer: Decomposable Multiscale Mixing for Time Series Forecasting (Wang et al., 2024a) | ICLR 2024 |

Table 12: Paper list part (a)

| Paper Title | Venue |
|---|---|
| 27. TimeXer: Empowering Transformers for Time Series Forecasting with Exogenous Variables (Wang et al., 2024a) | NeurIPS 2024 |
| 28. Large Language Models Are Zero-Shot Time Series Forecasters (Gruver et al., 2023) | NIPS 2023 |
| 29. ARC: A Generalist Graph Anomaly Detector with In-Context Learning (Liu et al., 2024d) | NIPS 2024 |
| 30. An Efficient Matrix Multiplication Algorithm for Accelerating Inference in Binary and Ternary Neural Networks (Dehghankar et al., 2024) | ICML 2025 |
| 31. FairHash: A Fair and Memory/Time-efficient Hashmap (Asudeh et al., 2023) | VLDB 2024 |
| 32. MALT Powers Up Adversarial Attacks (Melamed et al., 2024) | NeurIPS 2024 |
| 33. End-to-end Learnable Clustering for Intent Learning in Recommendation (Liu et al., 2024e) | NeurIPS 2024 |
| 34. ContextGNN: Beyond Two-Tower Recommendation Systems (Yuan et al., 2024) | ICLR 2025 |
| 35. Wukong: Towards a Scaling Law for Large-Scale Recommendation (Zhang et al., 2024a) | ICML 2024 |
| 36. Preference Diffusion for Recommendation (Liu et al., 2024c) | ICLR 2025 |
| 37. Augmenting Sequential Recommendation with Balanced Relevance and Diversity (Dang et al., 2025) | AAAI 2025 |
| 38. Why Does Dropping Edges Usually Outperform Adding Edges in Graph Contrastive Learning? (Xu et al., 2025) | AAAI 2025 |
| 39. Simple and Asymmetric Graph Contrastive Learning without Augmentations (Xiao et al., 2023) | NeurIPS 2023 |
| 40. Hierarchical Topology Isomorphism Expertise Embedded Graph Contrastive Learning (Li et al., 2024c) | AAAI 2024 |
| 41. Simple and Asymmetric Graph Contrastive Learning without Augmentations (Xiao et al., 2023) | NeurIPS 2023 |
| 42. HomoGCL: Rethinking Homophily in Graph Contrastive Learning (Li et al., 2023) | KDD 2023 |
| 43. SmoothGNN: Smoothing-aware GNN for Unsupervised Node Anomaly Detection (Dong et al., 2024) | WWW 2025 |
| 44. Representation Alignment for Generation: Training Diffusion Transformers Is Easier Than You Think (Yu et al., 2024) | ICLR 2025 |
| 45. LoRAMoE: Revolutionizing Mixture of Experts for Maintaining World Knowledge in Language Model Alignment (Dou et al., 2023) | ACL 2024 |
| 46. Reconstruction vs. Generation: Taming Optimization Dilemma in Latent Diffusion Models (Yao et al., 2025) | CVPR 2025 |
| 47. Patch-wise Structural Loss for Time Series Forecasting! (Kudrat et al., 2025) | ICML 2025 |
| 48. Parameter-Efficient Fine-Tuning with Discrete Fourier Transform (Gao et al., 2024) | ICML 2024 |
| 49. Attractor Memory for Long-Term Time Series Forecasting: A Chaos Perspective (Hu et al., 2024) | NeurIPS 2024 |
| 50. FITS: Modeling Time Series with 10k Parameters (Xu et al., 2023) | ICLR 2024 |
| 51. Learning Versatile Skills with Curriculum Masking (Tang et al., 2024) | NeurIPS 2024 |
| 52. Federated Transformer: Multi-Party Vertical Federated Learning on Practical Fuzzily Linked Data (Wu et al., 2024b) | NeurIPS 2024 |

Table 13: Paper list part (b)

| Paper Title | Venue |
| --- | --- |
| 53. In-context Time Series Predictor (Lu et al., 2024) | ICLR 2025 |
| 54. Retaining Key Information under High Compression Ratios: Query-Guided Compressor for LLMs (Cao et al., 2024) | ACL 2024 |
| 55. Estimating Conditional Mutual Information for Dynamic Feature Selection (Gadgil et al., 2023) | ICLR 2024 |
| 56. Deep Temporal Graph Clustering (Liu et al., 2023b) | ICLR 2024 |
| 57. DigiRL: Training In-The-Wild Device-Control Agents with Autonomous Reinforcement Learning (Bai et al., 2024) | NeurIPS 2024 |
| 58. Unsupervised Multimodal Clustering for Semantics Discovery in Multimodal Utterances (Zhang et al., 2024b) | ACL 2024 |
| 59. Occlusion-aware Non-Rigid Point Cloud Registration via Unsupervised Neural Deformation Correntropy (Zhao et al., 2025) | ICLR 2025 |
| 60. Mixed-Type Tabular Data Synthesis with Score-based Diffusion in Latent Space (Zhang et al., 2023a) | ICLR 2024 |
| 61. ReMasker: Imputing Tabular Data with Masked Autoencoding (Du et al., 2023) | ICLR 2024 |
| 62. TabWak: A Watermark for Tabular Diffusion Models (Zhu et al., 2025a) | ICLR 2025 |
| 63. TAROT: Targeted Data Selection via Optimal Transport (Feng et al., 2024) | CVPR 2024 |
| 64. Learning Divergence Fields for Shift-Robust Graph Representations (Wu et al., 2024a) | ICML 2024 |
| 65. Supercharging Graph Transformers with Advective Diffusion Wu et al. (2025a) | ICML 2025 |
| 66. DiffPuter: Empowering Diffusion Models for Missing Data Imputation (Zhang et al., 2024c) | ICLR 2025 |
| 67. Fair Set Cover (Dehghankar et al., 2025) | KDD 2025 |
| 68. The Road Less Scheduled (Defazio et al., 2024) | Neurips 2024 |
| 69. SLAB: Efficient Transformers with Simplified Linear Attention and Progressive Re-parameterized Batch Normalization (Guo et al., 2024) | ICML 2024 |
| 70. Turning up the Heat: MIN-p Sampling for Creative and Coherent LLM Outputs (Nguyen et al., 2024) | ICLR 2025 |
| 71. Data Unlearning in Diffusion Models (Alberti et al., 2025) | ICLR 2025 |
| 72. Your ViT is Secretly an Image Segmentation Model (Kerssies et al., 2025) | CVPR 2025 |
| 73. TABDIFF: A Mixed-Type Diffusion Model for Tabular Data Generation (Shi et al., 2024) | ICLR 2025 |
| 74. Bipartite Graph Convolutional Hashing for Effective and Efficient Top-N Search in Hamming Space (Chen et al., 2023b) | WWW 2023 |
| 75. Semi-supervised Node Importance Estimation with Informative Distribution Modeling for Uncertainty Regularization (Chen et al., 2025a) | WWW 2025 |
| 76. Towards Effective Top-N Hamming Search via Bipartite Graph Contrastive Hashing (Chen et al., 2024b) | TKDE 2024 |
| 77. Hyperbolic Image-Text Representations (Desai et al., 2023) | ICML 2023 |
| 78. Modeling Dense Multimodal Interactions Between Biological Pathways and Histology for Survival Prediction (Jaume et al., 2024) | CVPR 2024 |

Table 14: Paper list part (c)

| Paper Title | Venue |
| --- | --- |
| 79. Multimodal optimal transport-based co-attention transformer with global structure consistency for survival prediction (Xu & Chen, 2023) | ICCV 2023 |
| 80. MoME: Mixture of Multimodal Experts for Cancer Survival Prediction (Xiong et al., 2024) | MICCAI 2024 |
| 81. Prototypical Information Bottlenecking and Disentangling for Multimodal Cancer Survival Prediction (Zhang et al., 2024d) | ICLR 2024 |
| 82. Interpretable Vision-Language Survival Analysis with Ordinal Inductive Bias for Computational Pathology (Liu et al., 2024a) | ICLR 2025 |
| 83. Multimodal Prototyping for Cancer Survival Prediction (Song et al., 2024) | ICML 2024 |
| 84. Robust Multimodal Survival Prediction with the Latent Differentiation Conditional Variational AutoEncoder (Zhou et al., 2025) | CVPR 2025 |
| 85. InfoBatch: Lossless Training Speed Up by Unbiased Dynamic Data Pruning (Qin et al., 2023) | ICLR 2024 |
| 86. UniGAD: Unifying Multi-level Graph Anomaly Detection (Lin et al., 2024b) | NeurIPS 2024 |
| 87. PaGE-Link: Path-based Graph Neural Network Explanation for Heterogeneous Link Prediction (Zhang et al., 2023b) | WWW 2023 |
| 88. Rethinking Fair Graph Neural Networks from Re-balancing (Li et al., 2024e) | KDD 2024 |
| 89. LightGCL: Simple Yet Effective Graph Contrastive Learning for Recommendation (Cai et al., 2023) | ICLR 2023 |
| 90. Enhancing Activity Prediction Models in Drug Discovery with the Ability to Understand Human Language (Seidl et al., 2023) | ICML 2023 |
| 91. What to align in multimodal contrastive learning? (Dufumier et al., 2024) | ICLR 2025 |
| 92. AlphaFuse: Learn ID Embeddings for Sequential Recommendation in Null Space of Language Embeddings (Hu et al., 2025) | SIGIR 2025 |
| 93. Beyond Matryoshka: Revisiting Sparse Coding for Adaptive Representation (Wen et al., 2025) | ICML 2025 |
| 94. EmbedLLM: Learning Compact Representations of Large Language Models (Zhuang et al., 2024) | ICLR 2025 |
| 95. Highly Compressed Tokenizer Can Generate Without Training (Beyer et al., 2025) | ICML 2025 |
| 96. LLM-ESR: Large Language Models Enhancement for Long-tailed Sequential Recommendation (Liu et al., 2024b) | NeurIPS 2024 |
| 97. Offline Multi-Agent Reinforcement Learning with Implicit Global-to-Local Value Regularization (Wang et al., 2023a) | NeurIPS 2023 |
| 98. Enhancing Large Language Models in Coding Through Multi-Perspective Self-Consistency (Huang et al., 2024) | ACL 2024 |
| 99. Context Matters: Enhancing Sequential Recommendation with Context-aware Diffusion-based Contrastive Learning (Cui et al., 2024b) | CIKM 2024 |
| 100. Treatment-Aware Hyperbolic Representation Learning for Causal Effect Estimation with Social Networks (Cui et al., 2024a) | SDM 2024 |
| 101. PointCLIP V2: Prompting CLIP and GPT for Powerful 3D Open-world Learning (Zhu et al., 2023) | ICCV 2023 |
| 102. LLM-Pruner: On the Structural Pruning of Large Language Models (Ma et al., 2023) | NeurIPS 2023 |

Table 15: Paper list part (d)

## L CASE STUDY

In this section, we provide a step-by-step demonstration of how feedback progressively guides model-generated code toward the correct canonical implementation. We begin with the formal paper description (Fig. 6), which specifies the underlying algorithm and serves as the authoritative reference. From this description, a detailed instruction is constructed, translating the theoretical requirements into a precise programming task. The model's initial output, shown as generated code (Fig. 8), captures the general intent but diverges from the canonical code (Fig. 7) in subtle yet important ways, such as in the handling of min_tokens_to_keep. To close this gap, the feedback (Fig. 9) is produced, which diagnoses the deviation, explains its consequences, and prescribes a targeted correction using a scatter-based enforcement strategy. Finally, we present the revised code (Fig. 10), where the integration of feedback yields an implementation that fully aligns with the canonical method.

**Paper Description**

```
1   Formally, at each time step $t$, let $\mathcal{V}$ denote the vocabulary, and $P(x_
        {1:t-1})$ represent the conditional probability distribution over the vocabulary for the
        next token $x_t$. Min-\( p \) sampling works as follows:
2   \begin{enumerate}
3       \item \textbf{Calculate the Maximum Probability:} Identify the maximum probability token
            in the distribution, denoted as $p_{\max} = \max_{v \in \mathcal{V}} P(v \mid x_{1:t
            -1})$.
4       \item \textbf{Define the Truncation Threshold:} Set a base probability threshold, $p_{\
            text{base}} \in (0, 1]$, and scale it by $p_{\max}$ to determine the actual
            truncation threshold:
5       \begin{equation}
6           p_{\text{scaled}} = p_{\text{base}} \times p_{\max}
7       \end{equation}
8       This threshold ensures that tokens with sufficiently high relative probabilities are
            considered while filtering out less probable tokens in a context-dependent manner.
9       \item \textbf{Define the Sampling Pool:} Construct the sampling pool $\mathcal{V}_{\text{
            min}}$ consisting of tokens whose probabilities are greater than or equal to $p_{\
            text{scaled}}$:
10      \begin{equation}
11          \mathcal{V}_{\text{min}} = \{v \in \mathcal{V} : P(v \mid x_{1:t-1}) \geq p_{\text{
                scaled}} \}
12      \end{equation}
13      \item \textbf{Sample from the Pool:} Sample the next token $x_t$ from the reduced set $\
            mathcal{V}_{\text{min}}$ according to their normalized probabilities:
14      \begin{equation}
15          P'(v) = \frac{P(v \mid x_{1:t-1})}{\sum_{v' \in \mathcal{V}_{\text{min}}} P(v' \mid x_
                {1:t-1})} \quad \text{for } v \in \mathcal{V}_{\text{min}}
16      \end{equation}
17  \end{enumerate}
```

**Instruction**

```
1   ### **Instruction 1: Implement Weighted Pooling Utility**
2   * **Functionality Description:** Create a utility function that performs weighted pooling of
        hidden states within fixed-size windows using PyTorch tensor operations. The function
        should aggregate token representations by using a `scatter_add` operation to efficiently
        calculate a weighted sum for each window. It must also generate a new attention mask for
        the resulting pooled sequence, where a pooled position is considered valid if at least
        one token in its original window was valid. This function directly implements the
        summation part of the weighted pooling operation described in Equation 6 of the paper.
3   * **Expected Function Name:** `fix_window_size_pooling`
4   * **Input:**
5       * `hidden_states` (`torch.Tensor`): The input token representations, with a shape of `(
            batch_size, num_windows, window_size, hidden_size)`.
6       * `attention_mask` (`torch.Tensor`): A mask indicating valid tokens in the original, un-
            windowed sequence.
7       * `weights` (`torch.Tensor`): The weights to apply to each token representation before
            summing. Must have the same shape as `hidden_states`.
8   * **Output:**
9       * `pooled_hidden_states` (`torch.Tensor`): The resulting sequence of pooled
            representations, with a shape of `(batch_size, num_windows, hidden_size)`.
10      * `pooled_attention_mask` (`torch.Tensor`): The new attention mask corresponding to the
            pooled sequence.
```

Figure 6: A example of paper description and code generation instruction

**CANONICAL CODE**

```
1   def __call__(self, input_ids: torch.LongTensor, scores: torch.FloatTensor) -> torch.
        FloatTensor:
2       probs = torch.softmax(scores, dim=-1)
3       top_probs, _ = probs.max(dim=-1, keepdim=True)
4       scaled_min_p = self.min_p * top_probs
5       tokens_to_remove = probs < scaled_min_p
6       sorted_indices = torch.argsort(scores, descending=True, dim=-1)
7       # Reorder the `tokens_to_remove` mask to align with the sorted tokens.
8       sorted_indices_to_remove = torch.gather(tokens_to_remove, dim=-1, index=sorted_indices)
9       sorted_indices_to_remove[..., : self.min_tokens_to_keep] = False
10      indices_to_remove = sorted_indices_to_remove.scatter(1, sorted_indices,
            sorted_indices_to_remove)
11      scores_processed = scores.masked_fill(indices_to_remove, self.filter_value)
12      return scores_processed
```

Figure 7: Example of a generated code

**Generated Code**

```
1   def __call__(self, input_ids: torch.LongTensor, scores: torch.FloatTensor) -> torch.
        FloatTensor:
2       if not torch.is_tensor(scores):
3           raise TypeError("scores must be a torch.Tensor")
4       if scores.dim() != 2:
5           raise ValueError(f"scores must be a 2D tensor of shape (batch_size, vocab_size), got
                shape {scores.shape}")
6
7       device = scores.device
8       dtype = scores.dtype
9       batch_size, vocab_size = scores.shape
10      probs = torch.softmax(scores, dim=-1)  # shape (batch_size, vocab_size)
11      p_max, _ = probs.max(dim=-1, keepdim=True)
12      p_scaled = p_max * float(self.min_p)  # shape (batch, 1)
13      tokens_to_remove = probs < p_scaled  # (batch, vocab), boolean
14      k = min(self.min_tokens_to_keep, vocab_size)
15      if k > 0:
16          kept_counts = (~tokens_to_remove).sum(dim=-1)  # (batch,)
17          need_fix = kept_counts < k  # (batch,)
18
19          if need_fix.any():
20              topk_vals, topk_indices = torch.topk(probs, k=k, dim=-1)  # (batch, k)
21              keep_topk_mask = torch.zeros_like(tokens_to_remove, dtype=torch.bool, device=
                    device)
22              keep_topk_mask.scatter_(1, topk_indices, True)  # (batch, vocab)
23
24              need_fix_idx = need_fix.nonzero(as_tuple=False).squeeze(-1)
25              if need_fix_idx.numel() == 1:
26                  idx = need_fix_idx.item()
27                  # Un-remove top-k positions for this sequence
28                  tokens_to_remove[idx] &= ~keep_topk_mask[idx]
29              else:
30                  # If multiple sequences need fixing, use advanced indexing
31                  tokens_to_remove[need_fix_idx] &= ~keep_topk_mask[need_fix_idx]
32
33      processed_scores = scores.clone()
34      if self.filter_value == float("-inf"):
35          fill_val = float("-inf")
36      else:
37          fill_val = float(self.filter_value)
38
39      processed_scores = processed_scores.masked_fill(tokens_to_remove, fill_val)
40
41      return processed_scores
```

Figure 8: Example of a generated code

**Feedback**

```
1   # Feedback Report
2   ## Feedback 1
3   ... ...
4
5   ## Feedback 2
6   **Interface:** __call__
7   ### Description
8   `min_tokens_to_keep` logic deviates from the canonical sort/gather/scatter approach and uses a
          conditional top-k mask, which can lead to discrepancies in which tokens are preserved
9
10  ### Analysis
11  The canonical code enforces the `min_tokens_to_keep` safeguard by sorting the logits,
          gathering the removal mask, unmarking the top-k positions, and scattering back to the
          original order. This approach is guaranteed to mirror the paper's requirement in all tie
          or batch cases. The generated code's manual `topk` + boolean indexing branch may select
          different tokens when probabilities tie or in edge cases, causing mismatches (as seen in
          `test_min_tokens_to_keep`).
12
13  ### Actionable Feedback
14  Replace the existing `min_tokens_to_keep` block with the canonical pattern: sort logits,
          gather the removal mask, reset the first `min_tokens_to_keep` entries, and scatter back
          to the original vocabulary order.
15
16  ### Direct Code Feedback
17  ```text
18  In `__call__`, locate the section starting with the comment `# Enforce min_tokens_to_keep:`
          and replace it entirely with:
19  ```python
20          # ### Implementation Details: ensure at least min_tokens_to_keep tokens are kept
21          sorted_indices = torch.argsort(scores, descending=True, dim=-1)
22          sorted_mask = torch.gather(tokens_to_remove, dim=-1, index=sorted_indices)
23          sorted_mask[..., : self.min_tokens_to_keep] = False
24          tokens_to_remove = sorted_mask.scatter(1, sorted_indices, sorted_mask)
25  ```
26  This aligns the implementation with the canonical scatter-based method.
27  ```
```

Figure 9: Feedback to the generated code.

**Result Code**

```
1   def __call__(self, input_ids: torch.LongTensor, scores: torch.FloatTensor) -> torch.
        FloatTensor:
2
3       if not torch.is_tensor(scores):
4           raise TypeError("scores must be a torch.Tensor")
5       if scores.dim() != 2:
6           raise ValueError(f"scores must be a 2D tensor of shape (batch_size, vocab_size), got
                shape {scores.shape}")
7
8       device = scores.device
9       batch_size, vocab_size = scores.shape
10      probs = torch.softmax(scores, dim=-1)  # shape (batch_size, vocab_size)
11      p_max, _ = probs.max(dim=-1, keepdim=True)  # (batch, 1)
12      p_scaled = p_max * float(self.min_p)  # shape (batch, 1)
13      tokens_to_remove = probs < p_scaled  # (batch, vocab), boolean
14      min_keep = min(self.min_tokens_to_keep, vocab_size)
15      if min_keep > 0:
16          sorted_indices = torch.argsort(scores, descending=True, dim=-1)  # (batch, vocab)
17          sorted_mask = torch.gather(tokens_to_remove, dim=-1, index=sorted_indices)  # (batch,
                vocab)
18          sorted_mask[:, :min_keep] = False
19          new_tokens_to_remove = torch.ones_like(tokens_to_remove, dtype=torch.bool, device=
                device)
20          new_tokens_to_remove.scatter_(1, sorted_indices, sorted_mask)
21          tokens_to_remove = new_tokens_to_remove
22      processed_scores = scores.clone()
23      processed_scores = processed_scores.masked_fill(tokens_to_remove, float(self.filter_value)
            )
24
25      return processed_scores
```

Figure 10: Example of revised code.

# M PROMPTS

## M.1 CODE GENERATION PROMPT

**System Prompt**

```
1   You are an expert machine learning researcher.
2   Task description: Your task is to generate high-quality, well-documented, and contextually
         relevant code implementations based on the provided LaTeX version of a research paper and
          a specific user instruction. You will also have access to the existing code repository
         to ensure your generated code integrates seamlessly.
3   Commands actions:
4   ============= CODE READ TOOL =============
5   You also have access to a code reading tool.
6   This tool allows you to read the content of any file in the current repository. It helps you
         understand the codebase context before making changes or generating new code.
7   You can read a file using the following command:
8   READ <file path to be read>
9   READ is the word READ, and <file path to be read> is the relative path of the file you want to
          inspect. This command will return the full content of the specified file.
10  Use this command when you need to examine the contents of any file in the repository,
         including the current target file.
11
12  ============= CODE RETRIEVE TOOL =============
13  You also have access to a code retrieval tool.
14  This tool allows you to retrieve the most relevant code functions for a given natural language
          query. It helps you understand where and how specific functionality is implemented in
          the codebase.
15  You can retrieve code using the following command:
16  RETRIEVE
17  QUERY
18  RETRIEVE is the word RETRIEVE, and QUERY is your request in natural language. The response
         will return the top relevant functions, including their code and location in the
         repository. This will be the primary way to locate and explore code related to specific
         functionality or concepts.
19  Use this command when you want to investigate or modify code related to a particular feature,
         action, or behavior.
20
21  ============= REWRITE CODE EDITING TOOL =============
22  You also have access to a code replacing tool.
23  This tool allows you to entirely re-write/replace all of the current code and erase all
         existing code.
24  You can use this tool via the following command: ```REPLACE
25  <code here>
26  ```, where REPLACE is the word REPLACE and <code here> will be the new code that is replacing
         the entire set of old code. This tool is useful if you want to make very significant
         changes, such as entirely changing the code file content. Try limiting the use of
         rewriting and aim for editing the code more.
27  ============= CODE SUBMIT TOOL =============
28  You also have access to a code submission tool.
29  This tool executes the current code in the target file and returns the results of its
         execution, including any unit test outcomes and feedback related to the code's behavior.
30  You can submit code using the following command:
31  SUBMIT
32  SUBMIT is the word SUBMIT, and it will run the code currently present in the target file.
         After execution, you will receive the results of any unit tests, along with diagnostic
         messages or errors that occurred during runtime.
33
34  ============= REPO BROWSE TOOL =============
35  You also have access to a repository browsing tool.
36  This tool allows you to browse the entire code repository associated with the current task. It
          helps you understand the overall structure, locate files, and explore code across
         different modules or components.
37  You can browse the repository using the following command:
38  BROWSE
39  BROWSE is the word BROWSE, and it will return a list of files and directories in the
         repository. This will be useful for understanding how different parts of the codebase are
          organized and where specific functionality is implemented.
40
41  You should use these actions to access the code file and reterieve the code repository
         information. Before the actions you should relect about the context and make sure the
         command is following the correct syntex.
42  You should follow this format:
43  reflect: [Your reflection on the context and the action you are going to take]
44  action:
45  [The action you are going to take]
46  Your reflect should cover these aspects:
47
48  1. Execution Results
```

```
49
50  - Diagnose compilation or runtime errors (e.g., syntax errors, missing dependencies).
51
52  - Inspect test case outcomes, focusing on which tests failed and the corresponding error
        messages.
53
54  - Consider performance-related signals if available (e.g., timeouts, memory overuse).
55
56  2 Code Consistency
57
58  - Check alignment between the generated code and the method description in the paper.
59
60  - Ensure compatibility with the existing repository (function signatures, class structures,
        module dependencies).
61
62  - Maintain coherent structure and style consistent with the project.
63
64  3. Feedback Integration
65
66  - Extract actionable guidance from human (or simulated) feedback.
67
68  - Identify logical flaws, missing components, or suggested improvements.
69
70  - Translate natural-language feedback into concrete modification strategies.
71
72  4. History Awareness
73
74  - Review previous attempts to avoid repeating failed solutions.
75
76  - Identify patterns in past mistakes and refine strategy accordingly.
77
78  5. Next-Step Planning
79
80  - Identify the current code status.
81
82  - Decide on the priority of actions (e.g., reading files, searching the repository, or editing
        code).
83
84  - Determine the scope of changes (minor patch vs. major refactor).
85
86  - Identify additional information needs before generation.
```

## User prompt

```
1   '''## Research Code Generation Request
2   ---
3   **1. Relevant LaTeX Content:**
4   ---
5   Below you will find the necessary information to generate the requested code. Please process
        all sections carefully.
6   {latex code}
7
8   ---
9   2. Code Generation Instruction:
10  ---
11  {code generation instruction}
12
13  ---
14  3. Conversation History
15  ---
16  {conversation history}
17
18  ---
19  4. Current Code implementation
20  ---
21  {current code content}
22
23  ---
24  5. Feedback on Previous Submition
25  ---
26  {feedback content}
27
28  ---
29  6. Action Execution Result
30  ---
31  {action execution result}
32
33  '''
```

## M.2 FEEDBACK GENERATION PROMPT

### System Prompt

```
1  You are an expert Code Analysis Agent. Your task is to generate detailed and actionable
      feedback on a piece of generated code. This feedback will be used to improve future code
      generation attempts.
2
3  You will be provided with the following information to perform your analysis:
4
5  1.  **LaTeX Code from Research Paper:** This document (or snippet) describes the intended
       mathematical or algorithmic functionality that the target code should implement. Use this
       to understand the core logic, equations, and theoretical underpinnings.
6  2.  **User Instruction/Prompt:** This is the original instruction given to the code generation
       model that produced the 'Generated Code'. Evaluate if the generated code aligned with
       this instruction.
7  3.  **Canonical Code with Comments:** This is a reference or ideal implementation of the
       desired functionality. It contains specific comments highlighting key aspects, logic
       flows, or potential pitfalls. Use this as a reference for feedback.
8  4.  **Generated Code:** This is the code produced by another LLM that you need to analyze.
9  5.  **Error Information:** This is the execution error information of the generated
       information.
10  6.  **Generation Guidance & Feedback Specification:** This document outlines:
11      * **Generation Guidance:** The generation format for this task.
12      * **Feedback Categories:** The predefined categories your feedback should address.
13      * **Feedback Granularity:** The level of detail required for your feedback.
14
15  Based on the provided information, you must generate your feedback based on the user
      specification.
16
17  # **Your Task:**
18
19  Carefully compare the **LLM-Generated Code** and the **Canonical Code** for the specified **
      interface, function, or method** as instructed in the **Code Generation Instruction**.
20
21  Your tasks are as follows:
22
23  1. **Identify all differences:**
24      Explicitly list every relevant difference between the **LLM-Generated Code** and the **
         Canonical Code** within the instructed interface, function, or method. For each
         difference, clearly specify what is present in one version and absent or different in
         the other.
25
26  2. **Analyze each difference:**
27    Clearly explain the impact of the difference and why it lead to an error in the context of
         the instruction and the canonical solution (expected code).
28
29  2. **Analyze each difference:**
30      For each difference, analyze its significance and determine whether it affects correctness
         , completeness, style, or performance. Select the most appropriate feedback category
         (T0--T4) for each difference. Clearly explain the impact of the difference and why it
         lead to an error in the context of the instruction and the canonical solution (
         expected code).
31
32
33  4. **Direct code-level feedback:**
34      For each actionable item, provide a detailed description of exactly how to modify the LLM-
         Generated Code to resolve the difference. The code should be identical with the code
         snippet .
35
36  **Output Format:**
37  Produce a JSON object with a `"differences"` array. For each difference, include the following
          fields:
38  - `"interface"`: The name of the interface, function, or method where the difference occurs
39  - `"category"`: The feedback category (T0-T4)
40  - `"description"`: A brief description of the difference
41  - `"analysis"`: How current implementation lead to an error, why it is not correct
42  - `"actionable_feedback"`: Clear, concrete, and actionable guidance for correction
43  - `"direct_code_feedback"`: Consistent with the actionable feedback, a detailed description of
          how to modify the code to resolve the difference. Use the canonical code snippet as the
         guidance feedback.
44
45  **Example Output:**
46  {{
47    "differences": [
48      {{
49        "interface": "calculate_total",
50        "category": "T1",
51        "description": "The LLM-Generated Code does not initialize the variable 'var' before use
             .",
```

```
52        "analysis": "The generated code omits the initialization of the variable 'var', which
               can cause a runtime error or incorrect results.",
53        "actionable_feedback": "Ensure that all variables are properly initialized before they
               are used.",
54        "direct_code_feedback": "Add the line `var = []` before the iteration to initialize the
               variable as shown in the Canonical Code."
55      }}
56      // Add more differences as needed
57    ]
58  }}
59  Instructions:
60
61  * Focus on all differences in the instructed code region, not just those leading to major
         errors.
62
63  * Differences can include changes in logic, missing or additional statements, variable
         initialization, return values, control flow, structure, function signatures, etc.
64
65  * Be explicit and systematic for each observed difference.
```

## User Prompt

```
1   **Task Information:**
2
3   Please generate code analysis feedback based on the following information:
4
5   1.  **Paper Description (LaTeX):**
6       ```latex
7       {latex code}
8       ```
9
10  2.  **Code Generation Instruction:**
11      ```
12      {code generation instruction}
13      ```
14
15  3.  **Canonical Code (Ground Truth/Example):**
16      ```{}
17      {canoncial code}
18      ```
19
20  4.  **LLM-Generated Code:**
21      ```{}
22      {generated code content}
23      ```
24  5.  **Error Information:**
25      ```{execution error log}
26      ```
27
28  6.  **Generation Guidance & Feedback Specification:**
29      **Subject: Understanding the 5 Feedback Categories for Code Generation**
30  To ensure we evaluate LLM-generated code consistently, we use a structured feedback system.
         This system helps us pinpoint the exact nature of an error.
31
32  Our system is based on two simple questions:
33  1.  **What kind of error is it?** (This is the **Category**, $T_0-T_4$)
34  2.  **How much help do we provide?** (This is the **Level**, $L_0-L_4$)
35
36  This document introduces the five **Categories** of feedback. When analyzing a piece of code,
         your first step is to identify which of these five categories the *most significant*
         error falls into.
37
38  ---
39
40  ### The 5 Feedback Categories ($T_0 - T_4$)
41
42  Here is a guide to each category, designed to help you quickly classify any error you
         encounter.
43
44  #### **$T_0$: Code Structure (Planning) Feedback**
45  * **In a Nutshell:** The overall architectural plan is wrong.
46  * **Core Question:** Is the code's high-level organization or structure fundamentally
         different from the intended design, even if some of the internal logic is correct?
47  * **Look for:**
48      * A single, monolithic function when it should have been broken down into smaller helper
             methods.
49      * A class that is missing essential methods required for its core purpose.
50      * Incorrect data flow or a flawed high-level implementation strategy.
51
52  #### **$T_1$: Code Correctness Feedback**
```

```
53  * **In a Nutshell:** The code is fundamentally broken and won't run.
54  * **Core Question:** Does the code fail due to a basic syntax mistake, a typo, or a
        fundamental Python error?
55  * **Look for:**
56      * `SyntaxError`: e.g., an unclosed parenthesis or unterminated string.
57      * `NameError`: e.g., using a variable before it has been assigned.
58      * `TypeError`: e.g., trying to add a string to an integer (when not part of the core
            algorithm's logic).
59
60  #### **$T_2$: Implementation Alignment Feedback**
61  * **In a Nutshell:** The code ignores or misinterprets the provided paper or instructions.
62  * **Core Question:** Can I point to a specific sentence, formula, or requirement in the
        provided text that this code directly contradicts?
63  * **Look for:**
64      * Implementing Equation (4) from the paper when Equation (3) was specified.
65      * Using the wrong parameter names in a function call compared to the instructions.
66      * Producing an output with the wrong shape or data type described in the paper.
67
68  #### **$T_3$: Knowledge & Context Feedback**
69  * **In a Nutshell:** The code fails because it's missing crucial knowledge that was **not**
        provided.
70  * **Core Question:** Is the fix something the LLM would need to know from general domain
        expertise, or is it an implicit "secret" of the original codebase that wasn't written
        down?
71  * **Look for:**
72      * Using placeholder logic for a complex but standard operation (e.g., "TODO: implement
            topology calculation here").
73      * Needing a specific, unstated hyperparameter (e.g., a learning rate of `0.001`).
74      * Using a generic library when a specific, domain-standard library (e.g., `gudhi`, `
            huggingface`) is the obvious choice.
75
76  #### **$T_4$: Repository Integration Feedback**
77  * **In a Nutshell:** The code reinvents the wheel or fails to use existing tools from the
        codebase.
78  * **Core Question:** Did the code rewrite a helper function or class that was already
        available in the project's existing files?
79  * **Look for:**
80      * Writing a new `normalize_text` function when `utils.text.normalize()` already exists.
81      * Incorrectly using a provided class from another module in the repository.
82      * Ignoring the established coding style or conventions of the repository.
83
84  ---
85
86  ### How to Choose the Right Category: A Decision Guide
87
88  Always start at the top of this list. The first "Yes" determines the error's category. This
        helps distinguish between similar issues (especially $T_2$ and $T_3$).
89
90  1.  **Is it a basic syntax/runtime error?**
91      * **Yes?** -> It's **$T_1$**.
92
93  2.  **No? Okay, does it ignore or re-implement existing code from the repo?**
94      * **Yes?** -> It's **$T_4$**.
95
96  3.  **No? Is the overall code architecture or structure the main problem?**
97      * **Yes?** -> It's **$T_0$**.
98
99  4.  **No? Can I find the fix *explicitly written* in the paper/instructions?**
100     * **Yes?** -> It's **$T_2$**. (The LLM failed to read carefully).
101
102 5.  **No? Is the fix based on knowledge *outside* the paper/instructions?**
103     * **Yes?** -> It's **$T_3$**. (The LLM lacked necessary context).
104
105 Please generate the feedback with the specified format.
```

## M.3 CATEGORY CLASSIFICATION PROMPT

### System Prompt

```
1  You are an expert code reviewer and research engineer.
2  Your role:
3  - Classify feedback about code into error categories(T0-T4).
4  - Judge whether the next version of the code (v2) adopted the feedback and whether the issue
        was resolved.
5  - Follow definitions and disambiguation strictly.
6  - Provide concise evidence (<=3 items) with location references.
7  - If unsure, use `"uncertain"`.
8
```

```
 9  ## Input Specification
10  The user prompt will provide the following blocks of content (each delimited clearly):
11
12  1. **[PAPER]** - Excerpts from the research paper (may include formulas, equations, section
        refs).
13  2. **[INSTRUCTION]** - The instruction given to generate code based on the paper.
14  3. **[GROUND_TRUTH_CODE]**(optional) - A reference or canonical code implementation.
15  4. **[V1_CODE]** - The first generated code version.
16  5. **[V1_RUN_LOG]** - Execution results and error logs for v1.
17  6. **[FEEDBACK]** - Feedback on v1 (may contain multiple feedback entries).
18  7. **[V2_CODE]** - The new generated code (after applying feedback).
19  8. **[V2_RUN_LOG]** - Execution results and error logs for v2.
20
21  ## Error Types
22  - **T0: Code Structure (Planning)** - The error is because the high-level design, architecture
        , data flow, modularization, or strategy is not aligned with the canonical or intended
        design.
23  - **T1: Code Correctness** - The error is because of general syntax, runtime, or basic
        programming logic errors.
24  - **T2: Implementation Alignment** - The error is because the code is implemented with
        misalignment to the paper algorithm, formulas, or instruction I/O requirements.
25  - **T3: Knowledge & Context** - The error is because of missing domain knowledge, conventions,
        implicit assumptions, or author preference.
26  - **T4: Repository Integration** - The error is because of not using the same function defined
        within this repo (other file), misuse of existing codebase (from other file), or
        reimplementing helpers.
27
28  ## Disambiguation Guidelines
29  - **T1 vs T2**:
30    If the issue would arise in any code regardless of the paper -> T1.
31    If it stems from not following the paper's algorithm or I/O specs -> T2.
32
33  - **T0 vs T2**:
34    T0 = structural/architectural guidance, modularization.
35    T2 = algorithmic/methodological alignment with the paper.
36
37  - **T4 vs T0**:
38    T4 = repository-specific (reuse of helpers, placement, conventions from other file).
39    T0 = general design/architecture not tied to repo assets.
40
41  - **T3 vs Others (Key Rule)**:
42    - Assign **T3** when the feedback relies on *external or implicit knowledge, or author
        preference/insights in code* not fully contained in the instruction, repo, or paper.
43    - Examples: domain-standard library usage, interpreting ambiguous terms, applying author
        preference/insights or common domain practices, filling in paper omissions.
44    - If the feedback can be resolved solely by:
45      - fixing syntax/runtime -> T1,
46      - aligning with explicit paper details or instruction specification -> T2,
47      - restructuring modules/classes -> T0,
48      - reusing repo assets -> T4.
49    - Otherwise, if it depends on **domain expertise or implicit assumptions**, classify as T3.
50
51  ## Evaluation Pipeline Specification
52  When analyzing inputs, follow these ordered steps:
53
54  1. **Parse Input**
55     - Collect paper excerpts, code generation instruction, ground truth (if any), v1 code &
          logs, feedback text, and v2 code & logs.
56     - Split multi-point feedback into atomic items.
57
58  2. **Classify Error Type (T0-T4)**
59     Use the following decision rules:
60     - **T0 (Structure/Planning)**: Feedback is about *how code is organized or architected*, e.
          g., "this logic should be a class method," "split into functions," "refactor data flow
          ."
61     - **T1 (Correctness)**: Feedback is about *generic programming bugs* like syntax errors,
          type mismatches, variable misuse, bad loop conditions, out-of-bounds. No relation to
          paper content.
62     - **T2 (Implementation Alignment)**: Feedback is about *not matching the paper or task spec
          *, e.g., wrong formula, wrong output shape vs. paper, incorrect algorithm step.
63     - **T3 (Knowledge & Context)**: Feedback requires *domain knowledge, implicit assumptions,
          or conventions* to understand, e.g., "use library.function() for NLP task," "by '
          attention' they mean Scaled Dot-Product," "paper omits activation X."
64     - **T4 (Repository Integration)**: Feedback is about *using or misusing existing repo code
          *, e.g., "use utils.DataLoader instead of reimplementing," "place logic in Model.
          forward," "use config object."
65
66  3. **Judge Adoption (adopted)**
67     - Compare v1 and v2.
68     - If v2 reflects meaningful changes related to the feedback -> `YES`.
69     - If no change -> `NO`.
```

```
70        - If partially implemented -> `PARTIAL`.
71
72   4. **Judge Resolution (resolved)**
73        - Check v2 logs, outputs, and alignment with paper/instructions.
74        - If the problem is fully fixed -> `YES`.
75        - If still present -> `NO`.
76        - If partially improved -> `PARTIAL`.
77
78   5. **Explain Decision**
79        - Write a concise explanation (1-3 sentences) citing evidence (e.g., code diff, log line,
               paper reference).
80
81   6. **Confidence Score**
82        - Assign a float 0-1 reflecting certainty in classification and judgments.
83        - Higher = more certain, lower = less certain.
84
85   7. **Output**
86        - Return results as the specified output format
87        error_type: ErrorType = Field(..., description="T0-T4")
88        adopted: AdoptStatus = Field(..., description="YES/NO/PARTIAL")
89        resolved: AdoptStatus = Field(..., description="YES/NO/PARTIAL")
90        explain_error_type: str = Field(..., min_length=1, max_length=500, description="Describe
               the reason the error is categorized to this error type.")
91        explain_adopted_solved: str = Field(..., min_length=1, max_length=500, description="
               Describe the reason the feedback is adopted or not and the error is resolved")
92        confidence_score: confloat(ge=0.0, le=1.0) = Field(..., description="0-1")
93
94
95
96   ## Important Notes:
97   - **The canonical code meaning in feedback** mentioned in the feedback is the expcted
           generated code(is the GROUND_TRUTH_CODE in the user prompt) for the task, but **is not
           the repo code**. You need to analysis the root cause of the error and catecory to the
           correct category.
98   - **Output content**:
99        If the error is in category 1, the explain_error_type should explain what syntex error or
               runtime error os.
100       If the error is in category 2, the explain_error_type should explain how the **instruction
               or paper** is specified and how the code is misaligned with the paper description or
               specification.
101       If the error is in category 3, the explain_error_type should explain how the code is
               implemented and what information is not specified in the **paper or instruction**.
102       If the error is in category 4, the explain_error_type should explain what and how the
               function that defined in the repository is misused or not used as it expected.
103  '''
```

## User Prompt

```
1    You are given the following data.
2    Please analyze and output structured JSON according to the schema above.
3
4
5    [PAPER]
6    {paper content}
7
8    [INSTRUCTION]
9    {code generation instruction}
10
11   [GROUND_TRUTH_CODE]
12   {ground truth(optional)}
13
14   [V1_CODE]
15   {generated code v1}
16
17   [V1_RUN_LOG]
18   {v1 code execution log}
19
20   [FEEDBACK]
21   {feedback contents}
22
23   [V2_CODE]
24   {revised code v2}
25
26   [V2_RUN_LOG]
27   {v2 code execution log}
28
29   ## Important Notes:
30   - **The canonical code meaning in feedback** mentioned in the feedback is the expcted
           generated code(is the GROUND_TRUTH_CODE in the user prompt) for the task, but not some
```

```
code exisits in the repo. You need to analysis the root cause of the error and catecory
    to the correct category.
```

# N ANNOTATION EXAMPLE

## N.1 PAPER DESCRIPTION

```
1  \section{Problem formulations}
2  \label{sec:background}
3  \subsection{Setting}
4  We focus on statistical inference problems for the parameter \(\theta^{*} \equiv \theta^{*}(\
       mathbb{P}) \in \RR^K\) defined on the joint distribution of \((X, Y) \sim \mathbb{P}\),
       where \(Y \in \mathcal{Y}\) is a scalar outcome and \(X \in \mathcal{X}\) be a \(K\)-
       dimensional vector representing features. We are interested in estimating \(\theta^{*}\)
       using labeled data \(\calL = \{(\X_i, Y_i), i = 1,\cdots,n\} \equiv (\XL, \YL)\),
       unlabeled data \ \(\calU = \{\X_i, i = n+1,\cdots,n+N\} \equiv \XU\), and a pre-trained
       ML model \(\wh f(\cdot): \mathcal{X} \rightarrow \mathcal{Y}\). Here, \(f(\cdot)\) is a
       black-box function with unknown operating characteristics and can be mis-specified. We
       also require an algorithm \(\cA\) that inputs the labeled data \(\calL\) and returns a
       consistent and asymptotically normally distributed estimator \(\widehat{\theta}\) for \(\
       theta^*\). There are three common ways in the literature to estimate \(\theta^*\):
5  \begin{itemize}
6      \item \textbf{Classical statistical methods} apply algorithm \(\cA\) to only labeled data
           \(\calL = (\XL, \YL)\), and returns the estimator and its estimated variance \([\wh\
           bt_\calL, \wh\myVar(\wh\bt_\calL)]\). Valid confidence intervals and hypothesis tests
            can then be constructed using the asymptotic distribution of the estimator. However,
            it ignores the unlabeled data and ML prediction.
7
8      \item \textbf{Imputation-based methods} treat ML prediction \(\wh f\) in the unlabeled
           data as the observed outcome, and apply algorithm \(\cA\) to \(\calU = (\XU, \fU)\).
           We denote the estimator and estimated variance as \([\wh\bmeta_\calU, \wh\myVar(\wh\
           bmeta_\calU)]\). This has been shown to give invalid inference results and false
           scientific findings \citep{wang2020methods,angelopoulos2023prediction,
           miao2023assumption,miao2024valid}.
9      \item \textbf{ML-assisted inference methods} use both \(\calL\) and \(\calU\) as input.
           These approaches add a debiasing term in the loss function (or estimating equation)
           for M-estimation problems, thus removing the bias from the imputation-based
           estimators and producing results that are statistically valid and universally more
           powerful compared to classical methods \citep{angelopoulos2023ppi++,
           miao2023assumption,miao2024valid}.
10 \end{itemize}
11 Next, we use an example to provide intuition on ML-assisted inference and our protocol.
12
13 \subsection{Building the intuition with mean estimation}
14 We consider the mean estimation problem, where \(\theta^* = \E[Y_i] \equiv \argmin_\theta\E[\
       frac{1}{2}\left(Y_i-\theta\right)^2]\). The classical method only takes the labeled data
        \(\YL\) as input and yields an unbiased and consistent estimator for \(\theta^*\):
15 \(
16     \wh\theta_\calL = \argmin_\theta  \meann \frac{1}{2}\left(Y_i-\theta\right)^2 = \meann Y_i
17 \). The imputation-based method only takes the unlabeled data \(\fU\) as input and returns
18 \(
19     \wh\eta_\calU = \argmin_\theta  \frac{1}{N} \sum_{i=n+1}^{n+N} \frac{1}{2}(\wh f_i-\theta)
           ^2 = \frac{1}{N} \sum_{i=n+1}^{n+N} \wh f_i
20 \). It is a biased and inconsistent estimator for \(\E[Y_i]\) if the ML model \(\wh f\) is mis
       -specified. To address this, ML-assisted estimator takes both labeled data \((\YL, \fL)\)
        and unlabeled data \(\fU\) as input and adds a debiasing term to the loss function to
       rectify the bias caused by ML imputation \citep{angelopoulos2023prediction,
       miao2023assumption,miao2024valid,gan2023prediction}:
21 \begin{small}
22 \begin{align*}
23     \wh\theta_{\texttt{MLA}}
24     &=
25     \argmin_\theta \frac{1}{2}
26     \{
27     \wh\omega_0\meanN (\wh f_i-\theta)^2
28     -
29     \underbrace{[
30     \wh\omega_0\meann (\wh f_i-\theta)^2
31     -
32     \meann \left(Y_i-\theta\right)^2
33     ]}_{\text{Debiasing term}}
34     \}\\
35     &=
36     \wh\omega_0\meanN\wh f_i -
37      \underbrace{
38     [\wh\omega_0\meann \wh f_i -\meann y_i]
39     }_{\text{Debiasing term}},
```

```
40  \end{align*}
41  \end{small}
42  where the modified loss ensures the consistency of the ML-assisted estimator and  the weight
        \(\wh\omega_0 = \frac{\wh\covl[Y, \wh{f}]/n}{\wh\varl[\wh{f}]/n + \wh\varu[\wh{f}]/N}\)
        ensures that ML-assisted estimator is no less efficient than the classical estimator with
        arbitrary ML predictions:
43  \(
44  \myVar(\wh\theta_{\texttt{MLA}})
45  =
46  \myVar(\wh\theta_\calL)
47  -
48  \frac{\operatorname{Cov}[Y, \widehat{f}]}{n \operatorname{Var}[\widehat{f}]+n^2 \operatorname{
        Var}[\widehat{f}] / N}
49  \leq
50  \myVar(\wh\theta_\calL) \).
51
52  Our proposed method is motivated by the observation that the \textbf{sufficient statistics} of
        the ML-assisted estimator \(\wh\theta_{\texttt{MLA}}\) and its estimated variance \(\wh\
        myVar(\wh\theta_{\texttt{MLA}})\) are the following summary statistics:
53  \[
54      \wh \bt_{\texttt{ss}}
55      =
56      (\meann y_i, \meann \wh f_i, \meanN \wh f_i)
57      \text{ and }
58      \wh\myVar(\wh \bt_{\texttt{ss}})
59      =
60      \begin{bmatrix}
61      \wh\varl[Y]/n & \wh\covl[Y, \wh{f}]/n & 0\\
62      \wh\covl[Y, \wh{f}]/n & \wh\varl[\wh{f}]/n & 0 \\
63      0 & 0 & \wh\varu[\wh{f}]/N \\
64      \end{bmatrix}
65  \]
66
67  Moreover, they can be easily obtained by applying the \textbf{same algorithm} \(\cA\) (mean
        estimation) to
68  \begin{itemize}[itemsep=1pt, parsep=0pt]
69      \item labeled data with observed outcome \(
70      \cA(\YL)
71      \rightarrow [\wh\theta_{\calL}, \wh\myVar(\wh\theta_{\calL})]
72      =
73      [\meann y_i, \wh\varl[Y]/n]
74      \)
75      \item labeled data with predicted outcome \(
76      \cA(\fL)
77      \rightarrow
78      [\wh\eta_{\calL}, \wh\myVar(\wh\eta_{\calL})] =
79      [\meann \wh f_i, \wh\varl[\wh{f}]/n]
80      \)
81      \item unlabeled data with predicted outcome\(\cA(\fU)  \rightarrow  [\wh\eta_{\calU},\wh\
            myVar(\wh\eta_{\calU})]
82      =
83      [\meanN \wh f_i, \wh\varu[\wh{f}]/N]
84      \)
85      \item bootstrap of labeled data \(\cA[(\YL, \fL)_q, q = 1, \dots, Q]\) for estimation of
            \(\wh\myCov(\wh\theta_{\calL}, \wh\eta_{\calL}) = \wh\covl[Y, \wh{f}]/n)\). Here, \((\
            YL, \fL)_q\) represents the \(q\)-th bootstrap of labeled data.
86  \end{itemize}
87
88  Combining these summary statistics for one-step debiasing \(\wh\omega_0\wh\eta_{\calU} - (\wh\
        omega_0\wh\eta_{\calL} - \wh\theta_{\calL})\) recovers \(\wh\theta_{\texttt{MLA}}\).
89
90  To summarize, an algorithm for mean estimation, coupled with resampling, is sufficient for ML-
        assisted mean estimation. This observation inspired us to generalize this protocol for a
        broad range of tasks. Our protocol illustrated in \cref{fig:workflow} only requires three
         steps: 1) using a pre-trained ML model to predict outcomes for labeled and unlabeled
        data, 2) applying existing analysis routines to generate summary statistics, and 3) using
         these statistics in a debiasing procedure to produce statistically valid results in ML-
        assisted inference.
91
92  \begin{figure}[H]
93      \centering
94      \includegraphics[width = 0.85\linewidth]{Figures/PSPS_workflow.pdf}
95      \caption{Workflow of \PSPS~for Task-Agnostic ML-Assisted Inference.}
96      \label{fig:workflow}
97  \end{figure}
98
99
100 \section{Methods}
101 \label{sec:method}
102 \subsection{General protocol for \PSPS}
```

```
103  Building on \cref{sec:background}, we formalized our protocol in \cref{fig:workflow} for ML-
         assisted inference:
104  \begin{algorithm}[H]
105    \caption{\PSPS~for ML-assisted inference}
106    \label{alg:psps}
107    \begin{algorithmic}[1]
108    \Require A pre-trained ML model \(\wh f\), labeled data \(\calL = (\XL, \YL)\), unlabeled
            data \(\calU = \XU\)
109    \State Use the ML model \(\wh f\) to predict the outcome in both labeled and unlabeled data.
110    \State Apply the algorithm \(\cA\) in the analysis routine to
111    \begin{itemize}
112        \item labeled data \((\XL, \YL)\) and obtain \([\wh \bt_\calL, \wh\myVar(\wh \bt_\calL)
                ]\)
113        \item labeled data \((\XL, \fL)\) and obtain \([\wh \bmeta_\calL, \wh\myVar(\wh \bmeta_\
                calL)]\)
114        \item unlabeled data with \((\XU, \fU)\) and obtain \([\wh \bmeta_\calU, \wh\myVar(\wh \
                bmeta_\calU)]\)
115        \item \(Q\) bootstrap of labeled data \((\XL, \YL, \fL)_q, q = 1,\dots,Q\) and obtain
                \(\wh\myCov(\wh \bt_\calL, \wh \bmeta_\calL)\).
116    \end{itemize}
117    \State Employ one-step debiasing to the summary statistics in step2:
118    \begin{center}
119    \(
120        \btpsps = \wh\bomega_0\trans\wh{\bmeta}_{\calU} -  (\wh\bomega_0\trans\wh{\bmeta}_{\calL
                } - \wh{\bt}_{\calL}),
121    \)
122    \end{center}
123    where \(\wh\bomega_0 = [\wh\myVar(\wh \bmeta_\calL) + \wh\myVar(\wh \bmeta_\calU)]^{-1}\wh\
             myCov(\wh \bt_\calL, \wh \bmeta_\calL)\) and \(\wh\myVar(\btpsps) = \wh\myVar(\wh \bt_\
             calL) - \wh\myCov(\wh \bt_\calL, \wh \bmeta_\calL)\trans[ \wh\myVar(\wh \bmeta_\calL) +
             \wh\myVar(\wh \bmeta_\calU)]^{-1}\wh\myCov(\wh \bt_\calL, \wh \bmeta_\calL)\)
124    \Ensure ML-assisted point estimator \(\btpsps\), standard error \(\sqrt{\wh\myVar(\btpsps)
             }\), \(\alpha\)-level confidence interval for the \(k\)-th coordinate \( \mathcal{C}_{\
             alpha,k}^{\PSPS}=(\wh{\bt}_{\PSPS_k} \pm z_{1-\alpha / 2} \sqrt{\wh\myVar(\btpsps)_{kk
             }})\), and (two-sided) p-value \( 2 ( 1 - \Phi(| \frac{\wh{\bt}_{\PSPS_k}}{\sqrt{\wh\
             myVar(\btpsps)_{kk}})} | ) )\), where \(\Phi\) is the CDF of the standard normal
             distribution.
125    \end{algorithmic}
126  \end{algorithm}
127
128  The only requirements for our protocol are: i) algorithm \(\cA\), when applied to labeled data
          \((\bX_\calL, Y_\calL)\), returns a consistent and asymptotically normally distributed
         estimator of \(\bt^*\); ii) labeled and unlabeled data are independent and identically
         distributed. Under these assumptions, the summary statistics have the following
         asymptotic properties:
129  \begin{align}
130  \label{eq:esteq}
131  n^{1 / 2}\left(\begin{array}{c}
132  \wh{\bt}_{\calL} - \bt^* \\
133  \wh{\bmeta}_{\calL} - \bmeta \\
134  \wh{\bmeta}_{\calU} - \bmeta \\
135  \end{array}\right) \coD \mathcal{N}\left\{
136  \left(\begin{array}{c}
137  \0_K \\
138  \0_K \\
139  \0_K
140  \end{array}\right)
141  ,
142  \left(\begin{array}{ccc}
143  \V(\wh \bt_\calL)& \V(\wh \bt_\calL, \wh \bmeta_\calL) & \0\\
144  \V(\wh \bt_\calL, \wh \bmeta_\calL) &  \V(\wh \bmeta_\calL) & \0 \\
145  \0 & \0 &  \rho\V(\wh \bmeta_\calU)
146  \end{array}\right)
147  \right\},
148  \end{align}
149
150  where \(\bmeta \equiv \bmeta(\mathbb{P}_{\wh f}) \in \RR^K\) is defined on \((\X, \wh f) \sim
         \mathbb{P}_{\wh f}\), \(\V(\cdot)\) denotes the asymptotic variance and covariance of a
         estimator, and \(\rho = \frac{n}{N}\). The asymptotic approximation gives
151  \(\V(\wh \bt_\calL) \approx n\myVar(\wh \bt_\calL), \V(\wh \bt_\calL, \wh \bmeta_\calL) \
         approx n\myCov(\wh \bt_\calL, \wh \bmeta_\calL), \V(\wh \bmeta_\calL) \approx n\myVar(\wh
         \bmeta_\calL)\) and \(\V(\wh \bmeta_\calU) \approx N\myVar(\wh \bmeta_\calU)\). Here, we
         do not require \(\wh \bmeta_\calL\) and \(\wh \bmeta_\calU\) to be consistent for \(\bt
         ^*\), thus allows arbitrary ML model.
152
153  With the summary statistics following a multivariate normal distribution asymptotically, the
         debiased estimator \(\btpsps = \wh\bomega_0\trans\wh{\bmeta}_{\calU} -  (\wh\bomega_0\
         trans\wh{\bmeta}_{\calL} - \wh{\bt}_{\calL})\) is consistent for \(\theta^*\) and
         asymptotically normally distributed (\cref{thm:asynor}). Therefore, by plugging in a
         consistent estimator for its asymptotic variance \(\V(\btpsps) \approx n\myVar(\btpsps)\)
         , valid confidence interval and hypothesis testing can be achieved.
```

```
154
155   \begin{remark}
156   \PSPS~is more "task-agnostic" than existing methods in three aspects:
157   \begin{enumerate}
158       \item For M-estimation tasks, currently, only mean and quantile estimation, as well as
                linear, logistic, and Poisson regression, have been implemented in software tools and
                 are ready for immediate application. For other M-estimation tasks, task-specific
                derivation of the ML-assisted loss functions and asymptotic variance via the central
                limit theorem are necessary. After that, researchers still need to develop software
                packages and optimization algorithms to carry out real applications. In contrast, \
                PSPS~only requires already implemented algorithms and software designed for classical
                inference based on labeled data.
159       \item For problems that do not fall under M-estimation but have asymptotically normally
                distributed estimators, only \PSPS~can be applied, and all current methods would fail
                . The principles behind ML-assisted M-estimation do not extend to these tasks.
160       \item Even for M-estimation tasks that have already been implemented, \PSPS offers the
                additional advantage of relying solely on summary statistics. The "task-specific
                derivations" refer not only to statistical tasks but also to scientific tasks. Real-
                world data analysis in any scientific discipline often involves conventions and
                nuisances that require careful consideration. For example, our work is partly
                motivated by GWAS \citep{uffelmann2021genome}. Statistically, GWAS is a linear
                regression that regresses an outcome on many genetic variants. While the regression-
                based statistical foundation is simple, conducting a valid GWAS requires accounting
                for numerous technical issues, such as sample relatedness (i.e., study participants
                may be genetically related) and population structure (i.e., unrelated individuals of
                the same ancestry are both genetically and phenotypically similar, creating
                confounded associations in GWAS). Sophisticated algorithms and software have been
                developed to address these complex issues \citep{mbatchou2021computationally}. It
                will be very challenging if all these important features need to be reimplemented in
                an ML-assisted GWAS framework. With our \PSPS~protocol, researchers can utilize
                existing tools that are highly optimized for genetic applications to perform ML-
                assisted GWAS. This adaptability is not just limited to GWAS, but is a major feature
                of our approach across scientific domains. \PSPS~enables researchers to conduct ML-
                assisted inference using well-established data analysis routines.
161   \end{enumerate}
162   \end{remark}
163
164   \begin{remark}
165       The "federated data analysis" feature of \PSPS~refers to the fact that we only require
                summary statistics as input for inference, rather than individual-level raw data (
                features $\X$ and label $Y$). For example, consider a scenario where labeled data is
                 in one center and unlabeled data is in another, yet researchers cannot access
                individual-level data from both centers simultaneously. Under such conditions,
                current ML-assisted inference, which relies on accessing both labeled and unlabeled
                data to minimize a joint loss function, is not feasible. However, \PSPS~circumvents
                this issue by aggregating summary statistics from multiple centers, thereby
                performing statistical inference while upholding the privacy of individual-level
                data.
166   \end{remark}
167
168   \subsection{Theoretical guarantees}
169   In this section, we examine the theoretical properties of \(\PSPS\). In what follows, \(\coP\)
           denotes convergence in probability and \(\coD\)  denotes convergence in distribution.
           All proofs are deferred to the \cref{app:proof}.
170
171   The first result shows that our proposed estimator is consistent, asymptotically normally
           distributed, and uniformly better in terms of element-wise asymptotic variance compared
           with the classical estimator based on labeled data only.
172   \begin{theorem}
173   \label{thm:asynor}
174   Assuming equation~\eqref{eq:esteq} holds, then \(\btpsps \coP \bt^*\), and
175   \[
176       n^{1 / 2}(\btpsps - \bt^*) \coD
177       \mathcal{N}\left(\0, \V(\btpsps)\right),
178   \]
179   where \(\V(\btpsps) = \V(\wh \bt_\calL) - \V(\wh \bt_\calL, \wh \bmeta_\calL)\trans( \V(\wh \
           bmeta_\calL) +  \rho\V(\wh \bmeta_\calU))^{-1}\V(\wh \bt_\calL, \wh \bmeta_\calL)).
           Assume the \(k\)-th column of \(\V(\wh \bt_\calL, \wh \bmeta_\calL)\) is not a zero
           vector and at least one of \(\V(\wh \bmeta_\calL)\) and \(\V(\wh \bmeta_\calU)\) are
           positive definite, then \(\V(\btpsps)_{kk} \leq \V(\wh \bt_\calL)_{kk}\). With \(\wh\V(\
           btpsps) \coP \V(\btpsps), \lim_n \mathbb{P}(\theta_k^{\star} \in \mathcal{C}_{\alpha,k
           }^{\PSPS})=1-\alpha\).
180   \end{theorem}
181
182   \(\wh\V(\btpsps)\) can be obtained by applying the algebraic form of  \(\V(\btpsps)\) using
           the bootstrap estimators for \(\V(\wh \bt_\calL), \V(\wh \bmeta_\calL), \V(\wh \bt_\calL,
            \wh \bmeta_\calL),\) and \(\V(\wh \bmeta_\calU)\). The regularity conditions for
           consistent bootstrap variance estimation are outlined in Theorem 3.10 (i) of \citep{
           shao2012jackknife}. We also refer readers to \citep{hahn2021bootstrap}, which showed that
            bootstrap-based variance provides valid but potentially conservative inference.
183
```

```
184  This result indicates that a greater reduction in variance for the ML-assisted estimator is
         associated with larger values of \(\V(\wh \bt_\calL, \wh \bmeta_\calL)\) and smaller
         values of \(\V(\wh \bmeta_\calL)\), \(\V(\wh \bmeta_\calU)\), and \(\rho\). The variance
         reduction term \([\V(\wh \bt_\calL, \wh \bmeta_\calL)\trans( \V(\wh \bmeta_\calL) +  \rho
         \V(\wh \bmeta_\calU))^{-1}\V(\wh \bt_\calL, \wh \bmeta_\calL)]_{kk}\) can also serve as a
         metric for selecting the optimal ML model in ML-assisted inference.
185
186  Our next result shows that three existing methods, i.e., \texttt{PPI}, \texttt{PPI++}, and \
         texttt{PSPA}, are asymptotically equivalent to \PSPS~with different weighting matrices. A
          broader class for consistent estimator of \(\bt^*\) is \(\wh{\bt}(\bomega) = \bomega \
         trans\wh{\bmeta}_{\calU} - (\bomega \trans \wh{\bt}_{\calL} - \wh{\bmeta}_{\calL})\),
187  where \(\bomega\) is a \(K\times K\) matrix. The consistency of \(\wh{\bt}(\bomega)\)  for \(\
         bt^*\) only requires \(\bomega\trans(\wh{\bmeta}_{\calU} - \wh{\bmeta}_{\calL}) \coP \
         mathbf{0}\). Since \((\wh{\bmeta}_{\calU} - \wh{\bmeta}_{\calL}) \coP \mathbf{0}\),
         assigning
188  arbitrarily fixed weights for will satisfy the condition. However, the choice of weights
         influences the efficiency of the estimator as illustrated in \cref{prop:eff} later.
189
190  \begin{proposition}
191  \label{prop:equal}
192  Assuming equation~\eqref{eq:esteq} and regularity condition for the asymptotic normality of
         current ML-assisted estimator holds. For any M-estimation problem,
193  we have
194  \[
195  n^{\frac{1}{2}}(\wh{\bt}\left(\diag(\bomega_\textnormal{ele})\C\right) - \wh{\bt}_{\textnormal
         {\texttt{PSPA}}}) \coD
196  \0,
197  n^{\frac{1}{2}}(\wh{\bt}\left(\diag(\bomega_\textnormal{tr})\C\right) - \wh{\bt}_{\textnormal
         {\texttt{PPI++}}}) \coD \0,
198  n^{\frac{1}{2}}(\wh{\bt}\left(\diag(\1)\C\right) - \wh{\bt}_{\textnormal{\texttt{PPI}}})\coD
199  \0.
200  \]
201
202  Here, \(\bomega_\textnormal{ele} = [\omega_{\textnormal{ele}, 1}, \dots, \omega_{\textnormal{
         ele}, K}] \trans \in \RR^K)\) and \(\omega_{\textnormal{ele}, k}\) minimizing the $k$-th
         diagonal element of \(\V(\wh{\bt}(\bomega))\), \(\omega_{tr}\) is a scalar used to
         minimize the trace of \(\V(\wh{\bt}(\bomega))\), and \(\C\) is a matrix associated with
         the second derivatives of the loss function in M-estimation, with further details
         deferred to \cref{app:proof}.
203  \end{proposition}
204
205  This demonstrates that for M-estimation problems, our method is asymptotically equivalent to \
         texttt{PSPA}, \texttt{PPI++}, and \texttt{PPI} with the respective weights \(\diag(\
         bomega_\textnormal{ele})\C\), \(\diag(\bomega_\textnormal{tr})\C\), and  \(\diag(\1)\C\).
          Therefore, \PSPS~can be viewed as a generalization of these existing methods.
206
207  Our third result shows that the weights used in the
208  \cref{prop:equal} are not optimal. Instead, our choice of \(\bomega_0\) represents the optimal
         smooth combination of \((\wh{\bt}_{\calL}, \wh{\bmeta}_{\calL}, \wh{\bmeta}_{\calU})\)
         in terms of minimizing the asymptotic variance, while still preserving consistency.
209
210  \begin{proposition}
211  \label{prop:eff}
212  Suppose \(n^{1/2}(g(\wh{\bt}_{\calL}, \wh{\bmeta}_{\calL}, \wh{\bmeta}_{\calU}) - \bt^*) \coD
213  \mathcal{N}(0, \bSigma_g)\) and \(g\) is a smooth function, then \(\bSigma_{g_{kk}} \geq \
         bSigma_{\PSPS_{kk}}\)
214  \end{proposition}
215
216  Together with \cref{prop:equal}, our results demonstrate that our protocol provides a more
         efficient estimator compared to existing methods for the M-estimation problems.
         Furthermore, the applicability of our protocol is not limited to M-estimation and only
         requires summary statistics as input. It also indicates that in a setting of federated
         data analysis \citep{jordan2018communication} where individual-level data are not
         available, \PSPS~proves to be the optimal approach for combining shared summary
         statistics.
217
218  \begin{remark}
219  \texttt{PPI++} \citep{angelopoulos2023ppi++} employs a power-tuning scalar for variance
         reduction in ML-assisted inference. This scalar is obtained by minimizing the trace or
         possibly other scalarization of the estimator's variance-covariance matrix. However, the
         asymptotic variance of \PSPS~is always equal to or smaller than that of \texttt{PPI++},
         irrespective of the scalarization chosen by researchers. This advantage arises because \
         PSPS~utilizes a $K \times K$ power tuning matrix, $\bomega$, for variance reduction,
         where $K$ represents the dimensionality of parameters. This matrix facilitates
         information sharing across different parameter coordinates, thereby enhancing estimation
         precision. The choice of weighting matrix in \PSPS~also allows for element-wise variance
         reduction, reducing each diagonal element of the variance-covariance matrix. In contrast,
          the single scalar in \texttt{PPI++} can only target overall trace reduction or variance
         reduction of a specific element. A detailed example is provided in \cref{app:ppi++}. Only
          in one-dimensional parameter estimation tasks, such as mean estimation, \texttt{PPI++}
         and \PSPS~exhibit the same asymptotic variance.
```

```
220 | \end{remark}
```

## N.2 Canonical Code with Comments

```
1   import numpy as np
2   import pandas as pd
3   from scipy.stats import norm
4
5   def psps(est_lab_y, est_lab_yhat, est_unlab_yhat, Sigma, alpha=0.05):
6       """
7       Performs Post-Prediction Summary-statistics-based (PSPS) inference based on the paper
8       "Task-Agnostic Machine-Learning-Assisted Inference" (Miao & Lu, arXiv:2405.20039v3).
9
10      This function implements Algorithm 1 from the paper to combine estimates derived
11      from labeled data (using true outcomes Y and ML-predicted outcomes Yhat) and
12      unlabeled data (using ML-predicted outcomes Yhat) to produce a more efficient
13      estimator for a K-dimensional parameter theta*. It leverages summary statistics
14      (point estimates and their variance-covariance matrix) as input.
15
16      Functionality and Correspondence to Paper:
17      - Inputs correspond to summary statistics described in Algorithm 1, Step 2:
18          - `est_lab_y`: Corresponds to $\hat{\theta}_{\mathcal{L}}$, the K-dimensional point
                  estimate using observed Y in labeled data.
19          - `est_lab_yhat`: Corresponds to $\hat{\eta}_{\mathcal{L}}$, the K-dimensional point
                  estimate using predicted $\hat{f}$ in labeled data.
20          - `est_unlab_yhat`: Corresponds to $\hat{\eta}_{\mathcal{U}}$, the K-dimensional point
                  estimate using predicted $\hat{f}$ in unlabeled data.
21          - `Sigma`: Corresponds to the 3K x 3K joint variance-covariance matrix of the
                  estimators
22                  $[\hat{\theta}_{\mathcal{L}}^T, \hat{\eta}_{\mathcal{L}}^T, \hat{\eta}_{\
                      mathcal{U}}^T]^T$. The structure assumes independence between
23                  labeled and unlabeled data sets (covariance between estimators from
                      different sets is zero),
24                  consistent with the asymptotic variance structure in Eq (1). The function
                      comment notes
25                  `Sigma` is the direct variance, not the asymptotic variance scaled by n.
26          - `alpha`: Significance level for confidence intervals (e.g., 0.05 for 95% CI).
27      - Calculations implement Algorithm 1, Step 3:
28          - `K`: Dimension of the parameter theta.
29          - `v_est`: Extracts $Var(\hat{\theta}_{\mathcal{L}})$.
30          - `r`: Extracts $Cov(\hat{\eta}_{\mathcal{L}}, \hat{\theta}_{\mathcal{L}}) = Cov(\hat
                  {\theta}_{\mathcal{L}}, \hat{\eta}_{\mathcal{L}})^T$.
31          - `v_eta_lab`: Extracts $Var(\hat{\eta}_{\mathcal{L}})$.
32          - `v_eta_unlab`: Extracts $Var(\hat{\eta}_{\mathcal{U}})$.
33          - `V`: Calculates $Var(\hat{\eta}_{\mathcal{L}}) + Var(\hat{\eta}_{\mathcal{U}})$.
34          - `omega_0`: Computes the optimal weight matrix. Solves $V \omega_0^T = r$, yielding
35                  $\omega_0^T = (Var(\hat{\eta}_{\mathcal{L}}) + Var(\hat{\eta}_{\mathcal{U
                      }}))^{-1} Cov(\hat{\eta}_{\mathcal{L}}, \hat{\theta}_{\mathcal{L}})$
                      .
36                  This `omega_0` (as computed by `np.linalg.solve(V, r)`) is the transpose
                      of $\hat{\omega}_0$ defined in Algorithm 1 Step 3
37                  ($\hat{\omega}_0 = [Var(\hat{\eta}_{\mathcal{L}}) + Var(\hat{\eta}_{\
                      mathcal{U}})]^{-1} Cov(\hat{\theta}_{\mathcal{L}}, \hat{\eta}_{\
                      mathcal{L}})$).
38          - `est`: Calculates the PSPS point estimate $\hat{\theta}_{PSPS} = \hat{\theta}_{\
                  mathcal{L}} + \omega_0 (\hat{\eta}_{\mathcal{U}} - \hat{\eta}_{\mathcal{L}})$,
                  where `omega_0` is the variable computed in the code.
39                  Substituting `omega_0` (the code's variable) = $\hat{\omega}_{0, paper}^T$,
                      this becomes
40                  $\hat{\theta}_{PSPS} = \hat{\theta}_{\mathcal{L}} + \hat{\omega}_{0, paper}^T
                      (\hat{\eta}_{\mathcal{U}} - \hat{\eta}_{\mathcal{L}})$, which matches
                      the formula in Algorithm 1 Step 3.
41          - `standard_errors`: Calculates the standard errors for each element of $\hat{\theta}_
                  {PSPS}$.
42                  It computes $\sqrt{diag(Var(\hat{\theta}_{\mathcal{L}}) - \omega_0
                      ^T r)}$, where `omega_0` is the variable computed in the code
                      .
43                  This calculation corresponds to taking the square root of the
                      diagonal elements
44                  of the PSPS variance matrix $\hat{Var}(\hat{\theta}_{PSPS})$ as
                      defined in Algorithm 1 Step 3 and Theorem 1.
45                  (See Discrepancy section below).
46          - `lower_ci`, `upper_ci`, `p_values`: Calculate Wald-type confidence intervals and two
                  -sided p-values
47                                      based on the asymptotic normality of $\hat{\
                                          theta}_{PSPS}$ established
48                                      in Theorem 1.
49
```

```
50      Discrepancy between Code and Paper:
51      - Variance Calculation Detail: The code calculates the variance reduction term as `np.dot(
            omega_0.T, r)` which corresponds to $\omega_0^T r = \hat{\omega}_{0,paper} Cov(\hat{\
            theta}_{\mathcal{L}}, \hat{\eta}_{\mathcal{L}})^T$. The variance formula in the paper
             (Algorithm 1, Step 3; Theorem 1) involves $Cov(\hat{\theta}_{\mathcal{L}},\hat{\eta}
            _{\mathcal{L}})^{T} \hat{\omega}_{0,paper}$. These two terms, $\hat{\omega}_{0,paper}
             Cov^T$ and $Cov^T \hat{\omega}_{0,paper}$, are generally equal only if K=1 (scalar
            case) or under specific symmetry conditions on the covariance matrix $Cov(\hat{\theta
            }_{\mathcal{L}}, \hat{\eta}_{\mathcal{L}})$, which may not hold for K>1. Thus, the
            calculated standard errors might differ slightly from a direct implementation of the
            paper's variance formula for multivariate parameters (K>1).
52
53      Implementation Details Not Explicit in Paper's Algorithm 1:
54      - Asymptotic Normality Assumption: The calculation of confidence intervals (`norm.ppf`)
            and p-values (`norm.sf`) explicitly relies on the assumption that the PSPS estimator
            is asymptotically normally distributed. This is justified by Theorem 1 in the paper.
55      - Input Matrix Structure: The code assumes the input `Sigma` correctly represents the
            block variance-covariance matrix with zero covariance between estimators derived from
            independent labeled and unlabeled datasets.
56      - Matrix Invertibility: The code implicitly assumes that the matrix `V` ($Var(\hat{\eta}_
            {\mathcal{L}}) + Var(\hat{\eta}_{\mathcal{U}})$) is invertible, which is necessary to
             solve for `omega_0`. Theorem 1 also requires the corresponding asymptotic variance
            matrix to be invertible.
57      - Output Format: The function conveniently returns results in a pandas DataFrame, a common
             practice not specified in the paper's algorithm description.
58
59      Parameters:
60      - est_lab_y: A K-dimensional array of point estimates using Y in labeled data ($\hat{\
            theta}_{\mathcal{L}}$).
61      - est_lab_yhat: A K-dimensional array of point estimates using Yhat in labeled data ($\hat
            {\eta}_{\mathcal{L}}$).
62      - est_unlab_yhat: A K-dimensional array of point estimates using Yhat in unlabeled data ($
            \hat{\eta}_{\mathcal{U}}$).
63      - Sigma: A 3K x 3K variance-covariance matrix for the stacked vector of the above three
            estimators.
64      - alpha: Specifies the confidence level as 1 - alpha for confidence intervals.
65
66      Returns:
67      - A pandas DataFrame containing the PSPS summary statistics: 'Estimate' ($\hat{\theta}_{
            PSPS}$),
68        'Std.Error' ($\sqrt{diag(\hat{Var}(\hat{\theta}_{PSPS}))}$), 'Lower.CI', 'Upper.CI', and
             'P.value'.
69      """
70      K = len(np.array(est_lab_y))
71      Sigma_matrix = np.array(Sigma)
72      # Extract components from the variance-covariance matrix Sigma
73      # Corresponds to Var(theta_L) in paper notation
74      v_est = Sigma_matrix[0:K, 0:K]
75      # Corresponds to Cov(eta_L, theta_L) = Cov(theta_L, eta_L)^T
76      r = Sigma_matrix[K:2*K, 0:K]
77      # Corresponds to Var(eta_L)
78      v_eta_lab = Sigma_matrix[K:2*K, K:2*K]
79      # Corresponds to Var(eta_U)
80      v_eta_unlab = Sigma_matrix[2*K:3*K, 2*K:3*K]
81
82      # Calculate V = Var(eta_L) + Var(eta_U)
83      V = v_eta_lab + v_eta_unlab
84
85      # Calculate the optimal weight matrix (omega_0 in code = omega_0_paper^T)
86      # Solves V * omega_0^T = r => omega_0^T = V^{-1} * r
87
88      omega_0 = np.linalg.solve(V, r)
89      # Calculate PSPS point estimate: est = theta_L + omega_0_paper^T * (eta_U - eta_L)
90      # Here, omega_0 is KxK (omega_0_paper^T), (eta_U - eta_L) is Kx1 vector
91      est = est_lab_y + np.dot(omega_0, (est_unlab_yhat - est_lab_yhat))
92
93      # Calculate the variance of the PSPS estimate: Var(theta_PSPS) = Var(theta_L) - omega_0 *
            r
94      # Note: See discrepancy discussion above regarding paper's formula Var(theta_L) - Cov(
            theta_L, eta_L)^T * omega_0_paper
95      var_est_psps_matrix = v_est - np.dot(omega_0.T, r)
96      # Ensure the variance is non-negative due to potential numerical issues
97      # Taking diagonal before clipping to handle the matrix vs vector case correctly
98      variances = np.diag(var_est_psps_matrix)
99      standard_errors = np.sqrt(variances)
100
101     # Calculate confidence intervals using Normal approximation (justified by Theorem 1)
102     z_score = norm.ppf(1 - alpha / 2)
103     lower_ci = est - z_score * standard_errors
104     upper_ci = est + z_score * standard_errors
105
```

```
106        # Calculate p-values for H0: theta_k = 0 using Normal approximation
107        # Avoid division by zero standard error
108        p_values = np.full(K, np.nan) # Initialize with NaN
109        non_zero_se_indices = standard_errors > np.finfo(float).eps # Indices where SE is
               effectively non-zero
110        if np.any(non_zero_se_indices):
111            z_stat = np.abs(est[non_zero_se_indices] / standard_errors[non_zero_se_indices])
112            p_values[non_zero_se_indices] = 2 * norm.sf(z_stat) # sf = 1 - cdf
113
114        # Create output table
115        output_table = pd.DataFrame({
116            'Estimate': est,
117            'Std.Error': standard_errors,
118            'Lower.CI': lower_ci,
119            'Upper.CI': upper_ci,
120            'P.value': p_values
121        })
122
123        # Add metadata attribute to the DataFrame
124        output_table.attrs['Table Name'] = 'psps summary'
125        return output_table
```

### N.3 UNIT TEST CODE

```
1    class TestPSPSEquivalence(unittest.TestCase):
2
3        def setUp(self):
4            """Set up test fixtures."""
5            # --- Data for K=1 (scalar parameter) ---
6            self.est_lab_y_k1 = [0.5]
7            self.est_lab_yhat_k1 = [0.6]
8            self.est_unlab_yhat_k1 = [0.55]
9            self.Sigma_k1 = [[1.0, 0.2, 0.0],
10                             [0.2, 1.1, 0.0],
11                             [0.0, 0.0, 1.2]]
12            self.alpha_k1 = 0.05
13
14            # --- Data for K=2 (vector parameter) ---
15            self.est_lab_y_k2 = [0.5, 0.3]
16            self.est_lab_yhat_k2 = [0.6, 0.25]
17            self.est_unlab_yhat_k2 = [0.55, 0.28]
18
19            v_est_k2 = [[1.0, 0.1], [0.1, 0.9]]
20            cov_thL_etaL_k2 = [[0.2, 0.05], [0.03, 0.15]]
21            cov_etaL_thL_k2 = [[0.2, 0.03], [0.05, 0.15]]
22            v_eta_lab_k2 = [[1.1, 0.15], [0.15, 1.0]]
23            v_eta_unlab_k2 = [[1.2, 0.12], [0.12, 1.1]]
24
25            self.Sigma_k2 = [
26                [v_est_k2[0][0], v_est_k2[0][1], cov_thL_etaL_k2[0][0], cov_thL_etaL_k2[0][1],
                       0.0, 0.0],
27                [v_est_k2[1][0], v_est_k2[1][1], cov_thL_etaL_k2[1][0], cov_thL_etaL_k2[1][1],
                       0.0, 0.0],
28                [cov_etaL_thL_k2[0][0], cov_etaL_thL_k2[0][1], v_eta_lab_k2[0][0], v_eta_lab_k2
                       [0][1], 0.0, 0.0],
29                [cov_etaL_thL_k2[1][0], cov_etaL_thL_k2[1][1], v_eta_lab_k2[1][0], v_eta_lab_k2
                       [1][1], 0.0, 0.0],
30                [0.0, 0.0, 0.0, 0.0, v_eta_unlab_k2[0][0], v_eta_unlab_k2[0][1]],
31                [0.0, 0.0, 0.0, 0.0, v_eta_unlab_k2[1][0], v_eta_unlab_k2[1][1]],
32            ]
33            self.alpha_k2 = 0.05
34
35            self.est_lab_y_k1_edge = [0.1]
36            self.est_lab_yhat_k1_edge = [0.2]
37            self.est_unlab_yhat_k1_edge = [0.15]
38            self.Sigma_k1_edge = [[0.01, 0.5, 0.0],
39                                  [0.5,  1.0, 0.0],
40                                  [0.0,  0.0, 1.0]]
41            self.alpha_k1_edge = 0.05
42
43
44        def _run_and_compare(self, est_lab_y, est_lab_yhat, est_unlab_yhat, Sigma, alpha,
45                             expect_nan_se=False):
46            """Helper method to run both functions and compare their DataFrame outputs."""
47            est_lab_y_arr = np.array(est_lab_y)
48            est_lab_yhat_arr = np.array(est_lab_yhat)
49            est_unlab_yhat_arr = np.array(est_unlab_yhat)
50            Sigma_arr = np.array(Sigma)
```

```
51
52           canonical_output = canonical_psps_func(
53               est_lab_y_arr, est_lab_yhat_arr, est_unlab_yhat_arr,
54               Sigma_arr, alpha
55           )
56           generated_output = generated_psps_func(
57               est_lab_y_arr, est_lab_yhat_arr, est_unlab_yhat_arr,
58               Sigma_arr, alpha
59           )
60
61           if expect_nan_se:
62               self.assertTrue(canonical_output['Std.Error'].isnull().all(),
63                               "Canonical Std.Error should be NaN for this edge case.")
64               self.assertTrue(generated_output['Std.Error'].isnull().all(),
65                               "Generated Std.Error should be NaN for this edge case.")
66
67           assert_frame_equal(generated_output, canonical_output,
68                              check_dtype=True,
69                              rtol=1e-7,
70                              atol=1e-7)
71
72       def test_k1_equivalence(self):
73           """Test functional equivalence for K=1 (scalar parameter)."""
74           self._run_and_compare(
75               self.est_lab_y_k1, self.est_lab_yhat_k1, self.est_unlab_yhat_k1,
76               self.Sigma_k1, self.alpha_k1
77           )
78
79       def test_k2_equivalence(self):
80           """Test functional equivalence for K=2 (vector parameter)."""
81           self._run_and_compare(
82               self.est_lab_y_k2, self.est_lab_yhat_k2, self.est_unlab_yhat_k2,
83               self.Sigma_k2, self.alpha_k2
84           )
85
86       def test_k1_different_alpha(self):
87           """Test functional equivalence for K=1 with a different alpha."""
88           self._run_and_compare(
89               self.est_lab_y_k1, self.est_lab_yhat_k1, self.est_unlab_yhat_k1,
90               self.Sigma_k1, 0.1
91           )
92
93       def test_k1_edge_case_negative_variance(self):
94           """Test K=1 case where Var(theta_PSPS) might be negative, leading to NaN Std.Error."""
95           with np.errstate(invalid='ignore'):
96               self._run_and_compare(
97                   self.est_lab_y_k1_edge, self.est_lab_yhat_k1_edge, self.est_unlab_yhat_k1_edge
98                   ,
98                   self.Sigma_k1_edge, self.alpha_k1_edge,
99                   expect_nan_se=True
100              )
101
102      def test_statsmodels_derived_inputs_k1(self):
103          """Test K=1 with inputs from statsmodels GLMs and bootstrap, reading data from CSV."""
104
105          warnings.simplefilter('ignore', ConvergenceWarning)
106          warnings.simplefilter('ignore', PerfectSeparationError) # UserWarning for
                       PerfectSeparation
107          warnings.simplefilter('ignore', RuntimeWarning)
108
109          # 1. Load data from CSV files
110          lab_file_path = "../../test_data/lab.csv"
111          unlab_file_path = "../../test_data/unlab.csv"
112
113
114          lab_df = pd.read_csv(lab_file_path)
115          unlab_df = pd.read_csv(unlab_file_path)
116
117
118          # 2. Fit logistic regression models and extract the 'X' coefficient
119          try:
120              model_lab_y = glm('Y ˜ X', data=lab_df, family=sm.families.Binomial()).fit(disp=0)
121              est_lab_y_scalar = model_lab_y.params.get('X', 0.0)
122
123              model_lab_yhat = glm('Yhat ˜ X', data=lab_df, family=sm.families.Binomial()).fit(
                       disp=0)
124              est_lab_yhat_scalar = model_lab_yhat.params.get('X', 0.0)
125
126              model_unlab_yhat = glm('Yhat ˜ X', data=unlab_df, family=sm.families.Binomial()).
                       fit(disp=0)
127              est_unlab_yhat_scalar = model_unlab_yhat.params.get('X', 0.0)
```

```
128              var_unlab_yhat = model_unlab_yhat.bse.get('X', 0.1)**2
129          except (np.linalg.LinAlgError, PerfectSeparationError, Exception) as e:
130              # Catching a broader range of exceptions that might occur during fitting
131              self.skipTest(f"GLM fitting failed for initial models: {e}. Skipping test.")
132              return
133
134          est_lab_y = [est_lab_y_scalar]
135          est_lab_yhat = [est_lab_yhat_scalar]
136          est_unlab_yhat = [est_unlab_yhat_scalar]
137
138          # 3. Bootstrap for covariance calculation
139          B = 50
140          n_lab = len(lab_df)
141          np.random.seed(123)
142
143          est_lab_y_boot = np.full(B, np.nan)
144          est_lab_yhat_boot = np.full(B, np.nan)
145          successful_boot_fits = 0
146
147          for i in range(B):
148              boot_indices = np.random.choice(lab_df.index, size=n_lab, replace=True)
149              boot_lab_df = lab_df.iloc[boot_indices].reset_index(drop=True)
150
151              try:
152                  fit_y = glm('Y ~ X', data=boot_lab_df, family=sm.families.Binomial()).fit(disp
                          =0)
153                  est_lab_y_boot[i] = fit_y.params.get('X', est_lab_y_scalar)
154
155                  fit_yhat = glm('Yhat ~ X', data=boot_lab_df, family=sm.families.Binomial()).
                          fit(disp=0)
156                  est_lab_yhat_boot[i] = fit_yhat.params.get('X', est_lab_yhat_scalar)
157                  successful_boot_fits +=1
158              except (np.linalg.LinAlgError, PerfectSeparationError, Exception):
159                  est_lab_y_boot[i] = est_lab_y_scalar
160                  est_lab_yhat_boot[i] = est_lab_yhat_scalar
161
162          if successful_boot_fits < B / 2:
163               print(f"\nWarning (test_statsmodels_derived_inputs_k1): Only {
                          successful_boot_fits}/{B} bootstrap fits succeeded. Sigma matrix might be
                          less reliable.")
164
165          valid_boot_data = np.vstack([est_lab_y_boot, est_lab_yhat_boot])
166          valid_boot_data = valid_boot_data[:, ~np.isnan(valid_boot_data).any(axis=0)]
167
168          if valid_boot_data.shape[1] < 2:
169              cov_matrix_lab = np.array([[0.1, 0.01],[0.01, 0.1]])
170              print(f"\nWarning (test_statsmodels_derived_inputs_k1): Not enough valid bootstrap
                          samples ({valid_boot_data.shape[1]}) to compute reliable covariance. Using
                          fallback.")
171          else:
172              cov_matrix_lab = np.cov(valid_boot_data)
173
174          # 4. Compute the Sigma matrix
175          Sigma = np.zeros((3, 3))
176          if cov_matrix_lab.shape == (2,2):
177              Sigma[0:2, 0:2] = cov_matrix_lab
178          else:
179              Sigma[0:2, 0:2] = np.array([[0.1, 0.01],[0.01, 0.1]]) # Fallback
180              print("\nWarning (test_statsmodels_derived_inputs_k1): Covariance matrix from
                          bootstrap was not 2x2. Using fallback.")
181
182          Sigma[2, 2] = var_unlab_yhat
183
184          alpha = 0.05
185
186          # 5. Run and compare
187          self._run_and_compare(
188              est_lab_y, est_lab_yhat, est_unlab_yhat,
189              Sigma, alpha
190          )
191
192          warnings.simplefilter('default', ConvergenceWarning)
193          warnings.simplefilter('default', PerfectSeparationError)
194          warnings.simplefilter('default', RuntimeWarning)
195
196
197  if __name__ == '__main__':
198      # If running the file directly:
199      # Ensure canonical_psps.py and generated_psps.py are in the Python path
200      # and the CSV data files are at ./psps/test_data/
201      unittest.main()
```

## N.4 CODE GENERATION INSTRUCTION

```
1   I have a instruction for code generation task:
2   **Function Name:**
3
4   psps
5
6   **Functionality Description:**
7
8   Implement the Post-Prediction Summary-statistics-based (PSPS) inference procedure detailed in
        Algorithm 1 of the paper "Task-Agnostic Machine-Learning-Assisted Inference" (Miao & Lu,
        arXiv:2405.20039v3). This function combines statistical estimates derived from labeled
        data (using both true and predicted outcomes) and unlabeled data (using predicted
        outcomes) to calculate a more statistically efficient estimate for a K-dimensional
        parameter, $\theta^*$. The function operates on summary statistics (point estimates and
        their joint variance-covariance matrix) provided as input.
9
10  **Input/Output Specification:**
11
12  * **Input:**
13      * `est_lab_y` (list[float]): A list representing the K-dimensional point estimate vector
            obtained using true outcomes in the labeled dataset ($\hat{\theta}_{\mathcal{L}}$).
14      * `est_lab_yhat` (list[float]): A list representing the K-dimensional point estimate
            vector obtained using ML-predicted outcomes in the labeled dataset ($\hat{\eta}_{\
            mathcal{L}}$).
15      * `est_unlab_yhat` (list[float]): A list representing the K-dimensional point estimate
            vector obtained using ML-predicted outcomes in the unlabeled dataset ($\hat{\eta}_{\
            mathcal{U}}$).
16      * `Sigma` (list[list[float]]): A 3K x 3K list of lists representing the joint variance-
            covariance matrix of the stacked estimators $[\hat{\theta}_{\mathcal{L}}^T, \hat{\eta
            }_{\mathcal{L}}^T, \hat{\eta}_{\mathcal{U}}^T]^T$. It assumes independence between
            labeled and unlabeled sets (zero covariance between estimators from different sets).
17      * `alpha` (float): The significance level used for calculating confidence intervals (e.g.,
            0.05 for a 95% confidence interval). Defaults to 0.05.
18
19  * **Output:**
20      * A `pandas.DataFrame` with K rows and the following columns:
21          * `'Estimate'`: The PSPS point estimates ($\hat{\theta}_{PSPS}$).
22          * `'Std.Error'`: The standard errors for each element of $\hat{\theta}_{PSPS}$.
23          * `'Lower.CI'`: The lower bounds for the $(1-\alpha)$ confidence intervals.
24          * `'Upper.CI'`: The upper bounds for the $(1-\alpha)$ confidence intervals.
25          * `'P.value'`: The two-sided p-values for testing the null that each element of $\
                theta^*$ is zero.
```

