# OpenReview forum: "RECODE-H: A Benchmark for Research Code Development with Interactive Human Feedback"
_ICLR.cc/2026/Conference — ICLR 2026 Poster_

### Official Review · Reviewer_oRzg · 2025-10-23

**Soundness:** 2
**Presentation:** 2
**Contribution:** 2
**Rating:** 2
**Confidence:** 4

**Summary:**

In this paper, the authors present a new benchmark called RECODE, which is a set of 102 code execution tasks from research papers and their respective repositories. Their dataset is built via a clever combination of expert human annotation effort (using PhD-level experts to select the target code and repositories and provide code generation instructions) and LLM boostrapping using LLMs (e.g., for helping to annotate code explanations and construct on-the-fly unit tests). The particular focus of their dataset is on explicitly modeling multi-turn coding (intuitively, they say that the benchmark aims to capture standard programming where developers "interatively refine implementations through cycles of execution, debugging and feedback") and on testing directly the effect of different forms of feedback using a novel feedback hierarchy that they define in Section 3.2. This benchmark looks similar to those they cite in the RelatedWork, as well as some they don't cite, notably the work below (where the landmarks annotations appear to be similar in spirit to the types of feedback they collect):

[Bogin et al.] SUPER: Evaluating Agents on Setting Up and Executing Tasks from Research Repositories. EMNLP 2024

Most uniquely, however, their problems are annotated with different levels of feedback (generated using LLMs) using the feedback hierarchy from Section 3.2, and a set of concrete unit tests that allow them to measure functional code correctness.

They couple their benchmark with a new agent design called ReCodeAgent. Based on how this agent is described, both in Figure 1 as well as in the begining of Section 4, is it really unclear how this agent is nothing more than a ReACT agent or a closely related variant such as a Reflexion agent. Indeed, their earlier claim (starting on line 035) that "existing benchmarks .. for evaluating LLMs in research code generation mainly adopt a one-shot setting, where models are expected to produce final code in a single interaction* simply seems incorrect given that the models in virtually all of the studies they cite involve ReACT agents, much like ReCodeAgent, that engage in precisely the kind of act-observe loop they show visually in Figure 1. This point needs to be directly addressed by the authors and is the main source of my  concern about this paper.

Their main empirical results are show across seven LLMs (including 4 LLM model familities) in Table 2, where they also carefully report the performance effect of different types of feedback (not surprisingly, the most detailed feedback, level 4, is clearly the most hepful in improving end task performance).  This is coupled with other fairly expected conclusions, e.g., (citing the authors) *Model size and capability play a clear role in performance*. Further error analyis is provided (Table 3).

**Strengths:**

-- A new research coding benchmark with explicit feedback annotations that allow for more granular analysis of multi-turn coding. I could imagine this benchmark being used by others working in this area.

**Weaknesses:**

-- **Misleading motivation and discussion of past work**. As noted above, claims like "existing benchmarks for evaluating LLMs in research code generation mainly adopt a one-shot setting, where models are expected to produce final code in a single interaction" and "the ability of [LLMs] to generate correct and execute code remains limited" (line 011) seems inconsistent with virtually all of the papers cited. In the latter case, most studies involve REACT agents that are by design multi-turn agents. I would like to see the authors directly address.

--  **Limited to No Novelty of their ReCodeAgent** As noted above, their proposed solution, and their sole modeling approach, seems to be nothing more than a ReACT or Reflexion agent.

-- **Limited empirical validation** Experiments are limited only to their new dataset. Especially if they claim that their coding agent is unique, it would be expected that this approach shows improvements on other tasks.

**Questions:**

-- In what way is the agent workflow in Figure 1 (top left) not standard ReACT?

-- If it is different from ReACT, did you try to compare against a standard ReACT or relfexion approach?  Or compare your approach on other benchmarks?

---

> ### Author Response · Authors · 2025-11-23
> **Response Weakness 1 and Quetion 1**
>
> We thank the reviewer for the helpful feedback and would like to clarify the points below.
>
> **1. Clarifying “One-Shot” vs. “Interactive”**
>
> Our use of “one-shot” refers to the **human–agent interaction pattern**, not the agent’s internal ReAct loop.
> In existing research code benchmarks (e.g., SciReplicate-Bench, ResearchCodeBench), the user gives **one initial prompt**, and the agent then runs autonomously. All feedback comes from **execution logs only**, with no iterative human guidance.
> In contrast, **RECODE** evaluates multi-turn **human-in-the-loop refinement**, where a researcher provides structured feedback across Levels 1–4.
>
> **2. Why We State That Capabilities Remain Limited**
>
> Our claim refers to the **autonomous setting**: models struggle to generate correct research code without human feedback.
> This is reflected in **Level 0** (standard ReAct baseline), where performance remains low (e.g., GPT-5 MRR = 0.060; Recall = 0.294). Similar limitations are documented across prior work(e.g., SciReplicate-Bench, ResearchCodeBench).
>
> **3. How ReCodeAgent Differs From Standard ReAct**
>
> ReCodeAgent is built on ReAct but extends it by **expanding the observation space**:
>
> - Standard ReAct: Observation = execution logs/errors only.
> - ReCodeAgent: Observation = execution logs **+ structured researcher feedback**.
>
> This modification changes the Reflection stage: the agent incorporates external, research-specific guidance that standard ReAct does not use.
> At **Level 0**, our agent behaves exactly as standard ReAct/Reflexion. As shown in Table 2 in the paper, LLM agents consistantly benifit from higher level of feedback compared with standard ReAct/Reflexion.
>
> **4. Action**
>
> We have revise the agent design to clearly distinguish **internal agentic steps** from **multi-turn human–agent collaboration**, and to explicitly state that our contribution lies in evaluating and incorporating structured researcher feedback in Section 2 and 4.

---

> ### Author Response · Authors · 2025-11-23
> **Response Weakness 2**
>
> We appreciate the reviewer’s observation and agree that ReCodeAgent is built upon the ReAct paradigm. Importantly, our paper is primarily a *benchmark* contribution. The key novelty of our work lies in:
> 1. Introducing **RECODE**, a PhD-level, expert-annotated benchmark for *interactive* research code development.
> 2. Defining the first **structured hierarchical feedback protocol** for research-code implementation.
>
> ReCodeAgent serves as a strong, transparent baseline for demonstrating how different forms of feedback influence model performance under our benchmark—not as the core technical contribution of the paper.
>
> Nevertheless, ReCodeAgent is **not equivalent to standard ReAct or Reflexion**, and our extensions are necessary to support the benchmark’s feedback-rich setting:
>
> 1. Expanded observation space (main difference):
>   Classic ReAct agents observe only *execution logs*.
>   ReCodeAgent expands this to include the full **hierarchical researcher feedback** from RECODE (Levels 0–4), which provides error diagnosis, implementation paper alignment signals, domain knowledge, and (at Level 4) groundtruth code snippets. These signals cannot be inferred from logs alone and fundamentally change the nature of the reflection process.
>
> 2. Modified reflection step:
>   Because ReCodeAgent receives structured, research specific feedback, its reflection stage integrates external methodological guidance, repository integration hints, and diagnostic reasoning. Standard ReAct/Reflexion do not incorporate any such structured external feedback and thus cannot operate effectively under our benchmark’s feedback hierarchy.
>
> 3. Empirical behavioral distinction:
>   At **Level 0**, ReCodeAgent behaves identically to standard ReAct/Reflexion, which our results confirm.
>   As richer feedback is introduced, ReCodeAgent’s behavior diverges significantly because classical agents cannot consume or interpret these higher level feedback forms. This leads to the substantial performance improvements shown in Table 2 (e.g., GPT-5 recall improves from 29.4% to 71.6%), which would be impossible for a log-only ReAct/Reflexion agent.
> 4. Intentionally minimal method contribution:
>   Our aim is not to propose a new agent architecture but to isolate the *effect of feedback*. We intentionally avoid architectural innovations to ensure that improvements stem from the **feedback hierarchy**, not from agent engineering. This approach is consistent with benchmark design practice.
>
> In summary, while ReCodeAgent follows the general ReAct loop, it **extends the observation and reflection phases to support structured, researcher level feedback**, which is absent in prior work. The methodological contribution of ReCodeAgent is intentionally modest, as the novelty of our paper comes from **RECODE**, its **expert curated research code tasks**, and the **hierarchical feedback design**, not from proposing a fundamentally new agent framework.

---

> ### Author Response · Authors · 2025-11-23
> **Response Weakness 3 and  Question 2**
>
> Thank you for the question. We provide a concise clarification below.
>
> **1. Comparison to Standard ReAct / Reflexion**
>
> As mentioned in previous response, Table 2 provides a direct comparison between the baseline (Level 0) and our feedback-enabled variants (Levels 1–4).
> For example, GPT-5 Recall improves from **29.4% → 71.6%**, isolating the effect of feedback.
>
> **2. Evaluation on Other Benchmarks**
>
> Existing benchmarks fall into two categories, neither of which supports the core design of ReCodeAgent.
> 1. SWE oriented benchmarks (e.g., SWE-Bench, InterCode, BigCodeBench) are designed for software engineering tasks and lack the research and paper context that ReCodeAgent relies on. They do not provide paper code alignment, methodological descriptions, or domain specific rationale. So applying ReCodeAgent to them would collapse it into a standard ReAct/Reflexion agent, defeating its purpose.
>
> 2. Research code benchmarks (e.g., SciReplicate-Bench, ResearchCodeBench) do not include the **expert curated annotations** required for high-quality researcher feedback, such as detailed functional explanations, discrepancy notes, and repository level guidance. Without such annotations, feedback cannot be accurate or reliable, and ReCodeAgent’s structured feedback integration cannot be properly evaluated.
> For these reasons, existing benchmarks cannot meaningfully assess the interactive, research-focused capabilities that ReCodeAgent is designed to test.

---

> ### Author Response · Authors · 2025-11-25
> **Difference with SUPER**
>
> **1. Discussion**
> SUPER evaluates agents on repository deployment and execution—tasks such as setting up environments, running scripts, and resolving runtime or configuration errors. These tasks focus on executing existing code. In contrast, RECODE focuses on algorithm and method implementation: agents must generate missing research code (e.g., forward passes, loss functions, data processing modules) directly from paper descriptions. This requires interpreting methodological details, not executing repositories.Therefore, the problem setting and capabilities evaluated are fundamentally distinct.
>
> **2. Action**
> We have cited SUPER and discussed in Section 2.
>
> > L.128 :"
> >  SUPER (Bogin et al., 2024) and CSR-Bench (Xiao et al., 2025) evaluate LLM agents in the aspects of repository deployment and
> execution, assessing their ability to configure environments, resolve dependency issues, and reliably run experimental workflows.
> > "

---

> ### Author Response · Authors · 2025-11-27
> **Official Comment by Authors**
>
> Dear Reviewer oRzg,
>
> We sincerely thank you for early detailed review. We would like to briefly follow up to inquire whether our earlier responses adequately addressed the concerns raised. If any questions remain or if further clarification would be helpful, we would be grateful for the opportunity to provide additional information.
>
> If there are no remaining issues, we would be deeply grateful if you might consider raising the score, if the reviewer find that our revisions have strengthened the quality of the manuscript. Regardless of the final decision, we sincerely appreciate the time, effort, and thoughtful consideration you have dedicated to evaluating our work.
>
> Thank you again for your valuable feedback.
>
> Warm regards,
>
> The Authors

---

### Official Review · Reviewer_61B7 · 2025-10-31

**Soundness:** 3
**Presentation:** 3
**Contribution:** 2
**Rating:** 4
**Confidence:** 4

**Summary:**

This paper presents RECODE, a benchmark of 102 research coding tasks in AI/ML to study the interactions between LLM-based agents and user feedback. The user feedbacks are simulated with LLMs and controlled to have five different granularity levels of information. Experimental results show that LLMs benefit from additional feedback, especially straightforward ones, but still struggle with those requiring deeper understanding of the research tasks.

**Strengths:**

1. The paper adds a nice new dimension around user feedback to the evaluation of agents for research coding, which is helpful and adheres to the real-world use cases of such agents.
2. The paper presents a reasonable amount of analysis of experimental results. The error analysis and feedback adoption analysis are interesting and may facilitate future research.

**Weaknesses:**

1. While it is appreciated that the authors assembled a team of 26 annotators, their roles in the entire annotation process are not very clear to me. It seems that LLMs (Gemini 2.5 Pro and GPT-4o-mini) are used to perform many annotations. What are the jobs of the annotators and how their involvement ensures the benchmark’s quality and real-world utility?
2. Relatedly, the quality of LLM-generated unit tests is unclear and not thoroughly discussed in the paper, which is critical to the reliability of the experimental results. It is also unclear to me how the “unit tests” can be leveraged to evaluate some tasks, such as training machine learning models or analyzing data.
3. The authors should list and cite all the papers adapted. Meanwhile, discussions and tests of data contamination are also missing.
4. The validity of “level 4” feedback, which provides ground truth code, may not be a valid setting since it directly “supervises” the code generation process and plays an exceptionally helpful role for most models. The evaluations would be more sound and clean by just stopping at “level 3.”
5. Some example tasks in the appendix would be appreciated to help the benchmark description be more grounded.

**Questions:**

Please see weaknesses.

**Details Of Ethics Concerns:**

All of the papers and resources used by this benchmark are not cited or attributed. This may hurt the original authors' intellectual properties and be subject to copyright and terms of use issues.

---

> ### Author Response · Authors · 2025-11-23
> **Response to Weakness 1**
>
> We thank the reviewer for the opportunity to clarify the annotation protocol. While LLMs were utilized to scale the drafting process and ensure formatting consistency, the quality and real-world utility of RECODE rely strictly on expert human oversight.
>
> Their role was not passive; they acted as the primary guarantors of correctness through three critical stages:
>
> 1. Expert Curation & Execution Verification
> Unlike automated scraping, our experts manually selected papers and repositories based on criterions. Crucially, to ensure real-world utility, annotators executed the official scripts for every selected repository to verify that the code runs successfully and fits within standard hardware constraints (<24GB GPU memory).
>
> 2. Semantic Alignment
> Mapping abstract paper descriptions to concrete code is a complex task where LLMs often hallucinate. Our annotators manually identified the specific code segments corresponding to paper methods. While Gemini-2.5-Pro drafted initial explanatory comments, human experts reviewed and validated every instance to ensure the logic flows and implementation details were accurately captured.
>
> 3. Rigorous Quality Control & Test Coverage
> For the unit tests, LLMs provided only candidate cases. The annotators were responsible for manually reviewing and refining these tests to guarantee they were correct and achieved 80% coverage of the canonical code. This strict metric ensures that the benchmark evaluates robust functional correctness rather than just surface-level syntax.

---

> ### Author Response · Authors · 2025-11-23
> **Response Weakness 2**
>
> We thank the reviewer for raising this important question regarding the reliability and applicability of our evaluation tests. We wish to clarify below that **the unit tests used in RECODE are not generated by LLMs**, and that all evaluation logic is the result of rigorous human expert design.
>
> **1. Human-Designed Evaluation Logic (LLMs Only Draft Initial Scaffolds)**
> To reduce annotator workload, LLMs were used **only to draft initial test-case templates**, such as simple input output scaffolds.
> However, **LLMs did not design the evaluation logic, correctness criteria, or assertions**, nor did they determine what should be tested.
>
> All substantive components of every test were **authored, validated, and finalized by PhD-level annotators**, including:
>
> 1. Specification of correctness conditions and expected behaviors,
> 2. Construction of assertions and comparison logic,
> 3. Alignment of tests with canonical implementations,
> 4. Verification of algorithmic intent based on the original research code and paper.
>
> Each test underwent detailed human review to ensure rigor, completeness, and coverage. We additionally enforced an **80% code coverage requirement**, ensuring that tests thoroughly exercise the intended logic rather than relying on superficial checks.
>
> **2. Applicability to Tasks Such as Training Loops and Data Analysis**
> We agree that classical unit tests cannot directly validate end-to-end processes like full model training or dataset-level analyses.
> For such tasks, RECODE uses **annotator designed differential tests**, not simple functional unit tests.
>
> 1. Annotators designed tests that run both the **canonical implementation** and the **model-generated implementation** under identical seeds and inputs.
> 2. The tests compare **intermediate quantities**, such as loss values, gradients, parameter updates, or processed data—within standard numerical tolerances.
> 3. This method validates the **algorithmic correctness** of steps like `train_step` without requiring costly or noisy convergence based evaluation.
>
> This ensures that even complex research tasks are evaluated on the correctness of their underlying computation, not on complex training outcomes.
>
>
> **Action:** In revision, we have added the discussion about unit test in section 3.1

---

> ### Author Response · Authors · 2025-11-23
> **Response Weakness 3**
>
> We thank the reviewer for highlighting these crucial aspects of benchmark validity.
>
> **1. Full Citation List**
> We acknowledge the importance of listing the adapted papers. We have reveal the paper list in Appendix K.
>
> **2. Data Contamination Analysis**
> Ensuring that evaluated LLMs have not seen the source repositories during pre-training is challenging for models released after late 2023. Recent frontier models (e.g., GPT-5 in 2024-09, Gemini-2.5 in 2025-01, Claude 4 Sonnet in 2025-03) are trained on large, opaque, and continually updated corpora that may include publicly available code and research papers. As a result, constructing a fully contamination-free benchmark is practically infeasible.
>
> To estimate the effect of potential pre-training contamination, we analyze model performance as a function of the publication year of the source papers. Our dataset includes tasks from **2023 (16.83%)**, **2024 (53.47%)**, and **2025 (29.70%)**. If contamination were a major factor, tasks from earlier years—more likely to have appeared in pre-training data—should yield higher performance than more recent tasks.
>
> Across all seven LLMs and all evaluation metrics (**MRR**, **Recall**, and execution-based scores**), we observe the opposite trend: **2023 tasks produce the lowest performance**, **2024 tasks perform substantially better**, and **2025 tasks match or exceed 2024 performance** for the strongest models. This pattern is inconsistent with a contamination-driven explanation.
>
> These results suggest that pre-training contamination is unlikely to be the dominant factor shaping model performance. Variations in task difficulty and model–task capability alignment provide a more plausible explanation for the observed differences across years.
>
> | Model              | Year | MRR   | Recall | Test Case | CodeBLEU | CodeBert |
> |--------------------|-------|--------|---------|--------------|------------|-----------|
> | **GPT-5**          | 2023 | 0.116 | 0.588 | 0.806 | 0.297 | 0.884 |
> |                    | 2024 | 0.230 | 0.685 | 0.847 | 0.322 | 0.890 |
> |                    | 2025 | 0.277 | 0.833 | 0.853 | 0.318 | 0.888 |
> | **GPT-5-mini**     | 2023 | 0.065 | 0.294 | 0.659 | 0.288 | 0.890 |
> |                    | 2024 | 0.187 | 0.741 | 0.856 | 0.306 | 0.890 |
> |                    | 2025 | 0.216 | 0.733 | 0.806 | 0.329 | 0.889 |
> | **GPT-5-nano**     | 2023 | 0.051 | 0.176 | 0.490 | 0.275 | 0.885 |
> |                    | 2024 | 0.097 | 0.352 | 0.562 | 0.301 | 0.853 |
> |                    | 2025 | 0.106 | 0.433 | 0.536 | 0.283 | 0.849 |
> | **DeepSeek**       | 2023 | 0.111 | 0.588 | 0.802 | 0.342 | 0.901 |
> |                    | 2024 | 0.212 | 0.704 | 0.817 | 0.342 | 0.897 |
> |                    | 2025 | 0.234 | 0.767 | 0.830 | 0.343 | 0.896 |
> | **Gemini-2.5 Pro** | 2023 | 0.127 | 0.588 | 0.723 | 0.332 | 0.897 |
> |                    | 2024 | 0.202 | 0.574 | 0.676 | 0.336 | 0.891 |
> |                    | 2025 | 0.154 | 0.600 | 0.740 | 0.330 | 0.879 |
> | **Gemini-2.5 Flash**| 2023 | 0.090 | 0.471 | 0.727 | 0.331 | 0.903 |
> |                    | 2024 | 0.161 | 0.630 | 0.782 | 0.355 | 0.900 |
> |                    | 2025 | 0.128 | 0.567 | 0.784 | 0.354 | 0.896 |
> | **Claude**         | 2023 | 0.087 | 0.412 | 0.561 | 0.336 | 0.891 |
> |                    | 2024 | 0.109 | 0.500 | 0.675 | 0.334 | 0.894 |
> |                    | 2025 | 0.130 | 0.467 | 0.622 | 0.337 | 0.892 |
>
> **Action:** We add the discussion about data contamination in Appendix C.

---

> ### Author Response · Authors · 2025-11-23
> **Response Weakness 4**
>
> We thank the reviewer for this thoughtful suggestion. We acknowledge that **Level 4 feedback** (providing the correct code snippet) represents a highly supervised setting that differs significantly from the open-ended problem-solving required in Levels 0–3. However, we respectfully argue that maintaining Level 4 is essential for the completeness of the benchmark for three key reasons, which we have clarified in the revision:
>
> **1. Prectical Application of Feedback Level 4**
> As stated in the introduction, RECODE is designed to simulate "realistic researcher agent collaboration". In real-world research workflows, a senior researcher or mentor often identifies a specific implementation error and provides the exact correction to a junior engineer or assistant (e.g., "You utilized the wrong masking operation; use this specific scatter-gather pattern instead"). Level 4 simulates this specific "Code Review" interaction. It evaluates whether the agent can successfully act as a collaborator that recognizes and integrates a provided solution, rather than just generating one from scratch.
>
> **2. Establishing an "Integration Capability" Upper Bound**
> We utilize Level 4 not to test *generation reasoning*, but to establish the **upper bound** of a model's "teachability" and instruction adherence. It answers the critical question: *“If an expert provides the exact solution, is the model capable of recognizing and integrating it without breaking the existing repository?”*
> Our results demonstrate that this is a non-trivial task. As shown in **Table 2**, even with Level 4 feedback, state-of-the-art models do not achieve 100% success within 10 turns of interaction of feedback. For instance, **DeepSeek-V3.1** achieves a Recall of only **70.6%** at Level 4, and **Gemini-2.5-pro** achieves a Recall of **58.8%**. This indicates that even when the "answer" is provided, models still struggle with the complex context management required to integrate it into a repository. Removing Level 4 would hide these specific integration failures.
>
> **3. Disentangling Reasoning Gaps from Adoption Gaps**
> By comparing Level 3 (Natural Language Guidance) and Level 4 (Code Snippet), we can diagnose the source of failure.
> 1. Adoption Analysis: As detailed in **Table 8**, feedback adoption is not guaranteed even when the fix is explicit. For example, **Gemini-2.5-pro** only adopts the provided feedback **58.9%** of the time at Level 4.
> 2. Error Types: Models still commit **Type 4 (Repository Integration)** errors even when the logic is provided. Level 4 is therefore necessary to measure the ability of the model to override its internal priors when presented with ground truth.
>
> We view Level 4 as a distinct "Oracle" baseline that complements the generative challenges of Levels 0–3. To address the reviewer's concern, we have revised **Section 3.2** to explicitly frame Level 4 as a reference setting for evaluating **integration and instruction adherence**, distinguishing it from the generative reasoning evaluated in lower levels.

---

> ### Author Response · Authors · 2025-11-23
> **Response Weakness 5**
>
> Thanks for the insightful suggestion. The example of the annotation is added in the appendix N.

---

> ### Comment · Reviewer_61B7 · 2025-11-24
>
> Thanks for the detailed responses and paper revision! I have raised my score accordingly and removed the ethics review flag.

---

> ### Author Response · Authors · 2025-11-27
> **Official Comment by Authors**
>
> Dear Reviewer 61B7,
>
> Thank you for your thoughtful followup and for updating your score from 4 to 8. Please feel free to follow up if you have further comments.
>
> Best regards,
>
> The Authors

---

### Official Review · Reviewer_AynR · 2025-10-31

**Soundness:** 3
**Presentation:** 3
**Contribution:** 2
**Rating:** 6
**Confidence:** 3

**Summary:**

The authors propose RECODE, a new benchmark for generating research code in the novel setting of multi-turn interactions with a (simulated) human, which iteratively provides hints/details/corrections to help steer the system. They show that interactive feedback helps, while also highlighting that models still struggle.

**Strengths:**

- Introducing the interactive multi-turn angle is novel and refreshing, and adds some extra realism to the coding challenge
 - novel framing, with levels of feedback
 - benchmark looks like a useful product of a lot of hard work
 - evaluation is thorough

**Weaknesses:**

- The interactive feedback is (for reproducibility) actually simulated so has a degree of artificality associated with it (even though humans helped create it). In what ways does this setup differ from a (truely) real setup with actual humans, e.g., noise, imprecision, etc. Could you include any of those elements in your framework?
 - The conclusions from the experiments seem obvious (e.g., more feedback helps). What did you learn that was surprising/interesting/informative? It seems to me that should be more nuanced findings. The error analysis is perhaps more informative. If you were to advise future researchers on where to invest their energy in building better coding agents, what would you tell them based on the learnings from your work?

Minor:
 - "evaluates LLMs through multi-turn interacations with human feedback" - makes it sound like there's a human in the eval loop. Perhaps "(pre-collected) human feedback" or "(simulated) human feedback" or something else to indicate the actual eval is fully automated.
 - Figure 2 would be more readable showing % rather than fractionals (e.g., "1.4" rather than "0.014" etc.)

**Questions:**

See weaknesses. Also:
 - Isn't level 4 feedback basically giving the system the answer? Why doesn't the coding agents score 100% as a result? Adding some discussion around this would be interesting, in particular that "coding" requires more than just knowing lines of code. What would you need to include in a "level 5" feedback to ensure the coding agents did score 100%?
 - Do your results suggest any advice for what *kinds* of feedback people should be giving to their coding agents, i.e., provide insights that make interactive coding agents more usable/effective?
  - The Conclusion seems pretty weak, surely there's more to conclude than just that LLMs continue to face coding challenges? What are the big insights you found?
   - Building/exanding this benchmark looks very labor intensive. Can you think of ways you might reduce the cost / semi-automate the process to extend it further?

---

> ### Author Response · Authors · 2025-11-23
> **Response Weaknesse 1**
>
> We appreciate the reviewer’s question regarding the differences between our simulated feedback and truly human feedback. As shown in Appendix E, we directly compared LLM augmented feedback with human annotated feedback, and the results indicate clear qualitative differences: human feedback is typically more partial, noisier, less specific, and less prescriptively corrective. This reduced clarity leads to smaller performance gains compared with LLM augmented feedback, which aligns with observations from prior interactive feedback benchmarks such as MINT and ConvCodeWorld.
>
> Our framework is designed to accommodate these properties. In principle, we can simulate several forms of human like noise by introducing intentional perturbations, such as probabilistic error injection, incomplete or underspecified messages, annotator style variability, or feedback drawn from distributions parameterized by annotator expertise. These mechanisms can be integrated into the existing feedback-sampling pipeline without modifying the overall framework.
>
> Accurately capturing the diverse spectrum of real human feedback, including heterogeneity in expertise, communication style, contextual awareness, and task dependent reasoning,remains an open challenge. Developing realistic generative models of human feedback and principled metrics for quantifying human synthetic alignment is an important direction for future work.
>
> **Action:** We have added the discussion in Appendix E.

---

> ### Author Response · Authors · 2025-11-23
> **Response Weakness 2 and Question 2**
>
> We thank the reviewer for the constructive comments. While it is intuitive that richer feedback improves performance, our study reveals several non-obvious and practically meaningful findings.
>
> **1. Key surprising insights.**
> 1. Semantic fidelity the real bottleneck. Our error analysis shows that modern LLMs rarely fail due to syntax or programming mechanics. Instead, failures are dominated by *implementation alignment errors* and *missing domain knowledge*, indicating that basic coding is largely solved while faithful interpretation of research logic remains challenging.
> 2. Feedback effects are highly non-linear. The largest performance gain occurs between *no feedback* and *minimal error description feedback*. Inexpensive diagnostic signals produce disproportionate improvements, whereas richer natural language feedback, which generally consume more human analysis burden, yields diminishing returns until explicit code snippets are provided.
> 3. Models differ sharply in feedback adaptability. GPT-5 and DeepSeek-V3.1 consistently leverage richer guidance, while Claude-Sonnet-4 and Gemini-2.5-pro plateau early, showing that feedback utilization is a model-dependent capability, not merely a function of scale.
>
> **2. Implications for future research.**
> Our findings suggest shifting focus from general coding competence toward improving
> 1. Semantic reasoning over research descriptions.
> 2. Alignment with methodological intent.
> 3. Stable multi-turn refinement.
>
> **3. Practical guidance for interactive coding agents.**
> Users obtain the greatest return by providing *low-cost execution feedback* and reserving human effort for clarifying semantic misalignments. For models with weak feedback uptake, escalating earlier to concrete code snippets is more effective than providing additional high level critique.
>
> We hope this clarified summary highlights the deeper contributions of our work beyond the expected observation.
>
> **Action:** We have updated the conclusion with adding more insights in section 5.5.

---

> > ### Comment · Reviewer_AynR · 2025-11-25
> >
> > Thank you for the clarifications and paper updates!

---

> ### Author Response · Authors · 2025-11-23
> **Reponse Question 1**
>
> We appreciate this thought provoking question.
>
> **1. Why Level 4 is not immediately 100%**
> In the main results, the maximum number of interaction turns is limited to **10**. Within this bound, even with Level 4 feedback—which already provides the correct code snippet—models do not reach 100% because repository-level coding involves more than possessing the correct algorithm. Additional turns are needed to handle context alignment, fix integration-related issues, and resolve secondary errors created during earlier revisions. The gap below 100% at 10 turns therefore reflects interaction limits, not a deficiency of Level 4.
>
> **2. Extended Interaction Shows Continued Improvement**
> To examine whether Level 4 feedback is ultimately sufficient, the interaction limit is extended to **15 turns**. As shown below, GPT-5 continues to improve substantially beyond the 10-turn horizon:
>
> | Feedback Level | Recall@15 (GPT-5) |
> |----------------|-------------------|
> | Level 0 | 0.549 |
> | Level 1 | 0.779 |
> | Level 2 | 0.880 |
> | Level 3 | 0.963 |
> | **Level 4** | **0.972** |
>
> These results indicate that performance has **not** saturated: with more turns, Level 4 feedback drives the model extremely close to 100%, and the trend suggests that performance would continue approaching 1.0 if additional interaction rounds were allowed. Level 4 already provides all necessary information; the remaining gap reflects integration friction rather than missing supervision.
>
> **3. On the Concept of “Level 5”**
> A hypothetical “Level 5” should be viewed purely as a **theoretical upper bound** that could force 100% accuracy, but it is not meaningful for evaluating LLM capabilities in copying the input code. Since Level 4 already conveys all relevant semantic information for correct implementation, anything beyond it would collapse the task into trivial file replacement instead of assessing reasoning or code integration ability. For this reason, the feedback hierarchy is kept at Levels 0–4, which capture the practically relevant spectrum of researcher-style guidance.
>
> **Action:** We add the discussion on extended feedback interaciton result in appendix H.

---

> ### Author Response · Authors · 2025-11-23
> **Response Question 3**
>
> We thank the reviewer for this helpful suggestion. We agree that our results support a stronger set of takeaways than simply noting that LLMs still struggle with coding. In the revision, we will update the Conclusion section of the PDF to explicitly incorporate the following key insights:
>
> **1. The bottleneck has shifted from syntax to scientific grounding.**
> Our error analysis shows that modern LLMs rarely fail due to basic coding mistakes. Instead, the dominant issues are misinterpreting research descriptions and missing implicit domain knowledge. This indicates that the core challenge is no longer syntax, but faithfully grounding high-level scientific narratives into code.
>
> **2. Minimal feedback produces the largest non-linear gains.**
> Across models, the biggest jump in performance occurs from Level 0 to Level 1 feedback. This “diagnostic nudge” effect suggests that models often have the latent capability to solve a task but need only lightweight signals to correct their reasoning paths.
>
> **3. Feedback receptivity is a distinct capability.**
> Our results show that the ability to generate code and the ability to incorporate feedback are separable skills. Stronger models benefit consistently from richer feedback, while weaker models plateau or even degrade when feedback becomes more detailed, revealing limits in their interactive adaptability.
>
>
> **Action:** We have update the conclusion in Section 6 with adding more insights.

---

> ### Author Response · Authors · 2025-11-23
> **Response Question 4**
>
> We thank the reviewer for this helpful suggestion. We agree that manual curation by domain experts is resource-intensive.
> To address the concern regarding scalability and future expansion, we would like to highlight that our current pipeline is already heavily semi-automated, and we have identified clear pathways to further automate the process:
>
>
> **1. Leveraging the Existing "Human-in-the-Loop" Pipeline**
> As detailed in Section 3.1, we currently employ a hybrid pipeline where LLMs handle the heavy lifting of content generation, while domain experts focus on verification. Specifically:
>
> 1. Annotation & Instructions:
> We utilize Gemini-2.5-Pro to generate initial drafts of explanatory comments and code generation instructions, which are then refined by annotators.
>
> 2. Case Generation:
> We leverage Gemini-2.5-Pro to automatically generate candidate test cases.
>
> 3. Feedback:
> We employ GPT-o4-mini to simulate expert feedback, ensuring scalability without requiring real-time human writing for every interaction. By shifting the human role from *creator* to *verifier*, we significantly reduce the time cost per task compared to traditional manual annotation.
>
>
> **2. Future Strategy: Automated Filtering & Pre-Validation**
> To extend the benchmark further, we propose introducing an automated **Validation Agent** phase to reduce the burden on human reviewers.
>
> 1. Automated Test Validation:
> Currently, humans carefully review generated unit tests. In the future, we can automate this by executing LLM-generated tests against the provided canonical code. Only tests that pass the canonical implementation (and correctly fail on mutated code) would be presented to human annotators, filtering out low quality generations automatically.
>
> 2. Automated Repository Selection:
> We can further automate the initial selection process (Section 3.1) by scraping repositories that meet strict “auto-metrics,” such as ensuring the presence of official execution scripts and low hardware requirements. This ensures that the semi-automated pipeline begins with high quality contexts, minimizing the “repair” time needed by human experts.

---

> ### Author Response · Authors · 2025-11-27
> **Official Comment by Authors**
>
> Dear Reviewer AynR,
>
> Thank you for your valuable feedback. If there are any remaining questions, please feel free to post them. We would be very happy to respond promptly and actively.
>
> If the current revised version has satisfactorily addressed your concerns, we would respectfully ask whether you might reconsider the rating of our work. Regardless of the outcome, we fully respect your final evaluation and are sincerely grateful for the time and care you have devoted to reviewing our submission.
>
> Thank you again for your thoughtful review.
>
> Best regards,
>
> The Authors

---

### Official Review · Reviewer_6h4u · 2025-11-01

**Soundness:** 3
**Presentation:** 2
**Contribution:** 3
**Rating:** 6
**Confidence:** 4

**Summary:**

This paper presents RECODE, a benchmark for 102 code-generation tasks, based on LaTeX snippets from recent ML papers. They introduce a ReAct-style multi-turn ReCodeAgent that interacts with a code repo and receives feedback from a simulated expert (similar to LLM as a judge). The authors find that richer feedback substantially improves performance, though with diminishing returns. The paper also analyzes error types and how effectively models adopt feedback.

**Strengths:**

1. The empirical findings are valuable to broader coding agents: early feedback levels yield large improvements, while higher levels show smaller gains; and that most model failures stem from misunderstanding the paper or repo semantics.
2. The benchmark targets realistic, interactive research-coding scenarios, which are increasingly relevant as LLM agents enter scientific domains.
3. Solid engineering effort, clear and intuitive definitions and hierarchies of feedback

**Weaknesses:**

1. It is not clear how the papers were selected and the list of papers is not disclosed. How do you ensure that evaluated LLMs haven’t already seen the source repositories during pretraining?
2. There is no code editing tool in the system prompt. The replace tool replaces the entire file content, but the prompt mentions "aim for editing the code more" -- it is not clear how this is implemented.
2. LLM-generated feedback is clean and complete; real human feedback is often noisy or partial. But results with human-generated feedback on these tasks are missing.
2. Prior benchmarks like SciReplicate-bench and ResearchCodeBench already explore research-code generation; the paper needs a clearer statement of what’s uniquely new here.

**Questions:**

See above weaknesses.
I am curious if you had results with a single-turn format. It would be interesting to see if the multi-turn format gives a big lift.
Minor typos/issues: Fig 7 and 8 have same captions

---

> ### Author Response · Authors · 2025-11-23
> **Response Weakness 1**
>
> Thank you for the insightful comment.
>
> **1. How the papers were selected.**
>
> To clarify our paper-selection process: all papers and repositories in RECODE were manually curated by expert annotators covering diverse AI research domains, including machine learning algorithms, NLP, computer vision, graph learning, time-series modeling, diffusion models, and recommendation systems. The paper selection follows the following restrictions:
>
> 1. Papers should be published in 2023 or later.
>
> 2. Papers should be published at top CS conferences (CVPR, ICML, NeurIPS, ICLR).
>
> 3. Papers should publish their official repository. The repository is correct and executable.
>
> 4. The paper and repository have clear correspondence between methodological descriptions and code structure.
>
> This curation protocol ensures that the benchmark reflects realistic, well-specified, and executable research code tasks.
>
>
> **2. How do you ensure that evaluated LLMs haven’t already seen the source repositories during pretraining?**
>
> Ensuring that evaluated LLMs have not seen the source repositories during pre-training is challenging for models released after late 2023. Recent frontier models (e.g., GPT-5 in 2024-09, Gemini-2.5 in 2025-01, Claude 4 Sonnet in 2025-03) are trained on large, opaque, and continually updated corpora that may include publicly available code and research papers. As a result, constructing a fully contamination-free benchmark is practically infeasible.
>
> To estimate the effect of potential pre-training contamination, we analyze model performance as a function of the publication year of the source papers. Our dataset includes tasks from **2023 (16.83%)**, **2024 (53.47%)**, and **2025 (29.70%)**. If contamination were a major factor, tasks from earlier years—more likely to have appeared in pre-training data—should yield higher performance than more recent tasks.
>
> Across all seven LLMs and all evaluation metrics (**MRR**, **Recall**, and execution-based scores), we observe the opposite trend: **2023 tasks produce the lowest performance**, **2024 tasks perform substantially better**, and **2025 tasks match or exceed 2024 performance** for the strongest models. This pattern is inconsistent with a contamination-driven explanation.
>
> These results suggest that pre-training contamination is unlikely to be the dominant factor shaping model performance. Variations in task difficulty and model–task capability alignment provide a more plausible explanation for the observed differences across years.
>
>
>
> Action: We have added the papers selcetion criterion in Section 3.1 and the disscusion about the data contamination in Appendix C.  We have reveal the paper list in Appendix K.

---

> ### Author Response · Authors · 2025-11-23
> **Response Weakness 2**
>
> The reviewer raises an insightful and much-appreciated question. Originally, we designed two **code-editing tools** for modifying code files. The first was **EDIT**, which allowed partial code replacement by specifying a *start line number*, an *end line number*, and the *new code snippet*. This tool was intended to enable more efficient and localized edits compared to rewriting the entire file.
>
> However, during evaluation, we observed that LLMs often struggled to correctly identify and modify the appropriate code regions when using **EDIT**. This frequently resulted in misplaced edits, compilation errors, and reduced overall system robustness. Because of these reliability issues, we ultimately relied solely on the **REPLACE** tool. This tool overwrites the entire file with newly generated code, ensuring correctness at the cost of being less granular than partial editing.
>
> **Action:** We have supplemented this explanation in Appendix D in the revision as follows:
> > L.790 :"
> > The REPLACE tool rewrites the entire contents of the target file with newly generated code. Although earlier versions of the system also included a span based EDIT tool for detailed modifications, empirical evaluation showed that models frequently misidentified edit boundaries or produced inconsistent patches, resulting in structural corruption. Whole file rewriting via REPLACE consistently yielded more reliable outputs. Therefore, REPLACE is retained as the sole editing mechanism.
> > "

---

> ### Author Response · Authors · 2025-11-23
> **Response Weakness 3**
>
> Thank you for the insightful comment. We clarify that our study **does include experiments using real human feedback**. As shown in **Figure 4** and detailed in **Appendix D**, we collected **50 intermediate human–LLM interaction samples**, where human annotators provided feedback on code that do not pass the tests. The model then revised the code using the same iterative protocol as with LLM-generated feedback.
>
> Results demonstrate that **human feedback is often partial, noisy, and lacks consistent corrective detail**, leading to **smaller performance gains** compared with LLM-generated expert-style feedback. This finding is consistent with prior work, such as **MINT** and **ConvCodeWorld**.
>
> We did not extend human annotation to the entire generation trajectory due to:
>
> 1. High annotation cost and expert labor requirements
> 2. Large variability in human expertise and feedback granularity, making fair and reproducible comparison difficult across trials
>
> Thus, we adopt **LLM-generated feedback as a controlled and repeatable expert reference**, enabling clearer analysis of feedback-driven improvements. We agree that broader human-based evaluation is valuable and consider it an **important direction for future work**.
>
> **Action:** We have added the analysis of difference between human feedback and llm generated feedback in Appendix E.

---

> ### Author Response · Authors · 2025-11-23
> **Response Weakness 4**
>
> What distinguishes **RECODE** from prior research code benchmarks like **SciReplicate-Bench** and **ResearchCodeBench** is that these benchmarks evaluate models only in a **single-shot, non-interactive** setting. In contrast, **RECODE** is the first benchmark that:
>
> 1. Adapts multi-turn, feedback-driven evaluation specifically to research code implementation.
> 2. Implements a reproducible feedback hierarchy tailored to research workflows .
> 3. Evaluates agents’ ability to iteratively refine complex research methods across repository level tasks, rather than isolated function-completion problems.
>
> This multi-turn design is not merely a convenience but a fundamental requirement. Real scientific workflows rely on repeated cycles of testing, diagnosis, and revision in response to partial failures and evolving constraints. By modeling this process, RECODE becomes the first benchmark to systematically evaluate iterative, feedback-driven research-code development, allowing us to assess a qualitatively distinct capability—how effectively LLMs manage the iterative debugging and refinement that real scientific practice demands.
>
> **Action:** We have added the analysis RECODE contribution in Section 2

---

> ### Author Response · Authors · 2025-11-23
> **Response Weakness 5**
>
> Thank you for the question. Yes, our evaluation includes a single-turn setting. The first-turn results shown in Figure 2 (Section 5.2) correspond to a condition with no interactive feedback, where the model receives only the initial task prompt. This serves as the single-turn baseline.
>
> As the results illustrate, performance after 10 rounds of interaction improves substantially over this baseline across all LLMs, demonstrating that multi-turn feedback provides a significant uplift in both correctness and convergence.
>
> In the revision, we have highlighted these discussions in Section 5.2.
>
> > L.404 :"
> > In addition, the first-turn results shown in Figure 2 also serve as our single-turn baseline. In this initial turn, the model receives only the task prompt without any interactive feedback, corresponding to the highlighted single-turn setting. As the figure indicates, subsequent turns that incorporate richer feedback yield substantial improvements over this baseline. After 10 turns, all models demonstrate markedly higher pass rates, confirming that multi-turn interaction provides a significant uplift beyond what is achievable in a single-turn scenario.
> > "

---

> > ### Comment · Reviewer_6h4u · 2025-11-25
> > **Rebbutal response**
> >
> > Thank you for the detailed responses to address my concerns. Appreciate the additional analyses on understanding any potential leakage -- it would be interesting to check why the models perform better on more recent papers (e.g., what kind of tasks or novelty are in these papers).
> > The issue with exact search and replace tool is a pretty common issue faced in the community. Thanks for the honest discloure.
> > I've raised my score from 6 to 8.

---

> ### Author Response · Authors · 2025-11-27
> **Official Comment by Authors**
>
> Dear Reviewer 6h4u,
>
> Thank you for your thoughtful followup and for updating your score.
>
> Regarding your question about why models perform better on more recent papers. Our dataset exhibits clear shifts in domain composition over time, which likely influence model performance. For example, from 2023→2025 we observe substantial growth in domains that modern LLMs are particularly strong at, such as Computer Vision & Multimodal (11.76% → 23.33%) and Diffusion & Generative Models (0% → 20.00%). In contrast, categories where models tend to struggle, such as Graph Learning and GNN, decrease in representation (29.41% → 13.33%). These domain shifts provide a plausible explanation for improved performance on more recent papers.
>
> By comparison, our task level difficulty metrics, such as average lines of code, number of required functions/classes remain consistent across years. Thus, the trend is more reasonably attributed to domain alignment effects rather than differences in task difficulty or pre-training contamination.
>
> We appreciate your careful review and constructive feedback.
>
> Best regards,
>
> The Authors

---

### Author Response · Authors · 2025-12-01
**Summary of the rebuttal outcomes**

Dear Area Chair,

Below is a concise summary of how the rebuttal addressed each reviewer’s concerns, whether the reviewer indicated satisfaction, and how the paper PDF was updated accordingly.

---

### **Reviewer 6h4u — Reviewer was satisfied with the rebuttal，score raised: 6 → 8 (prior to the November 27th incident)**

**Concerns:** paper selection transparency; potential pre-training contamination; EDIT vs REPLACE tool usage; missing human-feedback results; novelty vs prior benchmarks; clarification of single-turn baseline.
**Rebuttal:** provided detailed paper selection criteria and full paper list; added contamination analysis; clarified tool design; included human feedback experiments; highlighted benchmark novelty; formally explained single-turn baseline.
**PDF updates:** added selection protocol (Sec. 3.1), contamination analysis (App. C), tool explanation (App. D), human-feedback analysis (App. E), and contribution clarification (Sec. 2).
**Reviewer’s stance:** reviewer confirmed concerns were resolved and raised score to **8**.



---

### **Reviewer AynR — Reviewer was satisfied with the rebuttal，score remained: 6 (prior to the November 27th incident)**

**Concerns:** realism of simulated feedback; need for deeper insights; Level-4 performance ceiling; user guidance; scalability of annotation.
**Rebuttal:** compared human vs LLM feedback; articulated key non-obvious findings (semantic grounding bottlenecks, non-linear feedback gains, model-dependent feedback uptake); clarified Level-4 limitations with extended turn results; strengthened main conclusions; described semi-automated pipeline and future automation.
**PDF updates:** added deeper insights to Conclusion (Sec. 6), extended-turn analysis (App. H), and expanded human-feedback discussion (App. E).
**Reviewer’s stance:** reviewer expressed satisfaction with updates and kept the positive score **6**.

---


### **Reviewer 61B7 — Reviewer was satisfied with the rebuttal，score raised: 4 → 8 (prior to the November 27th incident)**

**Concerns:** unclear annotator roles; reliability of tests; missing paper list; contamination discussion; validity of Level-4 feedback; need for example tasks; citation-related ethics concern.
**Rebuttal:** clarified expert annotator responsibilities; explained that tests are human-authored with rigorous coverage; released full paper list; added contamination discussion; justified Level-4 as realistic code-review feedback; added example tasks; resolved attribution concerns.
**PDF updates:** expanded annotation workflow (Sec. 3.1), clarified test design, added contamination results (App. C), Level-4 justification (Sec. 3.2), and example tasks (App. N).
**Reviewer’s stance:** reviewer stated all concerns were resolved, removed ethics flag, and raised score to **8**.

---

### **Reviewer oRzg — Reviewer did not provide a follow-up response，remained: 2**

**Concerns:** positioning relative to prior ReAct-based work; novelty of agent design; absence of evaluation on external benchmarks; distinction from SUPER; interpretation of “one-shot.”
**Rebuttal:** clarified meaning of “one-shot”; distinguished RECODE as the first benchmark for structured multi-turn researcher feedback; positioned ReCodeAgent as a baseline adapted to hierarchical feedback; explained limitations of existing benchmarks for this evaluation; added comparison to SUPER.
**PDF updates:** strengthened related-work comparison (Sec. 2), clarified agent design (Sec. 4), and added discussion on SUPER.
**Reviewer’s stance:** no follow-up comment was posted.

---

### **Overall Summary**
Three reviewers explicitly indicated that their concerns were resolved during rebuttal. All substantive clarifications and additions requested across the reviews have been incorporated into the **updated PDF**.

Warm regards,

The Authors

---

### Meta-Review · Area_Chair_Q8P9 · 2026-01-09

**Summary:**

Across reviews, the central question was whether RECODE offers a substantial contribution as a new benchmark setting beyond existing research-code and agentic coding benchmarks, and whether the accompanying ReCodeAgent is more than a standard ReAct-style baseline. The rebuttal substantially improved clarity on dataset construction (paper selection protocol, released paper list, annotator roles, and test design), clarified the intended meaning of one-shot, and added targeted analyses (human vs LLM feedback samples, year-based contamination sanity checks, and extended-turn results).

Whether this benchmark contribution is incremental or substantive, the discussion suggests the work is not purely incremental as a benchmark contribution, since the structured multi-turn researcher feedback hierarchy and repository-level research implementation framing appear to be the distinctive piece. However, the agent contribution is incremental and largely a ReAct baseline adapted to consume the benchmark’s feedback hierarchy. The remaining weakness is that the strongest skeptic (oRzg) asked for deeper positioning and broader validation outside RECODE, and the rebuttal mainly argues why external benchmarks are not directly compatible rather than providing additional experimental evidence.

**Reviewer Concerns:**

Questions about how papers were selected and whether the paper list was disclosed were answered with explicit criteria and a commitment to release the full list in the appendix. Confusion about editing tools was resolved with a concrete explanation of why span-based editing was dropped in favor of whole-file replacement due to reliability failures. Concerns about the realism of feedback and the absence of human feedback results were addressed by pointing to a small-scale human-feedback experiment and by explicitly acknowledging that human feedback is noisier and yields smaller gains than LLM-generated expert-style feedback. The validity of Level 4 feedback was addressed with a clear framing as an upper bound for integration capability and with additional evidence showing performance remains below 100 percent under turn limits, including extended-turn results approaching saturation. Concerns about annotator roles, evaluation reliability, applicability of tests to training-loop tasks, examples, and attribution were addressed, to the point that score were increased.

The remaining concerns are concentrated in the objections raised by the strongest critic. The rebuttal clarifies that one-shot refers to the human-agent interaction pattern rather than the agent's internal ReAct loop, and it positions the benchmark novelty as structured researcher feedback rather than generic multi-turn act-observe-reflect. That explanation improves the narrative, but it does not fully close the loop on whether the paper's novelty claims are adequately substantiated through explicit side-by-side protocol comparisons with the closest prior work.
Similarly, the request for broader empirical validation beyond RECODE is answered largely by an argument that other benchmarks lack the needed research-paper alignment and feedback annotations, rather than additional experiments or ablations that would directly satisfy the  concern. The contamination discussion is improved, but the year-based trend remains an indirect proxy, and the 2023 as cut-off is already probably old for frontier models.

**Reviewer Scores:**

Reviewer 6h4u already raised their score from 6 to 8 and 61B7 also raised their score from 4 to 8 and removed the ethics flag.
Reviewer AynR expressed satisfaction with clarifications and updates but did not indicate a score change; it is plausible they would keep their score around 6. Reviewer oRzg did not follow up, and given the nature of their critique, it is likely their score would remain close to a reject unless new empirical evidence or sharper comparative positioning were added.

---

### Decision · Program_Chairs · 2026-01-26

Accept (Poster)